# USP22 regulates lipidome accumulation by stabilizing PPARγ in hepatocellular carcinoma

Zhen Ning[1,2,3,6], Xin Guo[1,2,6], Xiaolong Liu [1,6], Chang Lu[1,2,6], Aman Wang[2,3], Xiaolin Wang[1], Wen Wang[1], Huan Chen[1], Wangshu Qin[1], Xinyu Liu[1], Lina Zhou[1], Chi Ma[2,3], Jian Du[2,3], Zhikun Lin[1,2,3], Haifeng Luo[2,3], Wuxiyar Otkur[1], Huan Qi[1], Di Chen[1], Tian Xia[1], Jiwei Liu[2,3], Guang Tan[2,3✉], Guowang Xu [1✉] & Hai-long Piao [1,4,5✉]

Elevated de novo lipogenesis is considered to be a crucial factor in hepatocellular carcinoma (HCC) development. Herein, we identify ubiquitin-specific protease 22 (USP22) as a key regulator for de novo fatty acid synthesis, which directly interacts with deubiquitinates and stabilizes peroxisome proliferator-activated receptor gamma (PPARγ) through K48-linked deubiquitination, and in turn, this stabilization increases acetyl-CoA carboxylase (ACC) and ATP citrate lyase (ACLY) expressions. In addition, we find that USP22 promotes de novo fatty acid synthesis and contributes to HCC tumorigenesis, however, this tumorigenicity is suppressed by inhibiting the expression of PPARγ, ACLY, or ACC in in vivo tumorigenesis experiments. In HCC, high expression of USP22 positively correlates with PPARγ, ACLY or ACC expression, and associates with a poor prognosis. Taken together, we identify a USP22-regulated lipogenesis mechanism that involves the PPARγ-ACLY/ACC axis in HCC tumorigenesis and provide a rationale for therapeutic targeting of lipogenesis via USP22 inhibition.

[1] CAS Key Laboratory of Separation Science for Analytical Chemistry, Dalian Institute of Chemical Physics, Chinese Academy of Sciences, Dalian 116023, China. [2] The First Affiliated Hospital of Dalian Medical University, Dalian Medical University, Dalian 116000, China. [3] Liaoning Key Laboratory of Molecular Targeted Drugs in Hepatobiliary and Pancreatic Cancer, Dalian 116000, China. [4] University of Chinese Academy of Sciences, Beijing 100049, China. [5] Department of Biochemistry & Molecular Biology, School of Life Sciences, China Medical University, Shenyang 110122, China. [6]These authors contributed equally: Zhen Ning, Xin Guo, Xiaolong Liu and Chang Lu. ✉email: guangtan@dmu.edu.cn; xugw@dicp.ac.cn; hpiao@dicp.ac.cn

The liver acts as a pivotal organ for the metabolism of the three macronutrients, namely, sugar, lipid, and protein. Various liver diseases, including hepatocellular carcinoma (HCC), are related to metabolic abnormalities[1]. Of note, the global incidence of HCC is increasing rapidly, which is partly due to the epidemic of obesity and subsequent development and progression of metabolic-associated fatty liver disease (MAFLD)[2]. More importantly, the lipid-rich state is an important characteristic of obesity- and MAFLD-driven HCC[1,3]. HCC is a major cause of top five cancer-related deaths worldwide (https://gco.iarc.fr/resources). HCC is a highly heterogeneous cancer, and during last decade scientists clearly defined the landscape of genetic alterations in HCC, such as TERT, TP53 and CTNNB1 mutation, and amplification of VEGFA and FGF19/CCND1[4]. However, the molecular mechanism of these series of disease processes is still unclear.

Elevated de novo lipogenesis (DNL) is a crucial factor in the development of MAFLD[5] and cancers, including HCC[6]. The abnormal increase in the de novo synthesis of fatty acids is the key link of DNL[7]. Fatty acids and lipidomes function as key signaling factors, energy source, and building block of the cell membrane and play critical roles in cell proliferation. The de novo synthesis of the fatty acid pathway converts citrate to acetyl-coenzyme A (CoA), malonyl-CoA and finally palmitic acid by ATP citrate lyase (ACLY), acetyl-CoA carboxylase (ACC) and fatty acid synthase (FASN). Subsequently, after elongation, saturated fatty acids are converted to monounsaturated fatty acids by stearoyl-CoA desaturase (SCD), which is the preferred substrate for triglyceride (TG) generation[7,8]. Importantly, the expression of ACLY, ACC[9], FASN[6,10] and SCD[11] is upregulated and has been associated with poor prognosis in HCC. Small interfering RNAs (siRNAs) or small-molecule inhibitors targeting these proteins have effectively inhibited cell aggressiveness both in vitro and in vivo[12,13]. However, to date, there is limited progress on clinical treatments targeting lipid synthesis due to toxicity or complications[14]. Therefore, it is particularly important to find more effective targets for fatty acid synthesis.

Dysregulation of metabolic enzyme expression or activity is among the key causes of metabolic reprogramming. Ubiquitination and deubiquitination are crucial posttranslational modifications of metabolic enzymes that regulate their degradation, delocalization and activation in cells by tagging or removing ubiquitins from substrate proteins[15,16]. The dysregulation of ubiquitination and deubiquitination is closely related to cell lipid metabolism and promotes the occurrence and progression of various cancers, including HCC. In recent years, due to the potential roles of stabilization of oncoproteins, deubiquitinating enzymes (DUBs) have been widely studied in several different cancers. Previously, we conducted a study in the field of deubiquitination and successfully mapped the key deubiquitination molecular interaction networks in different cancers[17]. The USP family is a well-defined deubiquitinating enzyme sub-family, and its members, USP14[18] and USP30[19] play key roles in lipogenesis by directly deubiquitinating and stabilizing the key lipogenesis factors ACLY and FASN in tumorigenesis and high-fat diet-driven HCC. USP22 has been reported to be a cancer signature gene and is highly expressed in a variety of cancers. USP22 mainly plays the role of oncoprotein in chromatin remodeling by removing ubiquitin from histones (H2B and H2A) and subsequently activates or stabilizes transcription factors in cancer progression[20,21]. Additionally, USP22 participates in cell cycle regulation and apoptosis by deubiquitinating Cyclin D1[22], Cyclin B1[23], and SIRT1[24] in cancer cell proliferation. In addition, USP22 has been found to deubiquitinate and stabilize PDL1 and thus cause cancer immune resistance[25]. However, whether USP22 regulates lipogenesis has not been identified.

Peroxisome proliferator-activated receptor gamma (PPARγ) is a ligand-activated transcription factor belonging to the nuclear hormone receptor family and promotes lipogenesis by upregulating lipid synthesis enzymes, including ACC, ACLY and FASN[26–28]. PPARγ exists in two isoforms, PPARγ1 and PPARγ2 (28 amino acids extended at its N-terminal than PPARγ1). PPARγ1(here after PPARγ) is widely expressed in the liver, colon, immune system and hematopoietic cells, while PPARγ2 is selectively expressed in adipose tissue. In addition, PPARγ as transcription factor promotes metabolic adaptations of lipogenesis and aerobic glycolysis in liver cancer[29,30]. However, the upstream role of PPARγ in cancer biology is poorly characterized. Recent studies revealed that the phosphorylation of AKT2 promotes the expression of PPARγ and thereby promotes lipid synthesis and tumorigenesis in HCC[31]. In this study, we found that the abnormally high expression of USP22 is accompanied by a significant upregulation of lipid synthesis in HCC. We further demonstrated that USP22 promotes de novo synthesis of fatty acids and tumorigenesis by deubiquitinating PPARγ in HCC. Our findings provide an option for targeting fatty acid synthesis that may provide therapeutic benefits to patients with high USP22 expression in HCC.

## Results

**Abnormal lipid metabolism in human HCC with high USP22 expression.** Metabolic reprogramming is a characteristic event in the occurrence of HCC[12]. To describe the disordered metabolism in HCC, high-throughput metabolomics was performed to detect the differentially altered metabolites in 10 paired HCC and adjacent normal tissues (Fig. 1a). Compared with the adjacent normal tissues, 47 metabolites are significantly altered in the cancer tissues (Fig. 1b). Among upregulated metabolites (22 out of 26) are mostly lipids and lipid-like metabolites (Fig. 1b), which are mainly composed of fatty acids (FAs), phosphatidylcholine (PC), lysophosphatidylcholine (LPC), phosphatidylethanolamine (PE), lysophosphatidylethanolamine (LPE) and sphingomyelin (SM) (Fig. 1c). A further pathway enrichment analysis using 47 differential metabolites showed that the biosynthesis of PC, cardiolipin, phospholipids, PE and triacylglycerol were enriched in cancer tissues (Fig. 1d). These results confirmed that the contents of lipidome were significantly upregulated in HCC tissues, but what caused the abnormal upregulation of lipidome in HCC tissues remains unclear.

Previously, we conducted a bioinformatics analysis of USPs based on The Cancer Genome Atlas (TCGA) database and found that the abnormal expression of USPs was highly associated with poor prognosis in HCC (Supplementary Fig. 1a)[17], suggesting that USPs might play an important role in HCC progression. To determine whether the abnormally upregulated lipidome in HCC was related to the expression of USPs, we examined the expression of USPs in the above 10 pairs of cancer and adjacent normal tissues and found that USP22 was the most significantly high expressed USP member (Fig. 1e, f; Supplementary Fig. 1b), and this high expression USP22 was significantly correlated with lipids and lipid-like metabolites upregulation (Fig. 1g). Subsequently, gene set enrichment analysis (GSEA) based on the TCGA database revealed that USP22 was significantly associated with unsaturated fatty acids biosynthesis in HCC (Supplementary Fig. 1c). In addition, we analyzed the USP22 expression based on the TCGA pan-cancer transcriptomics data (http://gepia2.cancer-pku.cn/#index) and found that USP22 was highly expressed in HCC and cholangiocarcinoma but not in the other cancer types (Supplementary Fig. 2)[25]. Altogether, these data demonstrate that high expression of USP22 might be associated with the abnormal upregulation of lipid and lipid-like metabolites in HCC.

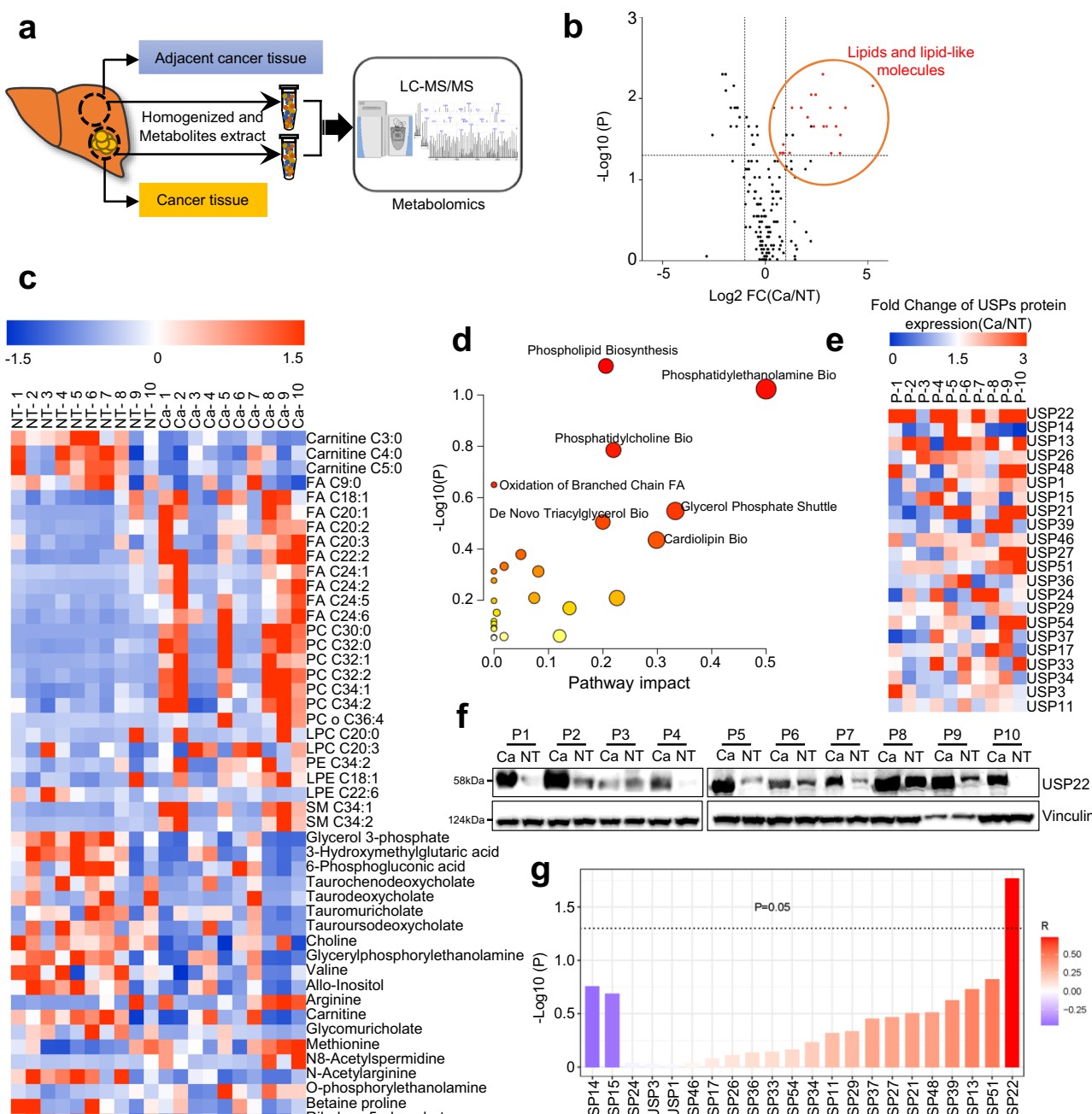

**Fig. 1 Abnormal lipid metabolism in human HCC with high USP22 expression. a** LC-MS–based nontargeted metabolomic analysis detecting differential metabolites between the tumor tissues and adjacent normal tissues (More than 2 cm from the edge of the tumor) of patients with HCC ($n = 10$, 6 female and 4 male patients, the age range is between 48 and 60). All patients were diagnosed with HCC by postoperative pathology and were free of other cancers and chronic diseases. **b** Volcano plots of metabolites in HCC and normal adjacent tissue. Red represents lipids and lipid-like molecules ($n = 22$). LC-MS-based nontargeted metabolomic analysis, and the data were corrected by total peak area. **c**, Heatmap analysis of significantly changed metabolites ($n = 47$) in cancer tissues (Ca) compared to paired normal adjacent tissue (NT). $p < 0.05$, paired two-sample Wilcoxon test. Red indicates increase, and blue indicates decrease. -1.5~1.5 indicates the Fold Change. **d** Enriched metabolic signaling pathways based on significantly changed metabolites ($n = 47$) cluster identified by pathway analysis (https://www.metaboanalyst.ca/). **e** Heatmap analysis of the fold change (Ca/NT) of USPs (which are related to the prognosis of HCC based on the TCGA HCC database) protein expression in the above cancer tissues and normal adjacent tissues in Supplementary Fig. 1b. Image j was used for quantification of western blot. Red indicates increase, and blue indicates decrease. **f** Western blotting of USP22 in ten pairs of matched adjacent non-tumor (NT) and cancer (Ca) tissues ($n = 10$). **g** Correlation analysis of USPs protein expression and lipid metabolite content (sum: Carnitine, FA, LPC, PC, LPE, PE and SM) in cancer tissues. R represents the Pearson correlation coefficient. Source data are provided in the Source Data file.

Nonetheless, the specific role of USP22 in lipid metabolism in HCC is still unclear.

**USP22 promotes lipid accumulation and tumorigenesis in HCC cells.** To define the specific role of USP22 in HCC, we first analyzed the Cancer Cell Line Encyclopedia and revealed the ninth highest expression of USP22 in liver cancer cells among 1,457 human cancer cell lines from 40 tumor types (Supplementary Fig. 3a). Additionally, USP22 protein expression was more abundant in the HCC cell lines than in normal liver epithelial cells (THLE-2) (Supplementary Fig. 3b). We observed that USP22 expression was highest in MHCC-97H cells and lowest in MHCC-97L cells, and relatively low in SNU449, HepG2, Bel-7402 and HUH7 cells. Subsequently, to exclude the cell line specificity we generated a USP22 stably knocked down cell line in MHCC-97H and transduced overexpression cell lines in MHCC-97L, Bel-7402, SNU449, HUH7 and HepG2 cells, respectively (Supplementary Fig. 3c).

Next, we found that 79 and 73 metabolites were significantly altered in MHCC-97H-shUSP22 and MHCC-97L-USP22 cells (Supplementary Fig. 3d, e). Indeed, lipid-related metabolites were altered in total (51/79) 64.6% (Supplementary Fig. 3d) and (39/73) 53.4% (Supplementary Fig. 3e) in USP22 stably knocked down and overexpressing cell lines, respectively. Subsequently, we identified 53 metabolites were significantly altered in USP22 engineered HCC cells (Fig. 2a, b). Among them, fatty acid metabolites were significantly decreased in USP22-knockdown MHCC-97H cells and increased in USP22-transduced MHCC-97L cells (Fig. 2b). Furthermore, based on these 34 metabolites we performed pathway enrichment analysis and showed that USP22 was remarkably related to the fatty acid synthesis pathways (Fig. 2c). Glucose is an important synthetic source for the de novo synthesis of fatty acids (Supplementary Fig. 4a). Subsequently, we used [U-$^{13}$C] glucose to trace fatty acid synthesis and observed that the USP22 knockdown cells significantly decreased palmitic acid and stearic acid labeling from glucose tracers (Fig. 2d; Supplementary Fig. 4b). Conversely, the USP22-overexpressing MHCC-97L cells exhibited significantly increased palmitic acid and stearic acid labeling from glucose (Fig. 2d; Supplementary Fig. 4b). In addition, USP22 significantly promoted the accumulation of triglycerides in HCC cells (Fig. 2e, f). We also performed similar experiments in SNU449, HepG2, HUH7 and Bel-7402 HCC cell lines. We observed that a large number of fatty acids and triglycerides were significantly upregulated in the USP22-overexpressing HCC cells (Supplementary Fig. 4c–e). Taken together, these results provide evidences that USP22 plays a key role in the accumulation of lipid content, and may promotes the de novo synthesis of fatty acids in HCC cells.

Next, to verify the tumorigenesis function of USP22, we performed both loss-of-function and gain-of-function analyses in HCC cell lines. Two independent USP22 shRNAs both decreased the colony-forming ability in soft agar (Supplementary Fig. 4f). In contrast, overexpression of USP22 in MHCC-97L, HUH7, HepG2, Bel-7402 and SUN449 cells significantly promoted colony formation (Supplementary Fig. 4f). To explore the function of USP22 in HCC cells in vivo, we subcutaneously implanted USP22-depleted MHCC-97H and USP22-overexpressing Bel-7402 cells into nude mice (Fig. 2g; Supplementary Fig. 4g). Hosts of USP22 shRNA-expressing MHCC-97H cells had smaller tumor volumes and weights than mice implanted with control shRNA-infected cells (Fig. 2h, i). Conversely, overexpression of USP22 increased the tumorigenic ability of Bel-7402 cells (Supplementary Fig. 4h, i). Meanwhile, a large number of lipid metabolites, including fatty acids were significantly decreased in the tumors derived from USP22-depleted MHCC-97H cells compared to shControl cells-derived tumors (Fig. 2j). Overall, our data confirm that USP22 plays an important role in HCC lipid accumulation and tumorigenesis.

**USP22 upregulates ACC and ACLY expression.** To investigate whether key enzymes in fatty acid synthesis were affected by USP22 in HCC cells, we performed high-throughput RNA sequencing analysis. Among the top 30 differentially expressed genes, previously reported FA biosynthesis-related genes, such as ACLY, emerged as responsive genes sensitive to USP22 depletion (Fig. 3a). GO-biological process (GO-BP) analysis confirmed that USP22 knockdown was associated with lipid biosynthesis process (Supplementary Fig. 5a). Follow-up GSEA identified that the gene sets of fatty acid biosynthesis were significantly enriched by both two independent USP22-shRNA-transduced MHCC-97H cells, however, the fatty acid degradation pathway was not (Fig. 3b; Supplementary Fig. 5b). Abnormal upregulation of fatty acid anabolism is closely related to the key metabolic enzymes of ACLY, ACC, FASN and SCD[8]. To confirm that USP22 regulates the fatty acid metabolic genes, we performed qPCR and immunoblot analysis of USP22-knockdown and USP22-overexpressing HCC cells. Both mRNA and protein expressions of ACC and ACLY were decreased in the USP22-knockdown MHCC-97H cells but increased in the USP22-overexpressing MHCC-97L, HUH7, HepG2, SNU449 and Bel-7402 cells (Fig. 3c, d; Supplementary Fig. 5c, d). However, the expressions of FASN and SCD were not affected by USP22 in the HCC cells (Fig. 3c, d; Supplementary Fig. 5c, d). To further investigate whether USP22 is involved in the regulation of fatty acid degradation, we examined the expression of fatty acid degradation-related enzymes (CPT1A, CPT2, ACOX1, ACADL, and ECHS1) in these USP22-engineered stable cells by using qRT-PCR and western blot, and found that USP22 did not affect the expression of these enzymes in HCC cells (Supplementary Fig. 5d–f).

We next sought to explore the correlation between USP22 and key fatty acid metabolic enzymes in HCC tissues. USP22 was significantly positively correlated with ACLY and ACACA (the gene encoding ACC) but not with FASN and SCD in the TCGA HCC database (Fig. 3e; Supplementary Fig. 5g), which was consistent with the transcripts expression results. To confirm whether USP22 promotes lipidome accumulation through ACC and ACLY, we transduced USP22 into ACC or ACLY knockdown MHCC-97L and SNU449 cells (Supplementary Fig. 5h, i) and found that transduction of USP22 could not promote intracellular fatty acid and TG accumulation in ACC or ACLY knockdown cells (Fig. 3f, g). Carnitine palmitoyltransferase (CPT) is the key enzyme for fatty acid degradation. Therefore, we further analyzed the correlation between USP22 and CPT in the TCGA HCC database. The results revealed that USP22 did not correlate with CPT1A and weakly correlated with CPT2 ($R = -0.153$, $p = 0.003$) (Supplementary Fig. 5j). Taken together, these results indicate that USP22 promotes lipidome accumulation by increasing the expression of ACC and ACLY in HCC, rather than inhibiting fatty acid degradation.

**USP22 specifically interacts with lipid metabolism key transcription factor of PPARγ.** Next, to identify the specifically targeted deubiquitinating substrate of USP22 which can regulate ACC and ACLY expression, we isolated the USP22-associated protein complex in HEK293T cells through tandem affinity purification followed by mass spectrometry analysis. Interestingly, PPARγ was identified on the prey list (Fig. 4a; Supplementary Table 1). Furthermore, we found that PPARγ is the only counterpart among several lipid metabolism-related transcription factors, including PPARα, PPARδ and SREBF1 (Fig. 4b).

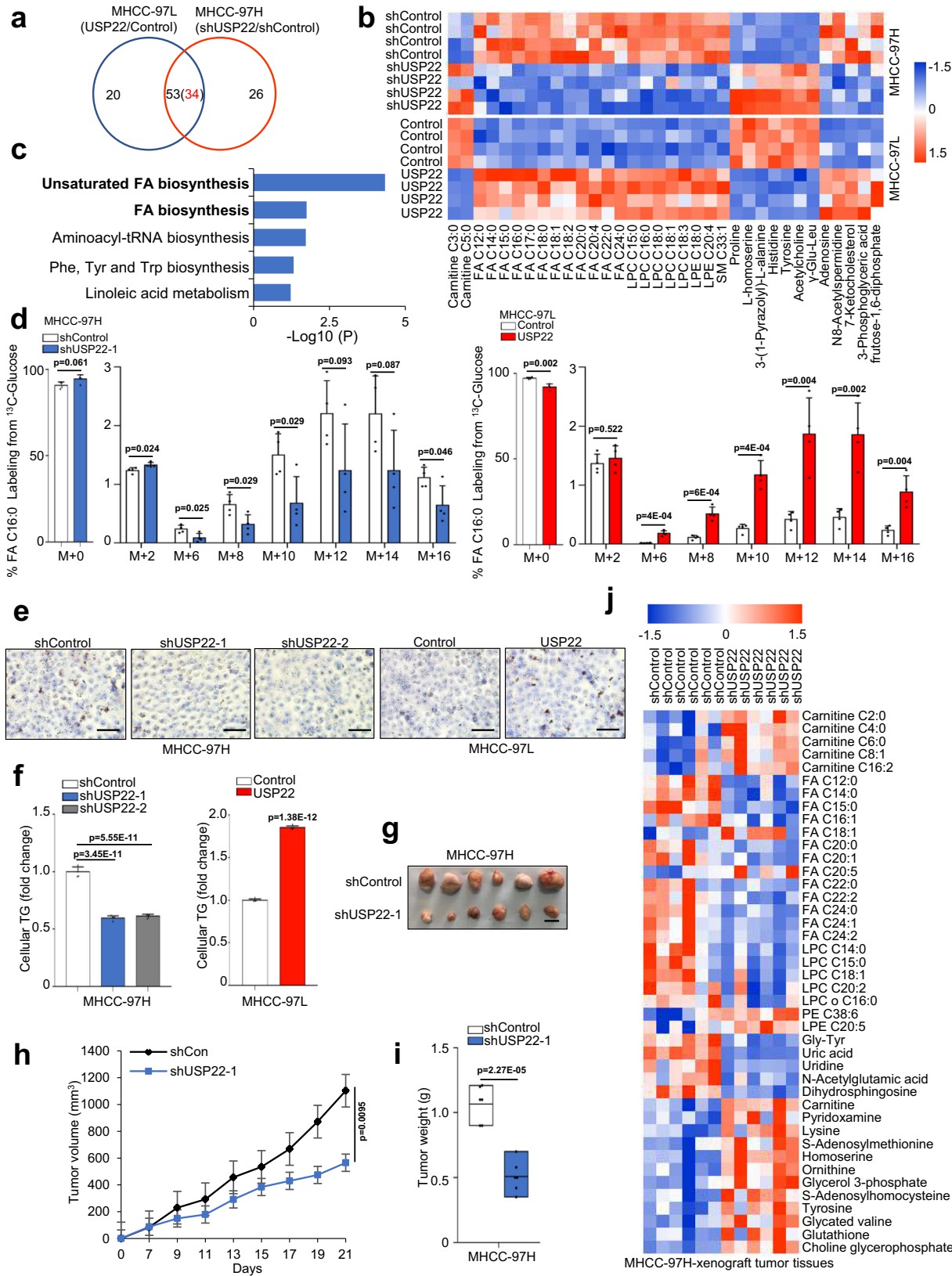

Moreover, endogenous PPARγ was present in endogenous USP22 immunoprecipitates from MHCC-97L, HUH7, HepG2, SNU449, and Bel-7402 cells (Fig. 4c). In vitro pulldown assays with purified recombinant proteins demonstrated that USP22 directly bound to PPARγ (Fig. 4d). Then, the specific interaction between USP22 and PPARγ was confirmed by co-immunoprecipitation assay with exogenously transduced USP22 and PPARγ in HEK293T cells (Fig. 4e). The immunofluorescent staining assay showed that USP22 and PPARγ were mainly colocalized in the nucleus in HCC cells (Fig. 4f). It has been reported that PPARγ contains three structural domains: AF-1, DBD and Hinge-LBD (Fig. 4g). Pull-down assays showed that USP22 was strongly

**Fig. 2 USP22 promotes lipid accumulation and tumorigenesis in HCC cells. a** Venn diagram of the overlap between USP22-knockdown (shUSP22-1 vs shControl) and USP22-overexpression (USP22 vs Control)-induced differential metabolites (The heat map of differential metabolites is shown in Supplementary Fig. 3d and 3e, $p < 0.05$, unpaired two-tailed Student's $t$-test). The red number 34 refers to the number of metabolites with the opposite trend. LC-MS-based nontargeted metabolomic analysis, and the data were corrected by total peak area. **b** Heatmap analysis of these 34 significantly changed metabolites. $p < 0.05$, unpaired two-tailed Student's $t$-test. Red indicates increase, and blue indicates decrease. -1.5~1.5 indicates the Fold Change. **c** Enriched signaling pathways identified by pathway analysis based on these 34 differential metabolites. (https://www.metaboanalyst.ca/). **d** The percentages of various isotopomers of FA 16:0 (palmitate) after trace to [U-$^{13}$C] glucose in MHCC-97H-shUSP22-1-and MHCC-97L-USP22cells. Medium was changed to RPMI 1640 containing [U-$^{13}$C] glucose (2 g/L) when the cell density was about 80%, and 24 h later cell culture plates were washed with PBS and snap-frozen in liquid nitrogen and subjected to LC-MS analysis. unpaired two-tailed Student's $t$-test. The data shown represent the means (±SD) of biological replicates. The experiments were repeated four times ($n = 4$). **e** Representative images of Oil red staining assay in MHCC-97H-shUSP22-1/2 cells and MHCC-97L-USP22 cells. Cells were analyzed after 24 h adherence. Scale bars, 50 μm. **f** Relative content of TG was analyzed in MHCC-97H-shUSP22-1/2 cells and MHCC-97L-USP22 cells. Cells were analyzed after 24 h adherence. The data shown represent the means (±SD) of biological replicates. The experiments were repeated five times ($n = 5$). One-way ANOVA test. **g–i** MHCC-97H-shUSP22-1 or shControl cells were injected into the right flank of nude mice. Tumor volumes were measured every 3 days. Tumor images (**g**), growth curves (**h**) and weight (**i**) were obtained at day 21 after dissection. Data in h are presented as mean values ± SD and data in **i** are presented as mean values with minima and maxima. unpaired two-tailed Student's $t$-test. Scale bars, 1 cm. $n = 6$ biologically independent tumor samples. **j** Heatmap analysis of significantly changed metabolites in MHCC-97H-shUSP22-1 cells-derived tumors compared to shControl cells-derived tumors based on nontargeted metabolomic analysis. Data were standardized by total peak area. $p < 0.05$, unpaired two-tailed Student's $t$-test. Source data are provided in the Source Data file.

associated with the DBD fragment and weakly interacted with the Hinge-LBD fragment of PPARγ (Fig. 4g). Based on the TCGA HCC database, a pathway enrichment analysis also showed that USP22 was significantly correlated with the PPAR pathway in HCC (Fig. 4h). Overall, our results confirm that USP22 specifically interacts with PPARγ, which is a key transcription factor related to lipid metabolism.

**USP22 deubiquitinates and stabilizes PPARγ.** To explore whether USP22 regulates the stability of the PPARγ protein, we performed deubiquitination assay. The polyubiquitination level of PPARγ was markedly increased in two independent USP22 shRNA knockdown MHCC-97H cells (Fig. 5a). In contrast, overexpression of USP22 increased deubiquitination of PPARγ but not the enzyme-dead mutant in MHCC-97L cells (Fig. 5b). Moreover, we found that USP22 significantly inhibited the Lys-48-linked polyubiquitination of PPARγ (Supplementary Fig. 6a, b). In addition, polyubiquitination of PPARγ was inhibited in USP22 highly expressed cancer tissues compared to adjacent normal tissues (Supplementary Fig. 6c). Tumors formed by USP22 knockdown MHCC-97H cells exhibited upregulation of PPARγ polyubiquitination compared to tumors formed by control shRNA expressing MHCC-97H cells (Supplementary Fig. 6d). Furthermore, USP22 significantly promoted the in vitro PPARγ deubiquitination (Fig. 5c). It has been reported that PPARγ interacts with the E3 ubiquitin enzymes of pVHL and CRL4B^AhR through DBD domain and is ubiquitinated by these enzymes through different lysine sites (K404/434 for pVHL, K240/265 for CRL4B^AhR) in cells and affected lipid metabolism[32,33]. Interestingly, we found that USP22 strongly bound to the PPARγ DBD domain (Fig. 4g) as well as pVHL and CUL4B proteins in the HCC cells (Supplementary Fig. 6e), and the interaction between PPARγ and USP22 significantly decreased the pVHL and CRL4B^AhR involved ubiquitination (Fig. 5d, e), indicating that USP22 regulates deubiquitination of PPARγ through other lysine sites. Next, to identify which lysine sites on the PPARγ are involved in USP22 deubiquitination process, we generated the several fragments of PPARγ based on different protein domains, and found that the domains which contained DBD domain could be deubiquitinated by USP22 (Fig. 5f). Consequently, we mutated all lysine sites on the DBD domain, and found the Lys-169 may be an important site for USP22 deubiquitination of PPARγ (Fig. 5g; Supplementary Fig. 6f). We also found that USP22 did not deubiquitinate the five lysine sites mutated PPARγ-5KR (K169/240/265/404/434 R)

(Fig. 5h). Additionally, we examined PPARγ protein levels in the presence of cycloheximide (CHX), an inhibitor of protein translation. Notably, overexpression of USP22, but not the knockdown of USP22 or enzyme-dead mutant (USP22 C185S), led to a prominent increase in the stability of endogenous PPARγ protein, whereas the stability of vinculin was not affected (Fig. 5i, j). In addition, we found that knockdown or overexpression of USP22 shifting the half-life time of endogenous PPARγ expression after 6 h CHX treatment in HCC cells (Fig. 5i, j). Taken together, these results indicate that USP22 directly interacts with PPARγ and functions as a bona fide PPARγ deubiquitinase in cells.

**USP22 increases ACC and ACLY expression by stabilizing PPARγ.** The expressions of *ACACA* and *ACLY* transcripts are regulated by lipid metabolism-related transcription factors, such as PPARγ. Given that USP22 is localized in nucleus (Fig. 4f), we hypothesized USP22 may control *ACLY* and *ACACA* expression by stabilizing PPARγ in HCC cells. To prove this hypothesis, we performed cytoplasmic and nuclear fractionation assay, and found that USP22 significantly affected the PPARγ expression in the nucleus (Fig. 6a). Furthermore, we found USP22 knockdown did not alter the expression of Flag-PPARγ-5KR in the nucleus (Supplementary Fig. 7a). In addition, high expression of PPARγ relatively correlated with USP22 expression in the nuclear fraction from HCC tissues (Supplementary Fig. 7b). The transcriptional activity assay of PPARγ revealed that USP22 significantly modulates the DNA binding activity of PPARγ in HCC cells (Fig. 6b). Furthermore, we confirmed that USP22 upregulates the transcriptional activity of PPARγ by the PPAR response element (PPRE) (Supplementary Fig. 7c). Besides, PPARγ may activate the transcription of *ACLY* and *ACACA* through directly binding to PPRE. To investigate this possibility, we performed chromatin immunoprecipitation (ChIP) assays in USP22 depleted MHCC97H cells, followed by qPCR of the conserved *ACLY* and *ACACA* promoter region encompassing one consensus PPARγ binding PPRE motif (Fig. 6c). ChIP with the PPARγ antibody was enriched for the *ACLY* and *ACACA* promoter region in control cells compared to the USP22 depleted cells (Fig. 6d). To confirm these results, we performed a dual-luciferase reporter assay with luciferase gene containing the *ACLY* and *ACACA* promoter regions. PPARγ significantly promoted the luciferase activity of the *ACLY* and *ACACA* promoters (Supplementary Fig. 7d). In addition, knockdown of USP22 significantly reduced the enrichment of promoter regions of *ACLY* and *ACACA* in Flag-PPARγ transduced

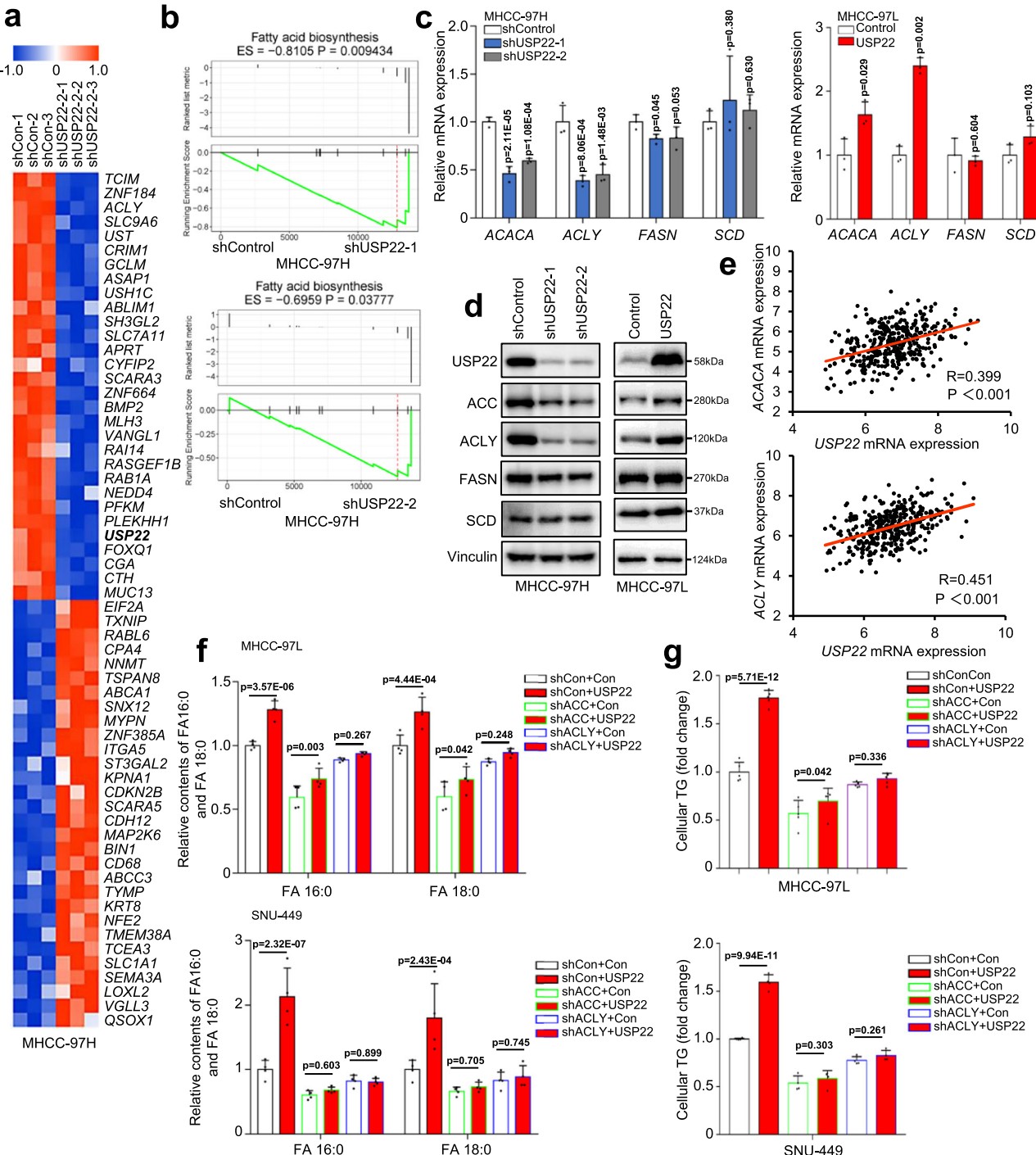

**Fig. 3 USP22 upregulates ACC and ACLY expression. a** Heatmap depicting the top 30 downregulated and upregulated genes in MHCC-97H cells transduced with USP22 shRNA ($Q < 0.05$). Red and blue represent the Log2 fold change of an increase or decrease in mRNA expression compared to the control group, respectively. **b** Gene set enrichment analysis (GSEA) of fatty acid biosynthesis gene sets in the expression profiles of MHCC-97H cells transduced with two independent USP22 shRNAs. **c**, **d** qRT-PCR (**c**) and western blot (**d**) analysis of *ACACA, ACLY, FASN* and *SCD*, in MHCC-97H-shUSP22-1/2 cells and MHCC-97L-USP22 cells. The data shown represent the means (±SD) of biological triplicates ($n = 3$). One-way ANOVA test. **e** Correlation analysis between *USP22* and *ACACA, ACLY* based on the TCGA LIHC database. R represents the Pearson correlation coefficient. **f** Relative contents of FA 16:0 and FA 18:0 were analyzed in MHCC-97L and SNU449 cells transduced with USP22 alone or in combination with ACC or ACLY shRNA. Cells were analyzed by LC-MS after 24 h adherence. One-way ANOVA test. The data shown represent the means (±SD) of biological replicates. The experiments were repeated four times ($n = 4$). **g** Relative content of TG was analyzed in the above cell lines from **f**. Cells were analyzed after 24 h adherence. The data shown represent the means (±SD) of biological triplicates. One-way ANOVA test. The data shown represent the means (±SD) of biological replicates. The experiments were repeated five times ($n = 5$). Source data are provided in the Source Data file.

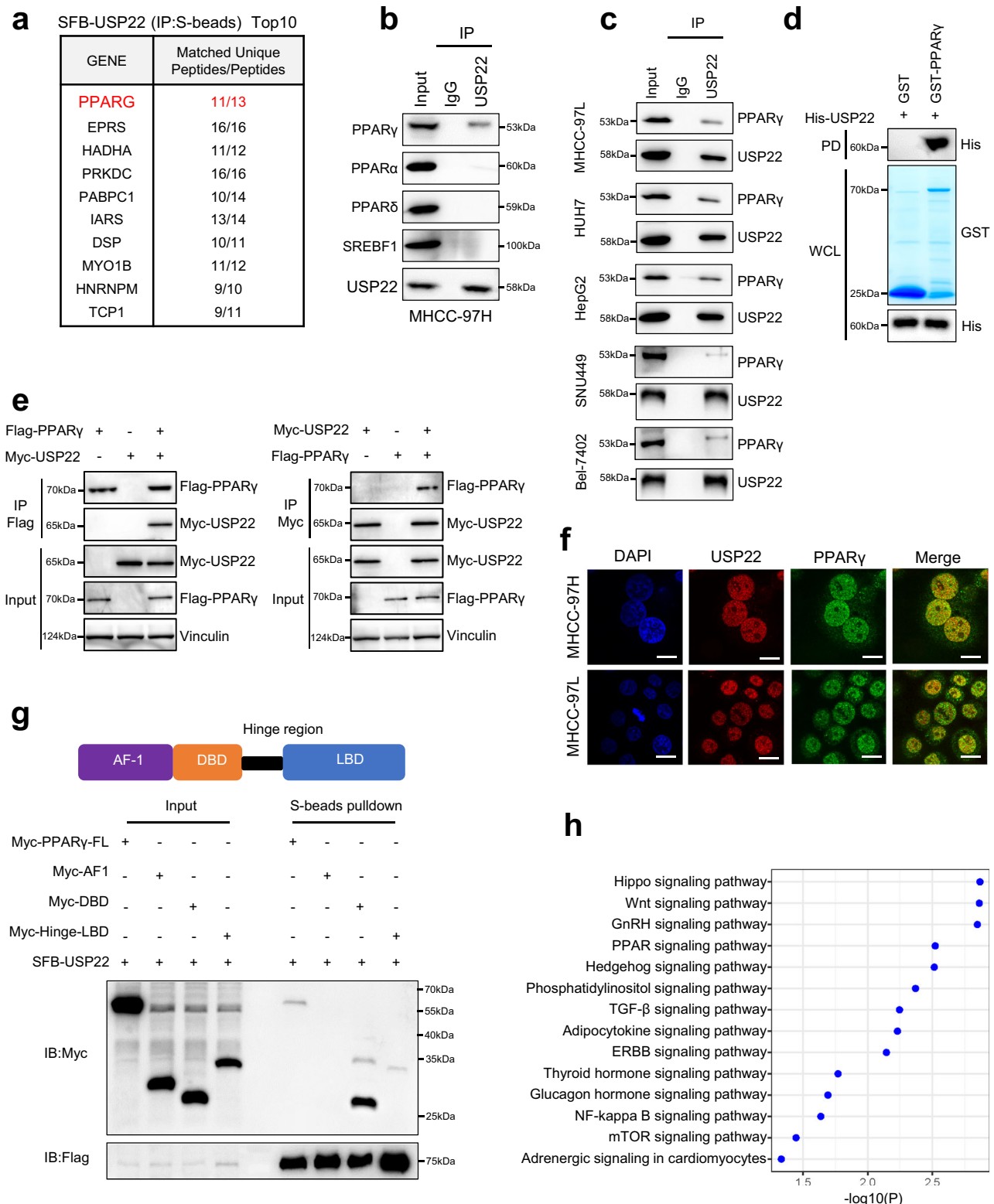

MHCC-97H cells but not in the Flag-PPARγ-5KR overexpression cells (Supplementary Fig. 7e). Same line with this, knockdown of USP22 in Flag-PPARγ overexpressing cells had a more significant inhibitory ability on cell proliferation compared to Flag-PPARγ-5KR transduced cells (Supplementary Fig. 7f). In agreement with these findings, reconstitution of PPARγ in USP22-engineered HCC cells, significantly regulated both mRNA (Fig. 6e) and protein (Fig. 6f) levels of ACLY and ACC. Next, we examined the protein

expression of USP22, PPARγ, ACC and ACLY in HCC cell lines, and found the expressions of these factors were positively correlated in HCC cells (Supplementary Fig. 7g). Moreover, tumors formed by the USP22 knockdown MHCC-97H cells exhibited downregulation of ACC and ACLY compared to tumors formed by control shRNA exprssingMHCC-97H cells (Supplementary Fig. 7h). It has been found that the activation of PPARγ is dependent on AKT[31], however, USP22-dependent upregulation of

**Fig. 4 USP22 specifically interacts with lipid metabolism key transcription factor of PPARγ. a** Tandem affinity purification–mass spectrometry detection of USP22-interacting proteins (obtained from S-beads pulldown) after HEK293T cells were transfected with SFB-USP22 for 24 h. **b** Cell lysates of MHCC-97H cells were immunoprecipitated with IgG or USP22 antibodies, and immunoblot assays were performed using USP22, PPARγ, PPARα, PPARδ and SREBF1 antibodies. **c** Cell lysates of MHCC-97L, HUH7, HepG2, SNU-449, and Bel-7402 cells were immunoprecipitated with IgG or USP22 antibodies, and immunoblot assays were performed using USP22 or PPARγ antibodies. **d** GST pulldown assay with purified His-USP22 and GST-PPARγ. PD Pulldown. **e** HEK293T cells were transfected for 24 h with plasmids encoding either Flag-PPARγ or Myc-USP22 alone or in combination. Cell lysates were immunoprecipitated with Flag and Myc antibodies, and immunoblotting was performed using Myc or Flag antibodies. **f** Triple immunoflorescence (IF) staining for USP22 (red), PPARγ (green), and nuclei (DAPI, blue) was performed in MHCC-97L and MHCC-97H cells. Scale bars, 10 μm. **g** Plasmids containing FL (full length), AF-1, DBD, Hinge-LBD domain of PPARγ were co-expressed with SFB-USP22 in 293 T cells. Lysates were immunoprecipitated with S-beads. **h** GSEA of signaling pathway with USP22-correlated genes based on TCGA LIHC database. All experiments were performed independently at least three times.

PPARγ, ACC and ACLY might not depend on AKT activation (Supplementary Fig. 7i). To further clarify whether USP22 in HCC tissues was correlated with PPARγ, ACC and ACLY, we performed IHC staining using HCC tissue microarrays (TMAs). The results revealed that USP22 protein expression was positively correlated with PPARγ, ACLY and ACC protein expression (Fig. 6g, h). Further analysis also revealed that the increased expression of USP22, PPARγ, ACC and ACLY were associated with steatosis in the HCC TMAs (Fig. 6i). Collectively, PPARγ as a transcription factor upregulates ACC and ACLY expression under USP22 deubiquitination.

**USP22-driven de novo fatty acid synthesis participates in HCC tumorigenesis through ACC and ACLY upregulation by PPARγ.** To determine whether the enhanced de novo synthesis of fatty acids promote cell growth, we modified the expression of PPARγ or ACC in USP22-knockdown or ectopically expressed cells. Firstly, we observed glucose oxidation to fatty acid synthesis by culturing cells with uniformly labeled [U-$^{13}$C] glucose in knockdown of PPARγ or ACC in the USP22-overexpressing MHCC-97L cells, and found that knockdown of PPARγ or ACC significantly decreased fatty acid labeling from glucose tracers (Fig. 7a). Conversely, PPARγ-transduction significantly increased fatty acid labeling from glucose tracers in USP22-knockdown MHCC-97H cells (Fig. 7b). In addition, oil red staining showed a similar TG accumulation trend in the above HCC cells (Fig. 7c–f). These results suggested that USP22 upregulates fatty acid biosynthesis through PPARγ stabilization.

To verify that USP22 regulates tumorigenesis by activating fatty acid synthesis, we conducted xenograft tumor experiments using the above cell lines. Notably, mice implanted with the USP22-overexpressing MHCC-97L cells formed larger tumors compared to those implanted with the control MHCC-97L cells; however, MHCC-97L cells with simultaneous USP22 overexpression and PPARγ, ACC or ACLY knockdown showed lower tumor growth rates (Fig. 7g–i). Furthermore, mice implanted with either the shControl MHCC-97H cells or the MHCC-97H cells with simultaneous USP22 knockdown and PPARγ overexpression showed similar tumor growth rates, whereas mice bearing the USP22-knockdown MHCC-97H cells showed markedly inhibited tumorigenesis (Fig. 7j–l). Indeed, we also observed that the tumors formed with USP22 overexpression upregulated PPARγ, ACC and ACLY expressions compared with control cell-formed tumors; however, this upregulation was inhibited by knockdown of PPARγ in the USP22-overexpressing MHCC-97L cells (Fig. 7m). In addition, PPARγ overexpression restored the downregulation of ACC and ACLY in the USP22-knockdown MHCC-97H cells (Fig. 7n). Collectively, these results reveal that overexpression of USP22 can lead to activation of de novo synthesis of fatty acid signaling in vivo through PPARγ mediated *ACACA* and *ACLY* expression in HCC.

**The USP22–PPARγ/ACC/ACLY axis contributed to HCC prognosis.** To investigate the relevance of our findings to human HCC, we analyzed the expression of USP22, PPARγ, ACC, and ACLY in HCC TMAs (which contains prognosis information) by IHC (Fig. 8a), and found that USP22 expression was positively correlated with PPARγ, ACC or ACLY expression (Fig. 8b). In addition, PPARγ was positively correlated with both ACC and ACLY (Supplementary Fig. 8a). We also found a significant positive correlation between *USP22* and *PPARG, PPARG and ALCY, PPARG and ACACA* transcript levels in the TCGA HCC database (Supplementary Fig. 8b). Next, we evaluated the prognostic value of USP22, PPARγ, ACC and ACLY in this HCC TMA dataset. Of note, patients with high levels of USP22 or PPARγ had much shorter overall survival than patients with low levels of USP22 or PPARγ in this HCC TMA dataset (Fig. 8c). In addition, patients with simultaneously high levels of USP22 and PPARγ had significant poor overall survival (Fig. 8c). However, neither patients with single high level of ACC or ACLY nor patients with simultaneously high levels of USP22 with ACC and ACLY showed a poor prognosis (Supplementary Fig. 8c).

To further reveal the contribution of ACC and ACLY to the prognosis of USP22-positive HCC patients, we examined the prognostic value of *USP22, ACACA* and *ACLY* in the TCGA HCC dataset of 332 patients (patients died within 3 months or followed up for less than 1 month were removed). HCC patients with single high level of *USP22* or *PPARG* or *ACACA* or *ACLY* showed a poor prognosis (Fig. 8d; Supplementary Fig. 8d). Furthermore, patients with simultaneously high levels of *USP22* and *PPARG* have a worse overall survival than those with single high level of above factors (Fig. 8d). In addition, the patients with simultaneously high expression levels of *USP22, ACACA* and *ACLY* had worst overall survival in HCC cohorts (Fig. 8d). On the other hand, when *USP22* was positively high expressed with either *ACLY* or *ACACA* in patients with HCC, they had a much worse overall survival (Supplementary Fig. 8d). Moreover, we analyzed triple correlation of *USP22/PPARG/ACACA* (or *ACLY*) and found that simultaneously high expression levels of *USP22/PPARG/ACACA* (or *ACLY*) had worse overall survival in HCC cohorts (Supplementary Fig. 8e). Collectively, we revealed a previously undescribed pathogenic relationship between USP22 and the de novo fatty acid synthesis factors of PPARγ, ACC and ACLY in HCC patients.

## Discussion
Dysregulation of cellular metabolism is a hallmark of cancer[34]. In addition to elevated glycolysis, de novo fatty acid synthesis is a common feature that needs to meet the biosynthetic requirements of growing tumors[13]. Fatty acid synthesis occurs at a lower rate in nondividing cells, which mainly absorbs lipids from the extracellular circulation. In contrast, DNL, especially de novo fatty acid synthesis, is an important lipid source for cancer cells[11]. Here, we have identified USP22 as a deubiquitinating enzyme that regulates

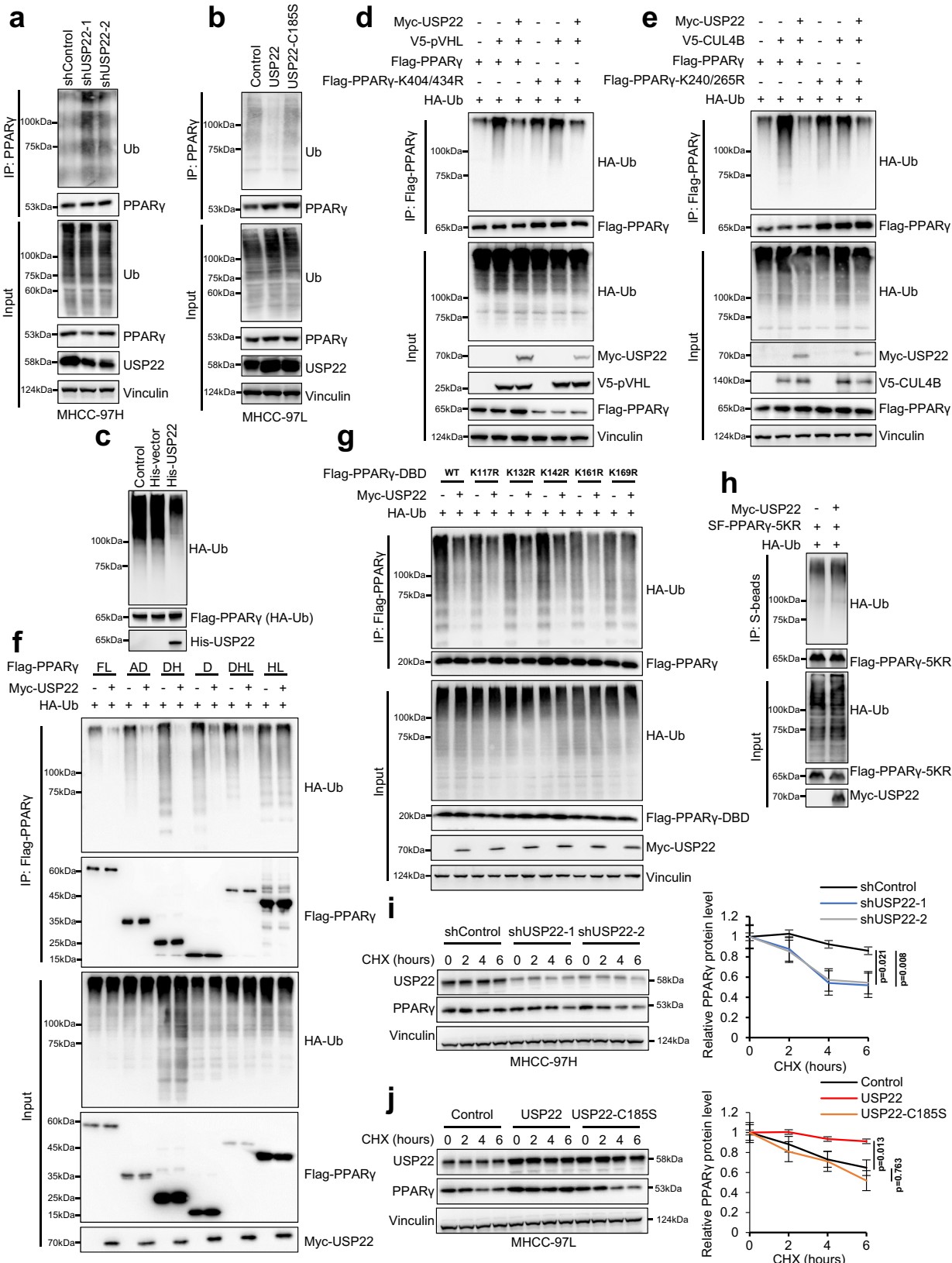

de novo fatty acid synthesis, which directly deubiquitinates and stabilizes PPARγ, and this stabilization in turn activates ACC and ACLY transcription (Fig. 8e).

MAFLD is an increasingly important risk factor for HCC, especially in developed countries[1]. MAFLD is characterized in part by the excessive accumulation of TGs in hepatocytes, which is due to elevated hepatic de novo fatty acid synthesis[5]. The resulting fatty acids are either stored in the form of TGs or used to synthesize sphingomyelin and glycerol phospholipid and eventually used as signal molecules or membrane building blocks.

**Fig. 5 USP22 deubiquitinates and stabilizes PPARγ. a, b** Ubiquitination assay of PPARγ in MHCC-97H-shUSP22-1/2 cells (**a**) or MHCC-97L-USP22, MHCC-97L-USP22 C185S cells (**b**) treated for 6 h with 10 μM MG132. **c** In vitro deubiquitination assay of ubiquitinated PPARγ protein with purified His-USP22. **d** Ubiquitination assay of PPARγ in HEK293T cells cotransfected with HA-Ub, Flag-PPARγ, Flag-PPARγ-K404/434 R, Myc-USP22 or V5-pVHL and treated with 10 μM MG132 for 6 h. **e** Ubiquitination assay of PPARγ in HEK293T cells cotransfected with HA-Ub, Flag-PPARγ, Flag-PPARγ-K240/265 R, Myc-USP22 or V5-CUL4B and treated with 10 μM MG132 for 6 h. **f** Ubiquitination assay of FL (full length), AD (AF-1-DBD), DH (DBD-Hinge), D (DBD), DHL (DBD-Hinge-LBD) and HL (Hinge-LBD) domains of PPARγ in HEK293T cells cotransfected with HA-Ub and Myc-USP22, and treated with 10 μM MG132 for 6 h. **g** Ubiquitination assay of DBD domain of PPARγ in HEK293T cells cotransfected with HA-Ub, Myc-USP22, Flag-DBD, Flag-DBD-K117R, Flag-DBD-K132R, Flag-DBD-K142R, Flag-DBD-K161R and Flag-DBD-K169R and treated with 10 μM MG132 for 6 h. **h** Ubiquitination assay of PPARγ in HEK293T cells cotransfected with HA-Ub, Myc-USP22, and SF-PPARγ-5KR (K169/240/265/404/434 R) and treated with 10 μM MG132 for 6 h. **i, j** Stability analysis of PPARγ protein in MHCC-97H-shUSP22-1/2 cells (**i**), MHCC-97L-USP22, MHCC-97L-USP22 C185S cells (**j**) and treated with 40 μM cycloheximide (CHX) for indicated times. Right panels are quantification of PPARγ protein levels. Data are presented as mean values ± SD. One-way ANOVA test. n = 3 independent experiments. Source data are provided in the Source Data file. All experiments were performed independently at least three times.

In this study, we observed that a variety of lipids in HCC tissues were significantly increased, including fatty acids, phospholipids and sphingomyelin, accounting for 70.83% of all differential metabolites (Fig. 1). These results are similar to our previous study[35] but more remarkable, which further increases our interest in the sample set and promotes us to explore the specific reasons.

Metabolic alterations are caused by abnormal expression or activation of related enzymes[36,37]. Recently, due to the potential roles of the stabilization of oncoproteins, DUBs have been widely studied in cancer progression. Approximately 100 DUBs are encoded in the human genome, and USP is the largest subfamily with nearly 60 members[38]. Several lines of evidences indicate that USPs play critical roles in lipogenesis. USP13 is abnormally highly expressed in ovarian cancer and participates in lipid synthesis and tumorigenesis through deubiquitination of ACLY[39]. USP14 directly interacts and stabilizes FASN, followed by elevated triglyceride accumulation[18]. Additionally, USP30 deubiquitinates and stabilizes ACLY and FASN and plays important roles in lipogenesis and HFD-driven HCC[19]. Interestingly, we also observed the differential expression of USP13 and USP14 between HCC and normal tissues, but the difference in USP22 expression was more significant than USP13 and USP14 (Fig. 1e). USP22 functions as an oncoprotein during tumorigenesis. It is highly expressed in a variety of cancers, including HCC[25]. This oncogenic function of USP22 accompanies with HIF1α in certain conditions[40]. USP22 also has been reported as a prognostic gene in the human pathology atlas[41]. USP22 participates in biological processes such as transcriptional regulation, the cell cycle, and embryonic differentiation by deubiquitinating histones H2A and H2B[21], SIRT1[24], Cyclin B1[23], and Cyclin D1[22]. Previous studies have shown that USP22 promotes fatty acid oxidation by stabilizing SIRT1 in the liver[40]. However, we found that USP22 promotes the de novo synthesis of fatty acids in HCC rather than fatty acid oxidation. Moreover, the aberrant lipid accumulation caused by USP22 depended on the increased expression of ACC and ACLY. ACC, as a central enzyme controlling DNL, augments HCC development, and the ACC small-molecule allosteric inhibitor ND-654 inhibits HCC[42]. The upregulated expression of ACLY increases fatty acid synthesis and promotes tumorigenesis[43]. In this study, we observed that individual knockdown of ACC or ACLY obviously inhibited the increase in tumorigenesis and fatty acid synthesis augmented by USP22 overexpression.

PPARγ is a key transcription factor that regulates lipid synthesis by upregulating the transcription of lipid synthesis enzymes, including ACLY, ACC and FASN. It is highly expressed in adipocytes and is involved in lipid uptake, synthesis, and storage[44]. Here, we found that PPARγ contributes to USP22-mediated ACC and ACLY upregulation in the nucleus (Fig. 6), and demonstrated a previous undescribed PPARγ interacting

PPRE motif from *ACACA* promoter. Notably, upregulated PPARγ promotes lipid synthesis and tumorigenesis accompanied with activation of Akt2 in HCC[31]. Our study found that USP22 might act upstream of p-AKT(S473), however with or without serum incubation USP22 still upregulates PPARγ expression. Therefore USP22 stabilizes PPARγ may independently from AKT activation, and this implies that USP22 involves multiple regulation pathways[45]. Importantly, previous study has clarified that AKT pathway is significantly activated in HCC patients[46]. Taken together USP22-driven fatty acid synthesis may associate with the AKT pathway activation in HCC. In addition, PPARγ is degraded by the UPS pathway in mammalian cells[32,33,47,48]. In this study, we demonstrate that USP22 stabilizes PPARγ through K48-linked deubiquitination, and significantly deubiquitinates the E3 ligases pVHL and CRL4B[AhR] ubiquitination sites (K404/434 and K240/265, respectively). We also identified Lys-169 site at the DBD domain of PPARγ is important for USP22 deubiquitination of PPARγ. However, since the E3 ligases pVHL and CRL4B[AhR] were not involved in this site ubiquitination, the new E3 ligase for this regulation mechanism should be identified in the future. PPARγ knockdown abolished the upregulation of ACC and ACLY expression and dramatically decreased tumorigenesis and fatty acid synthesis caused by USP22 overexpression in HCC cells and xenograft tissues. In addition, patients with HCC whether USP22 is simultaneously highly expressed with PPARγ or *ACACA* or *ACLY*, have poor prognosis and overall survival. In summary, we identified a previously undescribed signaling pathway of the USP22-PPARγ-ACLY/ACC axis that plays an important role in lipogenesis and HCC tumorigenesis and provides an option for cancer therapy targeting fatty acid synthesis.

## Methods

**Reagents**. Please see Supplementary Table 2.

**Clinical specimens**. Ten pairs of HCC (6 female and 4 male patients, the age range is between 48 and 60) samples were obtained from the first affiliated hospital of Dalian Medical University (Dalian, China). All patients were diagnosed with HCC by postoperative pathology and were free of other cancers and chronic diseases and all samples were collected with the informed consent of the patients and the experiments were approved by Research ethics committee at the first affiliated hospital of Dalian Medical University. Human HCC Tissue Microarrays were obtained from Shanghai Outdo Biotech Company or Shanxi ChaoYing Biotechnology Company.

**Cell culture experiments**. Human HCC cell lines MHCC-97H, HUH7, Bel-7402, Hep3B, HepG2 and SMMC-7721 were purchased from the cell bank of the Committee on Type Culture Collection of the Chinese Academy of Sciences (CTCC, Shanghai, China). THLE-2 and SNU449 were obtained from ATCC. MHCC-97H, SNU449, and MHCC-97L cell lines were maintained in RPMI 1640 medium (GIBCO, USA). HEK293T, HUH7, Bel-7402, Hep3B, HepG2 and SMMC-7721cells were cultured in DMEM (GIBCO, USA). Cell lines were maintained in culture supplemented with 10% FBS (GIBCO, USA) and 1% penicillin/streptomycin (Thermo) at 37°C with 5% $CO_2$ in a humidified incubator (Thermo).

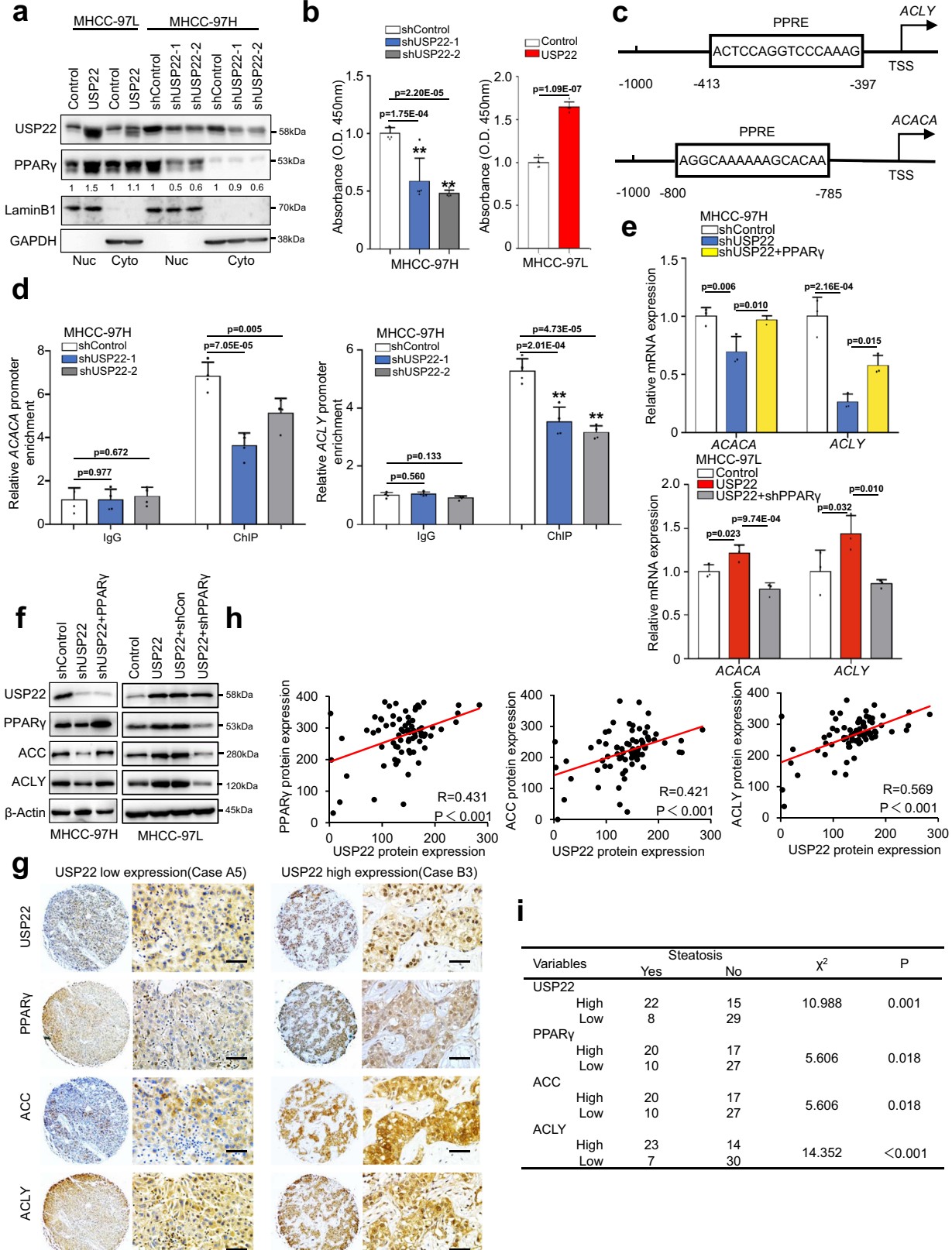

**Plasmids**. Lentiviral shRNAs were cloned in pLKO.1 within the AgeI/EcoRI sites at the 3′ end of the human U6 promoter. The targeted sequences were shown in Supplementary Table 2.

All the expression vectors used in this study (including SFB-PPARγ, Myc-USP22, pLoc-USP22 and Lenti-PPARγ) were constructed using Gateway Technology (Invitrogen). Briefly, cDNAs with attB homologous sequence were generated by PCR and then subcloned into pDONR221 vector as the entry clones.

Subsequently, the entry clones were recombined into gateway destination vectors with various tags (SFB and Myc) or lentiviral vectors (pLoc and Lenti). For Myc-USP22-C185S vector, mutation in the USP22 cDNA was generated by overlap extension PCR and then cloned into Myc vector as above.

**Generation of cell lines expressing shRNAs, pLoc-USP22 or Lenti-PPARγ**. The plasmid was co-transfected with psPAX2 and VSVG using Lipofectamine 2000 in

**Fig. 6 USP22 increases ACC and ACLY expression by stabilizing PPARγ. a** Western blot analysis of USP22 and PPARγ in cytoplasmic and nucleus fractions of MHCC-97H-shUSP22-1/2 cells and MHCC-97L-USP22 cells. LaminB1 and GAPDH were used as nucleus and cytoplasmic markers, respectively. **b** DNA binding activity of PPARγ in MHCC-97H-shUSP22-1/2 cells and MHCC-97L-USP22 cells. The analysis was performed by PPAR gamma Transcription Factor Assay Kit (ab133101, Abcam, USA). The data shown represent the means (±SD) of biological replicates. One-way ANOVA test. The experiments were repeated five time ($n = 5$). **c** Illustration of PPRE site in *ACLY* promoter and the predicted PPRE site in *ACACA* promoter. The PPRE motif from *ACACA* promoter was predicted by web site: https://epd.epfl.ch/index.php. **d** Chromatin immunoprecipitation (ChIP) analysis of PPARγ binding to the *ACLY* and *ACACA* promoters in MHCC-97H-shUSP22-1/2 cells. qPCR was performed with primers specific to the PPARγ-binding motifs. Data were normalized to the input. The data shown represent the means (±SD) of biological triplicates ($n = 3$). One-way ANOVA test. **e, f** qRT-PCR (**e**) and western blot analysis (**f**) of ACC and ACLY expression in MHCC-97H cells transduced with USP22 shRNA or in combination with PPARγ, and in MHCC-97L cells transduced with USP22 or in combination with PPARγ shRNA. The data shown represent the means (±SD) of biological triplicates ($n = 3$). One-way ANOVA test. **g** Representative IHC staining of USP22, PPARγ, ACC, and ACLY in HCC tissue microarrays (LV1021, no prognosis information, Shanxi ChaoYing Biotechnology Company). Scale bars, 50 μm. **h** Correlation analysis between USP22 and PPARγ, ACC, ACLY protein expression based on H-Score in HCC tissue microarrays (LV1021). R represents Pearson correlation coefficient. **i** The tissue microarray (LV1021) was stained with HE stain, and the steatosis was interpreted by the pathologist. Two-sided Chi-square test was performed to analyze the correlation between USP22, PPARγ, ACC, ACLY protein expression and steatosis. Source data are provided in the Source Data file.

HEK-293T cells, as directed by the manufacturer. Forty-eight hours after transfection, conditioned media containing recombinant lentiviruses was collected and filtered through non-pyrogenic filters with a pore size of 0.45 μm (Merck Millipore, Billerica, MA, USA). Target cells were treated with these supernatants and 8 μg/ml Polybrene (Sigma-Aldrich, St. Louis, MO, USA) immediately and then cultured for another 12 h. Following infection, cells were grown in media as usual. Puromycin (2 μg/ml, InvivoGen) or Blasticidin (8 μg/ml, InvivoGen) was added 48 h after infection.

**Cloning formation assay.** For the clone assay, cells were seeded at a density of 500 cells/well in 12-well plates in complete RPMI 1640 medium. The medium was changed every three days. After 14 days, cells were fixed and stained with crystal violet.

**Cell proliferation assay.** One thousand cells were plated in 12-well plates. Beginning on day 2 to day 10, cells were fixed with 10% methanol and stained with 0.1% crystal violet (dissolved in 10% methanol) every 2 days. After staining, wells were washed three times with PBS and destained with 10% acetic acid, and the absorbance of the crystal violet solution was measured at 590 nm.

**Immunohistochemistry.** The samples were fixed with 4% PFA, and embedded with paraffin. Standard IHC staining procedures were performed according to the instructions of IHC Kit. USP22 (1:100), ACC (1:750), ACLY (1:750) and PPARγ (1:50) antibodies were applied in this study. EDTA and Citrate solution were used for antigen retrieval depend on antibody instruction. H-score was used to assess the staining intensity.

**Xenograft Tumor Model.** Pathogen-free male athymic nude mice (4–5 weeks old, 18–22 g) were purchased from the Beijing Vital River Laboratory Animal Technology Co., Ltd (Beijing, China). All animal procedures were conducted in accordance with the guidelines of the Institutional Committee for the Ethics of Animal Care and Treatment in Biomedical Research of Dalian Medical University. All the mice were housed in specific pathogen-free (SPF) environments on a 12 h light/dark cycle at temperature 20–25 °C and humidity 50–60% at the Institute of Genome Engineered Animal Models for Human Disease of Dalian Medical University. During the tumor formation assay, Bel-7402 ($5 \times 10^6$), MHCC-97L ($5 \times 10^6$) or MHCC-97H ($1 \times 10^6$) cells were injected into the flank of the mice. The tumor volumes were measured using a caliper every 3 days. The mice were sacrificed after 3 weeks, and tumor volumes were then measured.

**Nucleus–cytoplasmic fractionation assay.** According to the manufacturer's recommendations, adherent cells were scraped, and the cell pellet was obtained by centrifugation. Next, resuspend cell pellet in 100 μL of 1× Pre-Extraction Buffer per $10^6$ cells, and transfer to a micro-centrifuge vial. Incubate on ice for 10 min. Vortex vigorously for 10 s and centrifuge the preparation for 1 min at $10,000 \times g$. Carefully remove the cytoplasmic extract from the nuclear pellet. Add Extraction Buffer containing DTT and PIC to nuclear pellet. Incubate the extract on ice for 15 min with vortex (5 s) every 3 min. The extract can be further sonicated for 3 times per 10 s to increase nuclear protein extraction for tissue extract. Centrifuge the suspension for 10 min at $14,000 \times g$ at 4 °C and transfer the supernatant into a new microcentrifuge vial. Measure the protein concentration of the nuclear extract.

**Immunoprecipitation and S-Protein pull down assay.** Immunoprecipitation and SFB pull-down experiment was performed as described previously[49]. Briefly, cells were lysed in E1A lysis buffer (250 mM NaCl, 50 mM HEPES [pH 7.5], 0.1% NP-40, 5 mM EDTA, protease inhibitor cocktail [Sigma]). The antibodies to USP22

and PPARγ were used for immunoprecipitation. HEK293T cells were transfected with SFB-tagged protein and lysed in NETN buffer (200 mM Tris-HCl [pH 8.0], 100 mM NaCl, 0.05% NP-40, 1 mM EDTA, protease inhibitor cocktail [Sigma]) for 20 min at 4 °C. Crude lysates were subjected to centrifugation at $14,000 \times g$ for 15 min at 4 °C. Supernatants were incubated with S-Protein Agarose for 4 h (Millipore, USA). The agaroses were washed three times with NETN buffer. Proteins were eluted by boiling in 1× SDS loading buffer and subjected to SDS-PAGE for immunoblotting. Unprocessed scans of immunoblots are provided in the Source Data file.

**GST pull-down assay.** GST pull-down assay was used to detect the direct interaction between PPARγ and USP22. Briefly, GST-tagged PPARγ (GST-PPARγ) and 6×His-tagged USP22 (His-USP22) proteins were expressed in BL21 (DE3) Escherichia coli via transforming pGEX-4T-1-GST-PPARγ and pET24a-6×His-USP22 plasmids, respectively. Then, the *E. coli* were collected, sonicated, and purified with cOmplete His-Tag Purification Resin (Roche) to obtain purified His-USP22 protein. GST-PPARγ protein was expressed and immobilized with Beyo-Gold™ GST-tag Purification Resin (Beyotime) following the manufacturer's instructions. The beads-PPARγ complexes were washed with GST pull-down binding buffer (50 mM Tris-HCl, 200 mM NaCl, 1 mM EDTA, 1% NP-40, 1 mM DTT, 10 mM MyCl₂, pH 8.0) and incubated with purified His-USP22 at 4 °C for 4 h on a rotating windmill. Finally, the beads were washed and analyzed by Western Blot.

**In vivo and in vitro deubiquitination assay.** In vivo deubiquitination assay: Cells were treated with 10 μM MG132 for 6 h before harvested. Then lysis was performed with RIPA buffer containing 1% SDS followed by mild sonication and a final boil at 95 °C for 10 min. SDS concentration of the cell lysates was diluted to 0.2% using SDS-free lysis solution. Immunoprecipitation was carried out with PPARγ antibody at 4 °C and ubiquitination level was further analyzed through Western blot.

In vitro deubiquitination assay was performed as described before[25]. First, Flag-PPARγ and HA-Ub expression vectors were co-transfected in HEK293T cells for 24 h. Then, cells were treated with 10 μM MG132 for 6 h, followed by immunoprecipitation with Flag affinity beads to extract the ubiquitinated PPARγ protein. Afterwards, the immunoprecipitates were washed three times with the ubiquitination wash buffer and followed by elution in BC100 buffer using 3×Flag peptide (F4799, Sigma-Aldrich). His-USP22 and vehicle His control proteins were expressed and purified, respectively. The ubiquitinated PPARγ protein was then incubated with 200 ng of recombinant USP22 protein or His tag in deubiquitination buffer (50 mmol/L Tris-HCl pH 8.0, 50 mmol/L NaCl, 1 mmol/L EDTA, 10 mmol/L DTT and 5% glycerol) for 2 h at 37 °C, respectively, and PPARγ ubiquitination level was analyzed by Western blot.

**PPARγ DNA binding activity assay.** PPARγ Transcription Factor Assay Kit (Abcam, ab133101) was used to detect intracellular PPARγ-DNA binding activity. Firstly, nuclear extracts of the cells were prepared using the Nuclear Extraction Kit (Abcam, ab113474), and the resultant nuclear proteins were added to wells pre-coated with a specific double-stranded DNA sequence containing the peroxisome proliferator response element. Following that, specific primary antibodies and horseradish peroxidase-conjugated secondary antibodies were added according to the instructions. Finally, after adding the developing and halting solutions, the absorbance at 450 nm was determined using a microplate reader.

**Luciferase assays.** DNA transfection and luciferase assays were carried out following the Dual luciferase reporter assay system (Promega). HEK293T cells were cultured in 24-well plate one day before transfection with $1 \times 10^5$ cells per well. The Renilla plasmid (pRL-TK) was transfected with indicated vectors and firefly

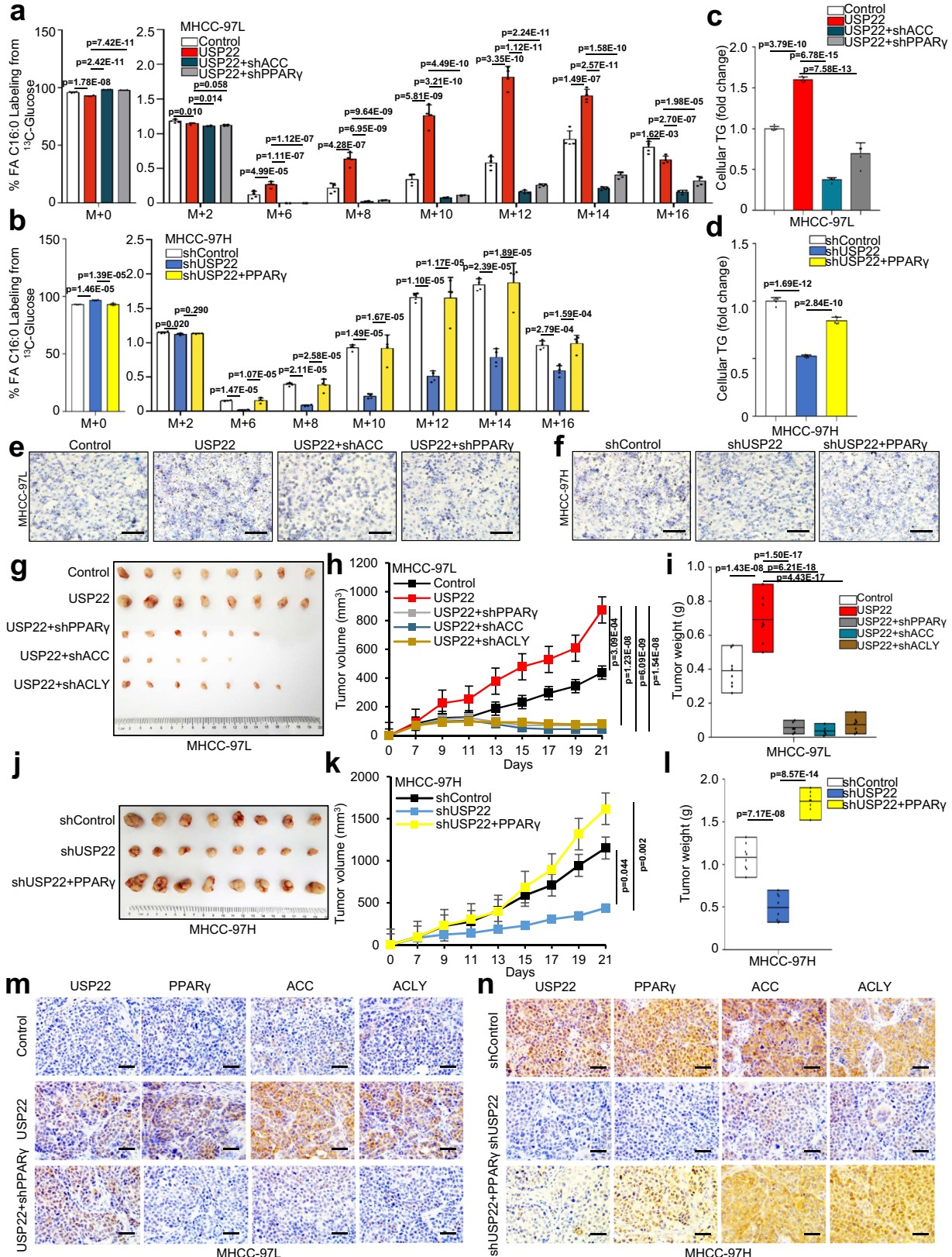

luciferase activity obtained from each sample was normalized to the Renilla luciferase activity from the same sample. The PPRE-Luc vector was transfected with SFB-PPARγ or SFB-PPARγ and Myc-USP22. Cells were harvested after 24 h and luciferase activity was measured by BioTek Cytation 5. For determination of PPARγ binding region on *ACACA* and *ACLY* promoters, the genomic fragment −488 nt to −331 nt from *ACLY* promoter and fragment −992 nt to −757 nt from *ACACA* promoter containing predicted PPRE were amplified and cloned into

pGL4.16 vector (Promega). A sequence containing nucleotides −662 nt to −482 nt from ACC promoter was also cloned into pGL4.16 as negative control. The constructed pGL vector was transfected into HEK293T cells with/without PPARγ expression plasmid and luciferase activity was measured and analyzed as above.

**Triglyceride detection assay**. TG was measured with Triglyceride Quantification Assay Kit (Solarbio, BC0625). $5 \times 10^6$ cells were collected and then added 1 ml

**Fig. 7 USP22-driven de novo synthesis participates in HCC tumorigenesis through ACC and ACLY upregulation by PPARγ. a, b** The percentages of various isotopomers of FA 16:0 after trace to [U-$^{13}$C] glucose in MHCC-97L-related stable cells (Control, USP22, USP22 + shACC and USP22 + shPPARγ) (**a**) and MHCC-97H-related stable cells (shControl, shUSP22-1 and shUSP22-1+PPARγ) (**b**). Medium was changed to RPMI 1640 containing [U-$^{13}$C] glucose (2 g/L) when cell confluence reached 80%, and 24 h later cells were washed with cold PBS and snap-frozen in liquid nitrogen and subjected to LC-MS analysis. $n = 4$ biologically independent experiments. **c, d** The relative content of TG was analyzed in above cell lines from **a** (**c**) and **b** (**d**). The data shown represent the means (±SD) of biological triplicates. Cells were analyzed after 24 h adherence. $n = 4$ (**c**) or $n = 5$ (**d**) biologically independent experiments. **e, f** Oil red staining assay in above cell lines from **a** (**e**) and **b** (**f**). Cells were analyzed after 24 h adherence. Scale bars, 100 μm. **g–i** MHCC-97L-related stable cells (Control, USP22, USP22 + shACC, USP22 + shACLY and USP22 + shPPARγ) were injected into the right flanks of null mice. Tumor volumes were measured every 3 days. Tumor images (**g**), growth curves (**h**) and weight (**i**) were obtained at day 21 after dissection. In **h** and **i**, $n = 8$ biologically independent tumor samples for Control and USP22 groups, $n = 6$ for USP22 + shPPARγ group, $n = 5$ for USP22 + shACLY group and $n = 7$ for USP22 + shACLY group. **j–l** MHCC-97H-related stable cells (shControl, shUSP22-1 and shUSP22-1-PPARγ) were injected into the right flanks of null mice. Tumor volumes were measured every 3 days. Tumor images (**j**), growth curves (**k**) and weight (**l**) were obtained at day 21 after dissection. $n = 8$ biologically independent tumor samples in **k** and **l**. **m** Representative IHC staining of USP22, PPARγ, ACC, and ACLY in xenograft tissues described in (**g**). Scale bars, 50 μm. **n** Representative IHC staining of USP22, PPARγ, ACC, and ACLY in xenograft tissues described in (**j**). Scale bars, 50 μm. Data in **a–d**, **h**, **k** are presented as mean values ± SD and data in **i** and **l** are presented as mean values with minima and maxima. One-way ANOVA test. Source data are provided in the Source Data file.

extraction reagent. Ultrasonic for 1 min and then the samples were centrifuged at 8000 g at 4 °C for 10 min. Supernatant was taken for testing according to the manufacturer's instructions.

**Oil-Red-O staining assay.** Oil-Red-O staining was performed with Oil Red O Kit (G1262, Solarbio). Cells were washed with PBS for twice, and fixed with the fixative buffer for 30 min. Wash the cells with distilled water twice and then incubate in 60% isopropanol for 5 min. The newly prepared oil red O staining solution was added and soaked for 20 min. Mayer hematoxylin staining solution was added for 2 min. Discard the dye and wash it for 3 times. Oil-Red-O staining pictures were taken using an Olympus IX71 microscope.

**Immunofluorescence.** After seeding cells on the 8-chamber slide, and cells were fixed with 4% paraformaldehyde and permeabilized with 0.1% Triton X-100 for 10 min. After blocked with 10% goat serum, the permeabilized cells were incubated with primary antibodies overnight at 4 °C. Then the cells were washed in PBS, stained with secondary antibodies (goat anti-mouse Alexa 488 and goat anti-rabbit Alexa 555) for 1 h at room temperature, followed by counterstaining with DAPI. Images were taken with an scanning confocal microscope.

**RNA isolation and quantitative real-time PCR.** Trizol reagent was used to isolate RNA from tumor cells. the Revert Aid First Strand cDNA Synthesis Kit was used to do reverse transcriptional PCR. StepOnePlus and the DNA double-strand-specific reagent SYBR-Green I were used to perform quantitative real-time PCR. The Cq technique is used to calculate fold changes. Results were normalized to *GAPDH* levels. The primer sequences were as follows:

*USP22*
F 5′-AGCAGCGGATTCACCATCTC-3′
R 5′-TGATGTATGCGATCACCAGTGT-3′
*PPARG*
F 5′-GATGCCAGCGACTTTGACTC-3′
R 5′-ACCCACGTCATCTTCAGGGA-3′
*ACACA*
F 5′-ATGTCTGGCTTGCACCTAGTA-3′
R 5′-CCCCAAAGCGAGTAACAAATTCT-3′
*ACLY*
F 5′-TCGGCCAAGGCAATTTCAGAG-3′
R 5′-CGAGCATACTTGAACCGATTCT-3′
*FASN*
F 5′-AAGGACCTGTCTAGGTTTGATGC-3′
R 5′-TGGCTTCATAGGTGACTTCCA-3′
*SCD*
F 5′-TCTAGCTCCTATACCACCACCA-3′
R 5′-TCGTCTCCAACTTATCTCCTCC-3′
*ACOX1*
F 5′-TGCTCAGAAAGAGAAATGGC-3′
R 5′-TGGGTTTCAGGGTCATACG-3′
*CPT1A*
F 5′-CCTCCGTAGCTGACTCGGTA-3′
R 5′-CGGAGTGACCGTGAACTGA-3′
*CPT2*
F 5′-GCCTAGATGACTTCCCCATTAA-3′
R 5′-AAAGGATTTATCAAACCAGCGG-3′
*ECHS1*
F 5′-GCCTCGGGTGCTAACTTTGA-3′
R 5′-GCCATCGCAAAGTGCATTGA-3′

*ACADL*
F 5′-TCTTTTCCTCGGAGCATGACA-3′
R 5′-GACCTCTCTACTCACTTCTCCAG-3′
*GAPDH*
F 5′-CATCTTCTTTTGCGTCGCCA-3′
R 5′-TTAAAAGCAGCCCTGGTGACC-3′

**RNA sequencing.** RNA was extracted from MHCC-97H cells using the RNeasy Mini kit (Qiagen) according to the manufacturer's protocol. RNA samples were quantified using Qubit 2.0 Fluorometer (Life Technologies, Carlsbad, CA, USA) and RNA integrity was checked with 4200 TapeStation (Agilent Technologies, Palo Alto, CA, USA). RNA sequencing library preparation used the NEBNext Ultra RNA Library Prep Kit for Illumina followed by manufacturer's instructions (NEB, Ipswich, MA, USA). Sequencing was done on the Illumina HiSeq instrument using a 2 × 150 Paired End (PE) configuration with 30–40 million reads per sample by GENEWIZ, LLC. (South Plainfield, NJ, USA). Sequencing libraries were constructed from total RNA using SMART-RNAseq Library Prep Kit (Hangzhou KaiTai, AT4201). In briefly, the mRNA were isolated from total RNA with Sera-Mag Magnetic Olido(dT) particles, and then chemically fragmented. The fragmented RNA was reverse-transcribed into cDNA using random primer containing a tagging sequence at their 3′ ends. And the cDNA libraries were subsequently amplified using the KAPA high-fidelity DNA polymer. Quality of the libraries was validated by the 2100 Bioanalyzer (Agilent Technologies). Subsequently, High-throughput sequencing was performed using a NovaSeq 6000 (Illumina). After sequences were mapped using hisat2 (version 4.8.2) against the Mus_musculus.GRCm38.dna.primary_assembly.fa, the reads for each library were converted to FPKM (fragments per kilobase of exon per million fragments mapped) by running Cuffdiff 2.1.137 to determine gene expression. Biological pathway analysis was performed using clusterProfiler. RNA sequencing and library construction were performed by technical staff at Hangzhou KaiTai Bio-lab.

**Chromatin immunoprecipitation (ChIP) assays.** ChIP assays were carried out using the Pierce Agarose ChIP Kit (Cat. No. 26156, Thermo Fisher Scientific) according to the manufacturer's instructions. Briefly, cells were fixed with 1% formaldehyde in culture medium at room temperature for 10 min. Then, the cross-linked cells were collected, lysed and digested with MNase. The sheared chromatin was subjected to immunoprecipitation with anti-PPARγ antibody or normal rabbit IgG control overnight at 4 °C with constant rotation. The isolated complexes were collected with protein A/G agarose beads and eluted by incubating at 65 °C in a buffer with high salt concentration. The fold-enrichment of PPARγ binding on promoter region was identified by quantitative real-time PCR. The primer sequences were as follows:

*ACLY*
F 5′-CAAGGGAAGGAGCAAGGGTA-3′
R 5′-CACCGCCTCTTGGGAGC-3′
*ACACA*
F 5′-ATATTAGCTGGGCGTGGTGG-3′
R 5′-CCACGGAAGTTGGTGTCAGA-3′

**Tandem affinity purification of SFB-tagged protein complex.** For affinity purification, SFB-USP22 and control vector transduced HEK293T cells were subjected to lysis in NETN buffer (100 mM NaCl, 20 mM Tris-Cl, 0.5 mM EDTA, 0.5% Nonidet P-40) with protease and phosphatase inhibitors at 4 °C for 20 min. Crude lysates were subjected to centrifugation (14,000 × g) at 4 °C for 15 min. Supernatants were incubated with streptavidin-conjugated beads (Amersham) for 4 h at 4 °C. The beads were washed three times with NETN buffer, and bound proteins

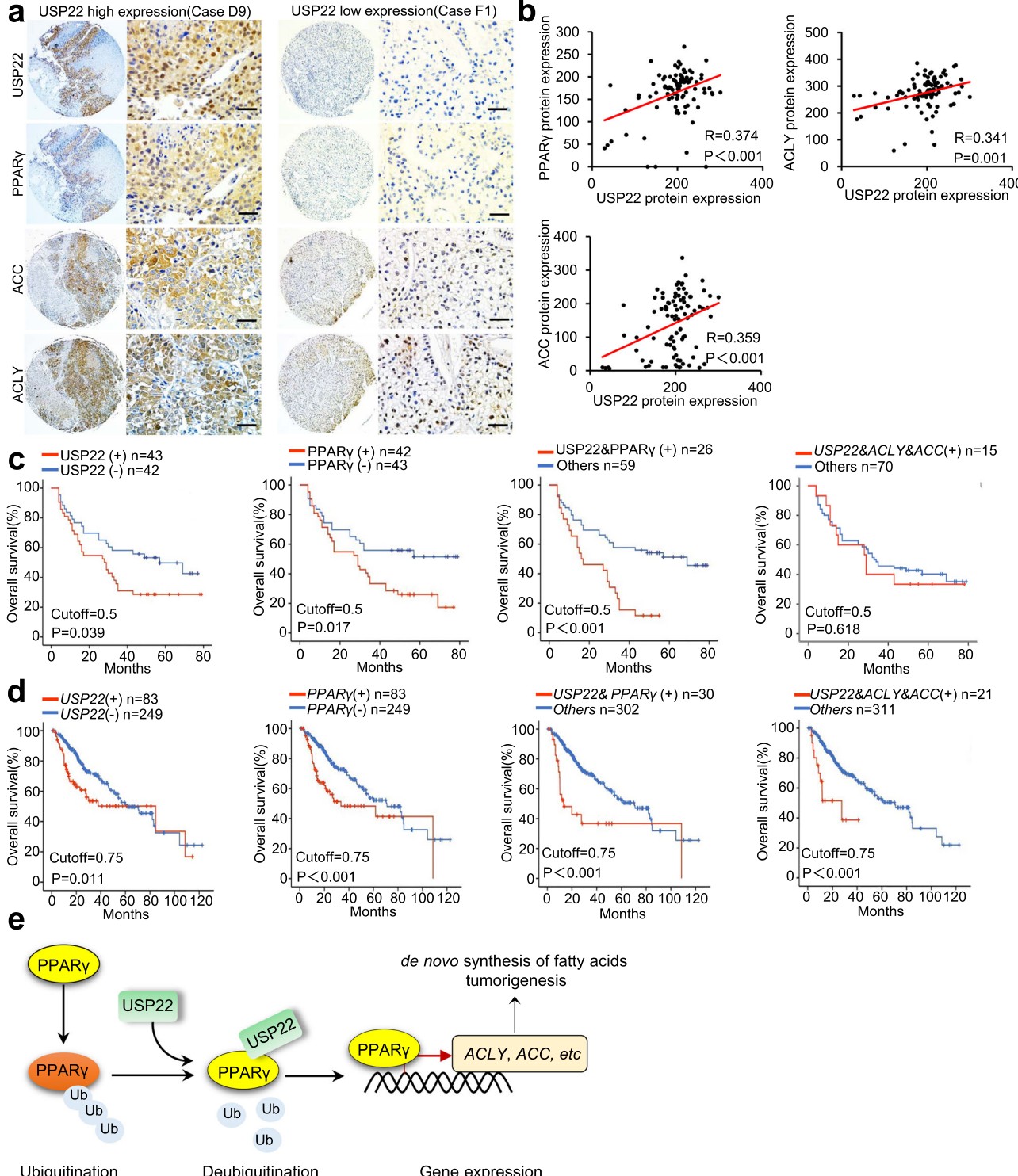

**Fig. 8 The USP22–PPARγ/ACC/ACLY axis contributed to HCC prognosis. a** Representative IHC staining of USP22, PPARγ, ACC, and ACLY in HCC TMAs (HLivH180Su11, contains prognosis information, Shanghai Outdo Biotech Company). Scale bars, 50 μm. **b** Correlation analysis between USP22 and PPARγ, ACC, ACLY protein expression based on H-Score in HCC TMAs (HLivH180Su11). R represents the Pearson correlation coefficient. **c** Kaplan–Meier curves of the survival analysis of USP22-positive, PPARγ-positive, USP22& PPARγ copositive, and USP22&ACC&ACLY copositive patients based on HCC TMAs prognosis data (HLivH180Su11). **d** Kaplan–Meier curves of the survival analysis of *USP22*-positive, *PPARG*-positive, *USP22&PPARG* copositive, and *USP22&ACACA&ACLY* copositive patients based on the prognosis data of TCGA LIHC database. **e** Diagram of the proposed mechanism. Source data are provided in the Source Data file.

were eluted with NETN buffer containing biotin (2 mg ml$^{-1}$; Sigma) overnight at 4 °C. The elutes were incubated with S protein beads (Novagen) for 4 h. The beads were washed three times with NETN buffer and subjected to sodium dodecyl sulphate polyacrylamide gel electrophoresis (SDS–PAGE). Protein bands were excised and subjected to mass spectrometry analysis. A data-dependent technique in which one MS scan was followed by twenty MS/MS scans with 15.0 s dynamic exclusion was used. Adjustment of the automatic gain control (AGC) was set to 5E4. The resulting MS/MS data were processed using Proteome Discoverer 1.3.

**Tracing with [U-$^{13}$C] glucose**. Medium was changed to RPMI 1640 containing [U-$^{13}$C] glucose (2 g/L) when the cell density was about 80%, and 24 h later cell culture plates were washed with PBS and snap-frozen in liquid nitrogen and stored at −80 °C.

**Sample preparation for metabolomics and lipidomics analysis**. Around 20 mg of tissues were mixed with 400 μL of 75% methanol aqueous solution for metabolites extraction. The supernatant was taken and lyophilized. The lyophilized powder was resuspended in 50 μL of 80% methanol aqueous solution. After centrifugation, the supernatant can be directed for UPLC (Waters, USA) -Q Exactive HF MS (Thermo Fisher Scientific, USA) analysis.

Metabolite extraction from cells was performed as follows: briefly, cells collected in a 10-cm dish were rinsed with PBS and instantly frozen in liquid nitrogen. Cells were then lysed with 1 mL of 80% methanol containing internal standards and then scraped off from the dish. The supernatant was collected after centrifugation and freeze-dried for metabolomics analysis using LC-MS.

Lipid extraction from cells was performed as follows: in brief, cells collected in a 10 cm dish were rinsed with PBS and instantly frozen in liquid nitrogen. Cells were then lysed with 1 mL of methanol containing internal standards, and then mixed with 1 mL chloroform and vortexed for 20 s. Subsequently, 400 μL water were added and again for 20 s vortex. The hydrophobic layer was collected and freeze-dried. The lyophilized powder was redissolved in 30 μl organic solvent (Chloroform/Methanol = 2/1) by 30 s vortex, then added 60 μl organic solvent (Acetonitrile/Isopropanpl/H2O = 65/30/5). After centrifugation, the supernatant can be directed for UPLC (Waters, USA) -Q Exactive HF MS (Thermo Fisher Scientific, USA) analysis.

**UPLC-MS based information acquisition**. The acquisition condition for metabolomics analysis was referred from previous study[50]. Briefly, Waters BEH C8 column (100 mm × 2.1 mm, 1.7 μm) and Waters HSS T3 (100 mm × 2.1 mm, 1.8 μm) (Waters, Milford, MA) column were used for metabolite separation in ESI+ mode and ESI− mode, respectively. Acetonitrile/water with 0.1% formic acid as additive was used in ESI+ mode, 95% methanol/water with 6.5 mM NH$_4$HCO$_3$ was utilized in ESI+ mode. The mass spectrometry conditions were also set according to our previous study mentioned above.

The acquisition condition for lipidomics analysis was referred from previous study[51]. Briefly, Waters BEH C8 column (100 mm × 2.1 mm, 1.7 μm) was used for lipid separation and the column temperature was at 55 °C. Acetonitrile/H$_2$O (60:40, v/v) and isopropanpl/acetonitrile (90:10, v/v) both containing 10 mM ammonium acetate was used as mobile phase. The MS capillary temperature was 320 °C with the auxiliary air heating temperature set at 350 °C. Full scan resolution was set as 120 K and the negative mode was used, m/z scan range was 70–1100 Dalton and the spray voltage was 3 kV.

The differential metabolites and lipids were identified based on their retention time, accurate mass, and spectrometric fragments as well as available standard compounds. In addition, for the data analysis of non-targeted metabolomics, we normalized the data by total peak area. For targeted FA detection, we normalized the data using protein dry weight and internal standard. Thermo Scientific Xcalibur (Ver. 4.2.47, Thermo Fisher Scientific, USA) was used for raw data collection of metabolic and lipid profiling. Thermo Trace Finder EFS (Ver. 3.2.512.0, Thermo Fisher Scientific, USA) was used for data processing, integrating peak area and deriving Excel table.

**Pathway enrichment analysis**. Spearman correlation coefficients were calculated between USP22 and all the other genes based on the RNA-seq expression matrix of the TCGA-LIHC tumor tissues or the RNA-seq data of MHCC-97H cells. Then, the genes were ranked based on the correlation coefficients and utilized as the input for the gene set enrichment analysis (GSEA) based pathway enrichment analysis. Significantly altered genes (Log2FC > 1, Q < 0.05) in the transcriptome data of MHCC-97H-shUSP22 cells were subjected to GO-BP (biological process) analysis using the website of https://david.ncifcrf.gov/. The RNA-seq expression data of TCGA-LIHC samples were downloaded from the TCGA pan-cancer atlas (https://gdc.cancer.gov/about-data/publications/pancanatlas). GSEA was performed by the clusterProfiler R package. Pathway information was obtained from Kyoto Encyclopedia of Genes and Genomes (KEGG, https://www.kegg.jp/) database. The Online tool (https://www.metaboanalyst.ca/home.xhtml) was used to do the pathway analysis for metabolomics data.

**Statistics and reproducibility**. Pearson correlation coefficient was used to evaluate the relationship between USP22 and PPARγ, ACC, ACLY and FASN protein and mRNA expression levels in human HCC tissues and TCGA database. One-way ANOVA-post-hoc pairwise comparison analysis was used to compare the means of more than two groups. Student's t-test (unpaired, two-tailed) and Wilcoxon-test (paired two-samples) was used to compare the mean value of two groups. Wilcoxon-test (paired two-samples) was used for metabolic data of tissues, and Student's t-test (unpaired, two-tailed) was used for metabolomics data of cells. The differences in survival were calculated using the Kaplan-Meier test. Data representative of two or more independent experiments. Bars and error represent mean ± standard deviations (SD) of replicate measurements and p < 0.05 is considered significant. Unless otherwise indicated, the experiments were performed independently in triplicate, and n is indicated in the figure legends. All the gel images of the relative protein expression were analyzed by ImageJ (version no.: 1.8.0_112; https://imagej.nih.gov/ij/). Statistical analysis was performed using the SPSS 21.0 software package (SPSS, Inc., Chicago, IL, USA).

**Reporting summary**. Further information on research design is available in the Nature Research Reporting Summary linked to this article.

## Data availability

All the experiment data that support the findings of this study are included within the paper, its Supplementary Information files, Source Data files and public repositories and also available from the corresponding author upon reasonable request. The raw RNA-seq data used in this study are available in the Sequence Read Archive (SRA, https://www.ncbi.nlm.nih.gov/sra/) under the Bioproject accession PRJNA809499. The LC-MS/MS data for PPARγ interactome are available within the article and its Supplementary Table 2. The data used in this study for gene expression profiling interactive analysis (http://gepia2.cancer-pku.cn) are available in The Cancer Genome Atlas (TCGA; https://tcga-data.nci.nih.gov/). Promoter sequences are available from Eukaryotic Promoter Database (EPD; https://epd.epfl.ch/). The reference library used in RNA-seq analysis (Mus musculus GRCm38.dna.primary_assembly.fa) is available in the GeneRIF (https://www.ncbi.nlm.nih.gov/gene/?term=GRCm38). The source data underlying Figs. 1–3, 5–8 as well as Supplementary Fig 3–5, 7 and 8 are provided as a Source Data file. All the other data supporting the findings of this study are available within the article and its Supplementary Information files. Source data are provided with this paper.

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

## Acknowledgements

We thank members of the Dr. Piao laboratory for helpful discussion. This study is supported by National Natural Science Foundation of China grants (No. 81972625 and No. 81672440, to H.-l.P., No. 21876169 to G.X., No. 81502024 to Z.N.), Liaoning Revitalization Talents Program (XLYC2002035 to H.-l.P.), Innovation program of science and research from the DICP, CAS (DICP ZZBS201803 to H.-l.P.), Project funded by China Postdoctoral Science Foundation (No. 2017M6110186 to Z.N.), the Youth Innovation Promotion Association CAS (2021186 to Xinyu Liu.).

## Author contributions

Z.N., G.X., G.T., and H.-l.P. devised and coordinated the project. H.-l.P. and G.X. supervised the project. Z.N., X.G., X.L. (Xiaolong Liu), and C.L. performed most of the experiments, Z.N., X.G., X.L. (Xiaolong Liu), C.L. and H.-l.P. analyzed data. A.W., X.W., W.W., H.C., W.Q., X.L. (Xinyu Liu), L.Z., C.M., J.D., Z.L., H.L., W.O., H.Q., D.C., T.X. and J.L. provided significant intellectual input. Z.N., G.T., G.X., and H-l.P. wrote the manuscript with input from all other authors.

## Competing interests

The authors declare no competing interests.
