## [Peer Review File · Nature Communications]

USP22 regulates lipidome accumulation by stabilizing PPAR γ in hepatocellular carcinomaREVIEWER COMMENTS

Reviewer #1 (Remarks to the Author); expert on hepatocellular carcinoma and metabolism:

Ning et al. conducted a study that explores the known pro-tumorigenic role of de-ubiquitination enzyme USP22 and transcription factor PPARg in HCC. They employed different liver cancer lines of human origin to show that PPARg-driven induction of lipogenic enzymes (ACC and ACLY) promotes xenograft growth. Notably, authors conducted bioinformatic analyses of already available HCC arrays to point to positive correlations between PPARg and lipogenic enzyme expression in HCC. Their findings also pointed to negative correlation between the transcript levels of PPARg or lipogenic enzymes with patient survival. Given that liver cancer is badly treatable malignancy, these findings in cellular models might point to possible future therapies based on inhibiting USP22. However, most of the conclusion are based on correlative observations while mechanistic studies do not irrefutably show the functional interaction between USP22/PPARg or PPARg/lipogenic enzymes and thus the manuscript requires further experimental work to firmly support those conclusions.

Major questions

1) Authors propose that PPARg is a target of USP22 de-ubiquitination (line 187-189). Although they find both proteins in complex and expression of USP22 seems to impact PPARg protein levels and ubiquitination, these findings do not prove that PPARg is a direct target of USP22. Neither these finding show that ubiquitination status of PPARg in their cellular models impact the recruitment of PPARg to promoters of its target genes. The functional observations were made in HEK293 cells (with exception of Fig3e) and it would be important to demonstrate that USP22 modulation in HCC lines impacts PPARg protein turnover and its transcriptional activity (nuclear expression, chromatin recruitment, transcriptional activity). It would be important to understand what ubiquitination sites USP22 is particularly affecting in PPARg?

2) Importantly, the role of PPARg in tumorigenesis is controversial with both pro and anti-tumorigenic function reported depending on the cancer type, inductor and associated genetic perturbations. The recent report that authors also cite in their manuscript (PMID 28394260) highlighted this complexity by showing that PPARg transcript is increased in HCC associated with increased insulin/Akt signaling. Authors should conceptualize their findings on potential involvement of USP22/PPARg in liver tumorigenesis taking into account known complexity of HCC.

3) Authors put forward on multiple occasions the message on a positive correlation between poor HCC prognosis and increased PPARg/USP22 transcripts (Fig5,7; ExtData1, 4,7). It would be advisable to regroup them together with analyses of triple correlation (USP22/PPARg/lipogenic enzymes) to the same section of the manuscript. Also, it is essential that authors clearly describe which data set were analyzed and put their findings in the context of known complexity of HCCs (transcriptional groups).

4) Given that PPARg2 is regarded as more potent pro-lipogenic isoform of PPARg, is there the specificity of USP22 to PPARg1 or PPARg2 proteins? What isoform of PPARg (g1 or g2) the authors have analyzed?

5) What is the mechanism of potential USP22-dependent PPARg selectivity towards inducing transcription of ACC and ACLY but not the other lipogenic enzymes such as FAS and SCD1? PPARg in liver was also shown to act transcriptionally upstream of aerobic glycolysis and lipogenesis (PMID 22334075). Do authors find the effect of USP22 selective to lipogenesis or HK2 and PKM2 transcript were also positively correlated with USP22/PPARg expression?

6) The authors overstate their findings (see below). As an example, the conclusion, line 316-319, is not supported by experimental evidence. The authors have demonstrated that expression of USP22, an enzyme with de-ubiquitinase activity, positively corelates with protein and transcript levels of transcription factor PPARg. The expression of USP22 is also corelated with expression of pro-lipogenic metabolic enzymes ACLY and ACC which are known transcriptional targets of PPARg. Given these positive correlations authors can conclude that USP22 acts upstream of PPARg yet its

direct implication as de-ubiquitinase of PPAR γ to promote transcription of ACLY and ACC should be experimentally addressed.

Also, the authors should test their hypothesis in HCC samples. According to their hypothesis the high expression of USP22 would result in low ubiquitination of PPAR γ , its nuclear localization and binding to promoters of pro-lipogenic metabolic enzymes positively correlating with increased transcript levels of those in HCC.

7) Given the positive correlation of USP22 and PPAR γ transcripts in HCC, authors should test if PPAR γ could serve as the transcription factor for USP22 expression and in this way maintain its own expression and transcriptional activity in HCC.

Specific Comments

1) The authors should revise the figures and their description in the text.

-The calling of figures in the text should be harmonized in the order they are presented. As an example, ExtData1b is coming first instead of ExtData1a (the same for 1d and 1c; ExtData3a and 3b, ExtData6a-d).

-The units on all graphs should be indicated (e.g. ExtData1b, 1c). The color code of bars should be indicated (ExtData1b, 1c, 2c, 2g). The authors should avoid using cut axes (Fig 2d, 2e, 6e, 6f).

-The Fig1C heatmap of significantly changing metabolites should be re-verified as for several metabolites the differences are hardly noticeable as judged by color (e.g. FA C22:2, C24:2, LPC C20:0, C22:6). In the same line, as judging from Fig1b, there are about 30 metabolites that seems to be significantly changing in T vs NT samples, yet the heatmap has close to 50 lines. The authors should provide more information on metabolomics analyses to increase clarity of these important analyses.

-The authors should specify how analyses in ExtData1b were conducted and what is exactly represented on graphics (number of USPs and the abbreviations on Y axis).

-It is not clear how authors draw the conclusion (line 123-126) on inverse association of USP22 with FA degradation in HCC. Besides the Fatty acid degradation is 5th item in the graphics and the authors do not comment on other items.

2) On numerous instances the conclusions come preliminary as they are not backed by the data presented in the corresponding section. As example, line 111-115, statement that lipid synthesis is induced in tumoral tissue is preliminary as increased levels of FA and other lipid species could be also caused by defects in their degradation or increased uptake. Similarly, line 154-156, the conclusion that UPS22 promotes de novo lipid synthesis is premature as authors did not study FAO. Finally, in discussion section, although the authors claim that FAO is not involved in the phenotype observed in cells with USP22 OE or KD, they did not formally rule it out (line 350-352). Also, the conclusion that USP22 regulates fatty acid biosynthesis through PPAR γ stabilization is overstatement (line 265-266 and line 288-289). The authors did not study the sites of ubiquitination involved and did not demonstrate that it is through this specific mechanism USP22 controls PPAR γ for ACC and ACLY transcription. The alternative formulation could be: PPAR γ contributes to ACLY and ACC driven lipogenesis downstream of USP22.

3) Authors should comment that in two cell lines USP22 GOF produces distinct phenotypes which are suggestive of FAO defects upon USP22 overexpression (Fig2c vs Fig 2g). To this end, the authors should specifically analyze the carnitine-FA conjugates and demonstrate the status of FAO in MHCC-97 cell models. It would be important to address whether USP22 affects HCC line tumorigenic properties by acting on their FAO.

4) The choice of HCC lines for analyses should be explained. It is no clear why not the same lines were used for GOF and LOF experiments especially that MHCC97 lines (L vs H) are known to have different proteomic and metabolic profiles compared to non-metastatic HCC lines (PMID: 30588254, 15243804). What is the relevance of Bel7402 cell line? The analyses in SNU449 line should be commented. What are the expression levels of USP22 and how metabolic profiles compare in these different cell lines (parental and engineered)? The authors should provide at least basic characterization of the engineered cell lines (proliferation/survival/morphology in depleted cells and in

cells overexpressing USP22? In the same line, given that it is embryonic kidney line, it is difficult to understand why HEK293 model with OE of USP22 was chosen for proteomics analyses (Fig3a).

5) The data in section line 218-230 seem repetitive to the ones presented earlier in the manuscript. The analyses in HCC provide the same message as in Fig1. The info on cell lines would be more appropriate in section were the cell line work was initiated (Fig2). It is important to include analyses of PPARg and lipogenic enzymes alongside USP22 presented on ExtData4b.

6) The methodology should be better described to facilitate the understanding of the work.

-The methodology of bioinformatics analyses should be elaborated indicating what datasets and how were analyzed.

-For metabolomics analyses, the details on patient selection should be provided (tumor type, patient sex, comorbidity, primary/secondary cancer...). How the adjacent tissue was controlled should be specified. The status (fasting/fed) and preferably time of the day should be indicated (lipid metabolism is circadian). The complete list of the compounds identified should be provided as a table. Given that authors comment on enrichment of specific lipid species in their metabolomics analyses (e.g. Fig.1d, 2c), they should detail the parameters of the metabolomics analyses including the information on all metabolites identified.

- In the same line, it is difficult to understand what is exactly presented as metabolomics findings e.g. on ExtData 2b,c and Fig2b. Authors should detail why panels contain different metabolites (e.g. Carn-C3:0 or FA C15) and how normalization was performed? This clarifying information should be introduced in the result section (specify the untargeted or targeted analyses, compound identification, data normalization and bioinformatic analyses).

7) For analyses in cells, what are the differences in the experimental conditions as USP22 overexpression in one case increases and in other case has no impact on PPARg levels (Fig3b, 3c, Fig 6a, 6b ExtDat3b, 3i vs Fig3f, 3g).

8) In Fig3, it is difficult to understand which tag corresponds to which overexpressed protein in analyses of complex/ubiquitination. It would be advisable to include this info in figure labeling. In the same section, the additional controls should be included. Panels Fig3h-j need control of non-specific binding of Ub-HA to beads as modulation of USP22 is expected to modify the poly-Ub. Also, authors should control the levels of Ub-HA in total extracts in all experimental conditions to demonstrate that the overexpression was similar.

9) In Fig 3e authors demonstrate the complex between endogenous PPARg and USP22, how they explain the difference in migration of USP22 and PPARg in input and in IP samples?

10) In Fig.4, the presentation of transcriptomic analyses should be improved. It is not clear how RT-qPCR analyses were conducted in Fig4c (and Fig6c,d) as there is no group assigned as 1. Neither it is clear how the bulk data were analyzed and how different is the expression of these 30-top gene hits presented on heat map.

11) In Fig. 4f, the examples of correlative IHC analyses of USP22 and ACC/ACLY are not clear. From the captures presented, the samples of USP22 labelled as low or high expression seem inverted? Authors should provide the information on steatosis status and PPARg expression in these tumor samples.

12) In ExtData5 and Fig.5, the authors show that knockdown or overexpression of USP22 affect cell proliferation. Does it also impact apoptosis? It would be important to characterize the status of PPARg in xenografts (expression, ubiquitination, transcriptional activity). Since the mechanism is transcriptional control by PPARg, the transcript levels of ACC and ACLY should be analyzed as well.

13) The text of the manuscript requires further editing. For example, it would be advisable to improve clarity of the formulations in lines 31-34, 52-54, 331-332, 332-334 which are difficult to follow.

14) Authors should revise the reference list.

-There are double references e.g. 5=11, 31=43.

-Line 78-80, used citations give impression all related to HCC which is not the case for Ref 19 and 8.

-Relevance of ref 30 in the context of line 92-95 is not clear.

-The reference to the statement on line 320 should be added. According to the WHO, HCC is in top five cancer related deaths.

-Not accurate statements, as in line 92-93. There is a wealth of literature on the anti-tumorigenic as well as pro-tumorigenic role of PPAR γ depending on the cancer type and genetic context. Authors should make an effort to introduce it pointing to the gaps of knowledge on the upstream control of PPAR γ expression and activity in relevance to their work.

15) Line 93-95, the authors should cite earlier report on Akt2 driven induction of PPAR γ in the context of steatosis associated liver cancer (PMID 22334075) in addition to PMID 28394260. In general, authors should mention in introduction the complexity of HCC disease (different groups with distinct transcriptional signatures as established by works of Zucman-Rossi lab and others). Furthermore, authors should overview the metabolomics findings in HCC and highlight the advancements that their study brings.

Reviewer #2 (Remarks to the Author); expert on ubiquitin system:

Revision NCOMMS-20-38410A-Z

In the present manuscript, Ning and collaborators identify the ubiquitin-specific protease 22 (USP22) as a key player of de novo fatty acid synthesis, involved in the regulation of the PPAR γ -ACLY/ACC axis in HCC tumorigenesis. In particular, they demonstrate that USP22 interacts with and promotes PPAR γ stabilization through K48-linked deubiquitination, thus inducing and increasing transcription of the lipid synthesis enzymes ACC and ACLY. Interestingly, the authors find that USP22 expression is positively correlated with PPAR γ expression in HCC, and simultaneously, high expression of USP22 and PPAR γ or USP22, ACC and ACLY contributes to tumor progression and it is associated with poor prognosis.

HCC is one of the most frequent solid tumors worldwide, characterized by clinical aggressiveness, resistance to conventional chemotherapy, and high lethality. For these reasons, there is an urgent need to better define the molecular alterations involved in the pathogenesis of HCC to develop new innovative therapeutic strategies. Albeit USP22 has already been described as promising therapeutic target in HCC (Ling S, et al. Gut 2020;69:1322–1334), the new role of the USP22-PPAR γ -ACLY/ACC axis on the regulation of the lipogenesis and its implication in HCC tumorigenesis, provide a new therapeutic approach targeting fatty acid synthesis for the treatment of this tumor.

The manuscript is well structured and the data shown are interesting and well presented. I still have a few concerns about the conclusions made from the obtained results. In particular, co-localization, protein stability assays and ubiquitylation experiments were performed using overexpressed proteins only and not in HCC cells, thus dampening my enthusiasm.

Comments

Figure 3d: the co-localization between USP22 and PPAR γ should be performed for endogenous proteins.

Figure 3e: There is an upward shift of the protein bands shown in the IP lines compared to the corresponding input lines. This point needs to be explained.

Figure 3f: the ability of USP22 to stabilize PPAR γ should be investigated by CHX experiments

analyzing endogenous levels of PPAR γ in USP22 stably knocked down in MHCC-97H and/or USP22 transduced overexpression in MHCC-97L.

Figures 3h, I and j: to appreciate the regulation of USP22 on PPAR γ ubiquitylation, endogenous levels of ubiquitylated PPAR γ should be evaluated in the presence of ectopic expression of USP22 or USP22 C1856S, and in genetic depletion of USP22 in HCC cell lines.

It is my opinion that the study could increase the novelty by investigating the potential role of USP22 in pVHL-mediated ubiquitylation of PPAR γ (Metabolism 2020 Sep;110:154302. doi: 10.1016/j.metabol.2020.154302).

In light of authors' conclusions, should be important to analyze the endogenous levels of PPAR γ , ACC and ACLY in Figure 1f.

In Figure 5h, the levels of PPAR γ should be showed.

For in vivo studies, the authors use in vivo xenografts (flank tumor) from HCC cell lines. An orthotopic HCC model would have been more relevant for supporting authors conclusions. The role of USP22 in the regulation of HIF1 α in HCC deserves to be discussed.

MW should be added in each western blot.

Reviewer #3 (Remarks to the Author); expert on metabolism, metabolomics and transcriptomics:

The authors investigated the molecular mechanism about the critical role of elevated de novo lipogenesis (DNL) in hepatocellular carcinoma (HCC) development. They performed large number of computational and experimental analysis and identified ubiquitin-specific protease 22 (USP22) as a key regulator for de novo fatty acid synthesis. They found that USP22 directly interacts with, deubiquitinates and stabilizes PPAR γ through K48-linked deubiquitination, and in turn, this stabilization increases ACC and ACLY transcription.

In addition, they found that USP22 promoted the de novo synthesis of fatty acid-based on labelling studies. They also reported that USP22 expression positively correlates with PPAR γ expression, and simultaneously, high expression of USP22 and PPAR γ or USP22, ACC and ACLY is associated with a poor prognosis in HCC

I agree with the authors that, they identified a previously undescribed USP22-regulated lipogenesis molecular mechanism that involves the PPAR γ -ACLY/ACC axis in HCC tumorigenesis. However, I do not agree that the mechanism they suggested in their study is supported by the experiments they performed. I have serious concerns related to the interpretation of the results. It would be useful to clarify the points below before the publication of the paper.

1) The authors reported that USP22 was highly expressed in HCC and cholangiocarcinoma but not in other cancer types. Based on TCGA database and Human Pathology Atlas, this statement is not correct. It is actually the other way around. <https://www.proteinatlas.org/ENSG00000124422-USP22/pathology>

2) Subsequently, pathway enrichment analysis based on the TCGA database revealed that USP22 was significantly negatively associated with FA degradation in HCC (Extended Data Fig. 1D). Their analysis indicated that the differences in the amount of fatty acids may be due to the alterations in FA oxidation rather than DNL. This may be one of the reasons why the authors could not verify the changes in the expression of FASN and SCD. In that case, the entire paper has to be rewritten considering this alternative potential mechanism.

- 3) It is known that the expression of FASN and SCD is increased or decreased based on the activity of DNL. It is unexpected to see that the expression of the FASN and SCD is not changed in all experiments that the authors performed.
- 4) If the authors focus on the DNL, the authors should measure TAG levels in the cell and animal experiments. It is really surprising that the authors have not measured TAG in any of the experiments performed.
- 5) In Figure 2 about the cell experiments, how the cell growth is affected? Is it normalized with cell growth? As I mentioned above, the authors should measure TAG levels in Figure 2b.
- 6) Based on Figure 3, this reviewer was not surprised to see that there is a link between USP22, PPARG and HADH which is involved in fatty acid oxidation. It is interesting that none of the genes on the list is involved in DNL. As I mentioned above, I strongly suggest the authors investigate the specific role of USP22 in lipid metabolism specifically in fatty acid oxidation.
- 7) Based on Figure 4, the authors should investigate the role of GCLM, SLC7A1, NNMT and other redox related genes. I think these genes have a critical role in alterations of the lipids specifically in fatty acid oxidation.
- 8) In Figure 5i, the authors measured the lipid structures. It is surprising that the abundant FA structures (C16:0, C18:0) are not on the list. C18:1 is also showing an opposite trend. I strongly recommend the authors to measure the total TAG levels as well.
- 9) In Figure 5i, It also interesting to see that Glutathione levels are significantly increased and this supports the changes in the expression of the genes in Figure 4. The carnitine levels are significantly increased as well.
- 10) In Figure 6 i and j, both control and shUSP22 had 500 Total Volume. On the other hand, shACC, shACLY and shPPARG had around 100 Total Volume. I suggest the authors add an explanation for these differences.
- 11) USP22 have been reported as a prognostic gene in the human pathology atlas. I suggest the authors cite Uhlen et al, 2017 Science paper to support their analysis.

REVIEWER COMMENTS

Reviewer #1 (Remarks to the Author); expert on hepatocellular carcinoma and metabolism:

Ning et al. conducted a study that explores the known pro-tumorigenic role of de-ubiquitination enzyme USP22 and transcription factor PPAR γ in HCC. They employed different liver cancer lines of human origin to show that PPAR γ -driven induction of lipogenic enzymes (ACC and ACLY) promotes xenograft growth. Notably, authors conducted bioinformatic analyses of already available HCC arrays to point to positive correlations between PPAR γ and lipogenic enzyme expression in HCC. Their findings also pointed to negative correlation between the transcript levels of PPAR γ or lipogenic enzymes with patient survival. Given that liver cancer is badly treatable malignancy, these findings in cellular models might point to possible future therapies based on inhibiting USP22. However, most of the conclusion are based on correlative observations while mechanistic studies do not irrefutably show the functional interaction between USP22/PPAR γ or PPAR γ /lipogenic enzymes and thus the manuscript requires further experimental work to firmly support those conclusions.

RE: Thanks for reviewer's important suggestions and comments for our work. We have carefully respond the below issues and fixed them in the revised manuscript. Please refer to the following replies.

Major questions

1) Authors propose that PPAR γ is a target of USP22 de-ubiquitination (line 187-189). Although they find both proteins in complex and expression of USP22 seems to impact PPAR γ protein levels and ubiquitination, these findings do not prove that PPAR γ is a direct target of USP22. Neither these finding show that ubiquitination status of PPAR γ in their cellular models impact the recruitment of PPAR γ to promoters of its target genes. The functional observations were made in HEK293 cells (with exception of Fig3e) and it would be important to demonstrate that USP22 modulation in HCC lines impacts PPAR γ protein turnover and its transcriptional activity (nuclear expression, chromatin recruitment, transcriptional activity). It would be important to understand what ubiquitination sites USP22 is particularly affecting in PPAR γ ?

RE: Thanks for your important suggestions.

1) To clarify whether USP22 directly targeting on PPAR γ and deubiquitination, we performed *in vitro* protein interaction assay and deubiquitination assay. We found that purified His-USP22 and GST-PPAR γ proteins existed direct binding interaction, and *in vitro* deubiquitination assays also confirmed that the USP22 significantly deubiquitination of PPAR γ . We provided this result in New Figure 1 (Fig. 4d and Fig 5c in revised version) and revised the manuscript.

2) Next, to clarify whether USP22-mediated the stability of PPAR γ protein affects its recruitment to target gene promoters, we performed ChIP experiments in MHCC-97H-shUSP22-1/2 cells (MHCC-97H cells transduced with two independent USP22

shRNAs) for the PPRE elements of the ACC and ACLY promoters, and the results confirmed that the enrichment of PPAR γ at the PPRE site on *ACLY* or *ACC* promoter decreased significantly after USP22 depletion. We provided this result in New Figure 2 (Fig.6c, d in revised version).

3) To clarify the USP22 deubiquitination of PPAR γ in HCC cell lines, we examined the ubiquitination, protein stability, nucleus and cytoplasmic fraction assay and the transcriptional activity of PPAR γ in MHCC-97H-shUSP22-1/2 and MHCC-97L-USP22 (MHCC-97L cells stably overexpressing USP22) cells. We provided this result in New Figure 3 (Fig. 5a, 5b, 5i, 5j, 6a, and 6b in revised version).

4) To identify the deubiquitination sites by USP22 targeting on PPAR γ , we performed several studies as followed. First, based on literatures we confirmed the E3 ubiquitin ligases pVHL and CUL4B regulated lysine sites. We found that USP22 significantly decreased the pVHL and CUL4B involved ubiquitination, it's indicated that USP22 regulates deubiquitination of PPAR γ through other lysine sites. Next, to identify which lysine sites on the PPAR γ were involved in USP22 deubiquitination process, we generated the several fragments of PPAR γ based on different domains, and found that the domains which containing the DBD domain could deubiquitinated by USP22. Consequently, we mutated all 7 lysine sites on the DBD domain, and found the Lys-169 may important site for USP22 deubiquitination of PPAR γ . Finally, we found that the USP22 did not inhibit ubiquitination levels of the five lysine sites mutated PPAR γ -5KR (K169/240/265/404/434R). We provided this result in New Figure 4 (Fig. 5d, 5e, 5f, 5g, 5h and Supplementary Fig. 6f in revised version).

New Figure 1. USP22 directly targeting on PPAR γ and deubiquitination in *in vitro*.
a) GST pull-down assay with purified His-USP22 and GST-PPAR γ . PD: pull-down (Fig 4d in revised version). **b)** *In vitro* deubiquitination assay of ubiquitinated PPAR γ protein with purified His-USP22 (Fig 5c in revised version).

New Figure 2. USP22-mediated the stability of PPAR γ protein affects its recruitment to target gene promoters. a) Illustration of PPRE site in *ACLY* promoter and the predicted PPRE site in *ACC* promoter (Fig 6c in revised version). b) Chromatin immunoprecipitation (ChIP) analysis of PPAR γ binding to the *ACACA* and *ACLY* promoters in MHCC-97H-shUSP22-1/2 cells. qPCR was performed with primers specific to the PPAR γ -binding motifs. Data were normalized to the input. The data shown represent the means (\pm SD) of biological triplicates. (Fig 6d in revised version).

New Figure 3. USP22 deubiquitination of PPAR γ in HCC cells. a-b) Ubiquitination assay of PPAR γ in MHCC-97H-shUSP22-1/2 cells (a) or MHCC-97L-USP22, MHCC-97L-USP22 C185S cells (b) treated for 6h with 10 μ M MG132. (Fig. 5a and 5b in revised version). c-d) Stability analysis of PPAR γ protein in MHCC-97H-shUSP22-1/2 cells (c), MHCC-97L-USP22, MHCC-97L-USP22 C185S cells (d) and treated with 40 μ M cycloheximide (CHX) for indicated times. (Fig. 5i and 5j in revised version). e) Western blot analysis of USP22 and PPAR γ in cytoplasmic and nucleus fractions of MHCC-97H-shUSP22-1/2 cells and MHCC-97L-USP22 cells. (Fig. 6a in revised version). f) DNA

binding activity of PPAR γ in MHCC-97H-shUSP22-1/2 cells and MHCC-97L-USP22 cells. (Fig. 6b in revised version).

New Figure 4. USP22 deubiquitinates and stabilizes PPAR γ . **a**) Ubiquitination assay of PPAR γ in HEK293T cells cotransfected with HA-Ub, Flag-PPAR γ , Flag-PPAR γ -K404/434R, Myc-USP22 or V5-pVHL and treated with 10 μ M MG132 for 6 h. (Fig. 5d in revised version). **b**) Ubiquitination assay of PPAR γ in HEK293T cells cotransfected with HA-Ub, Flag-PPAR γ , Flag-PPAR γ -K240/265R, Myc-USP22 or V5-CUL4B and treated with 10 μ M MG132 for 6 h. (Fig. 5e in revised version). **c**) Ubiquitination assay of FL (full length), AD (AF-1-DBD), DH (DBD-Hinge), D (DBD),

DHL (DBD-Hinge-LBD) and HL (Hinge-LBD) domain of PPAR γ in HEK293T cells cotransfected with HA-Ub and Myc-USP22, and treated with 10 μ M MG132 for 6 h. (Fig. 5f in revised version). **d**) Ubiquitination assay of DBD domain of PPAR γ in 293T cells cotransfected with HA-Ub, Myc-USP22, Flag-DBD, Flag-DBD-K117R, Flag-DBD-K132R, Flag-DBD-K142R, Flag-DBD-K161R and Flag-DBD-K169R and treated with 10 μ M MG132 for 6 h. (Fig. 5g in revised version). **e**) Ubiquitination assay of PPAR γ -K156/157R in HEK293T cells cotransfected with HA-Ub, SF-PPAR γ -K156/157R, Myc-USP22 and treated with 10 μ M MG132 for 6 h. (Supplementary Fig. 6f in revised version). **f**) Ubiquitination assay of PPAR γ in HEK293T cells cotransfected with HA-Ub, Myc-USP22, SF-PPAR γ and SF-PPAR γ -5KR (K169/240/265/404/434R) and treated with 10 μ M MG132 for 6 h. (Fig. 5h in revised version).

2) Importantly, the role of PPAR γ in tumorigenesis is controversial with both pro and anti-tumorigenic function reported depending on the cancer type, inductor and associated genetic perturbations. The recent report that authors also cite in their manuscript (PMID 28394260) highlighted this complexity by showing that PPAR γ transcript is increased in HCC associated with increased insulin/Akt signaling. Authors should conceptualize their findings on potential involvement of USP22/PPAR γ in liver tumorigenesis taking into account known complexity of HCC.

RE: Thanks for your important suggestions. We agree with reviewer's this comment. Indeed, the function of PPAR γ in tumorigenesis is controversial. We also found that the expression of both ACC and ACLY was upregulated after the activation of AKT by serum stimulation. However, our results also confirmed that USP22 dependent PPAR γ , ACC and ACLY upregulation might not be depends on AKT activation. Moreover, in this study, indeed we also observed that compared with the tumors formed USP22 overexpression upregulated PPAR γ , ACC and ACLY expression; however, this upregulation was inhibited by knockdown of PPAR γ in USP22-overexpressing MHCC-97L cells . In addition, PPAR γ overexpression restored the downregulation of ACC and ACLY in USP22-knockdown MHCC-97H cells. Overall, PPAR γ plays a key role in USP22-driven tumorigenesis. We provided this result in New Figure 5 (Supplementary Fig. 7d and Fig. 7 in revised version).

New Figure 5. USP22 dependent PPAR γ , ACC and ACLY upregulation might not

be depends on AKT activation. USP22 dependent PPAR γ , ACC and ACLY upregulation might not be depends on AKT activation. (Supplementary Fig. 7d in revised version).

3) Authors put forward on multiple occasions the message on a positive correlation between poor HCC prognosis and increased PPARg/USP22 transcripts (Fig5,7; ExtData1, 4,7). It would be advisable to regroup them together with analyses of triple correlation (USP22/PPARg/lipogenic enzymes) to the same section of the manuscript. Also, it is essential that authors clearly describe which data set were analyzed and put their findings in the context of known complexity of HCCs (transcriptional groups).

RE: Thanks for your suggestions. We reexamined and revised the manuscript and provide all information in revised version as Fig. 8 and Supplementary Fig. 8.

4) Given that PPARg2 is regarded as more potent pro-lipogenic isoform of PPARg, is there the specificity of USP22 to PPARg1 or PPARg2 proteins? What isoform of PPARg (g1 or g2) the authors have analyzed?

RE: Thanks for your suggestions. We targeted PPAR γ 1 in this study. Although PPAR γ 2 has a stronger function in promoting adipogenesis, but it is mainly expressed in adipocytes.

5) What is the mechanism of potential USP22-dependent PPARg selectivity towards inducing transcription of ACC and ACLY but not the other lipogenic enzymes such as FAS and SCD1? PPARg in liver was also shown to act transcriptionally upstream of aerobic glycolysis and lipogenesis (PMID 22334075). Do authors find the effect of USP22 selective to lipogenesis or HK2 and PKM2 transcript were also positively correlated with USP22/PPARg expression?

RE: Thanks for your suggestions.

1). We found that among the 4 enzymes of fatty acid synthesis genes (ACLY/ACC/FASN/SCD), USP22 only affects the expression of ACC and ACLY in HCC cells, and in both TCGA HCC database and two independent HCC TMA assays (LV1021, HLivH180Su11), and confirmed that the USP22 significantly positive correlate with ACC and ACLY expression. In addition, we also found that the USP22 could regulates the PPAR γ -mediated transcription of ACLY and ACC. We speculate that ACC and ACLY are more dependent on PPAR γ regulation during USP22-mediated fatty acid synthesis, but FASN and SCD may be regulated by other lipogenic transcription factors such as SREBF1 and PPAR α .

2). We confirmed that USP22 promotes triglyceride accumulation by TG assay and oil red staining (Fig 2e, f; Fig 7e, f in revised version), and we also found a positive correlation between USP22 and GPAT1 only among GPAT1, DGAT and MOGAT2, the key enzymes of triglyceride synthesis, in the TCGA database (New Figure 6). But, we do not bring these data into revised version, since this subject lies beyond the scope of

our investigation.

New Figure 6. USP22 positively correlated with the key enzymes of triglyceride synthesis in TCGA database.

3). According to review's suggestion, we investigated the USP22-mediated HK2 and PKM2 expression in HCC cells. The expression of PKM2 and HK2 were not affect by knockdown or overexpression of USP22 in the HCC cells. However, *USP22* mRNA positively correlated with *PKM* and *HK2* in the TCGA HCC data.

New Figure 7. USP22 did not affect PKM2 and HK2 in HCC cells.

6) The authors overstate their findings (see below). As an example, line 316-319, is not supported by experimental evidence. The authors have demonstrated that expression of USP22, an enzyme with de-ubiquitinase activity, positively correlates with protein and transcript levels of transcription factor PPAR γ . The expression of USP22 is also correlated with expression of pro-lipogenic metabolic enzymes ACLY and ACC which are known transcriptional targets of PPAR γ . Given these positive correlations authors can conclude that USP22 acts upstream of PPAR γ yet its direct implication as de-ubiquitinase of PPAR γ to promote transcription of ACLY and ACC should be experimentally addressed. Also, the authors should test their hypothesis in HCC samples. According to their hypothesis the high expression of USP22 would result in low ubiquitination of PPAR γ , its nuclear localization and binding to promoters of pro-lipogenic metabolic enzymes positively correlating with increased transcript levels of

those in HCC.

RE: Thanks for your suggestions.

1). We observed that knockdown of PPAR γ in the USP22-overexpressing MHCC-97L cells significantly inhibited USP22-driven lipid synthesis and tumorigenesis. In addition, PPAR γ overexpression restored the downregulation of ACC and ACLY and lipid synthesis and tumorigenesis in the USP22-knockdown MHCC-97H cells. **These results are presented in Fig. 7 in revised version.**

2). As review's suggestion we reexamined all the experiments in the HCC cells and revised manuscript. (1). USP22 deubiquitinates and stabilizes PPAR γ , while knockdown of USP22 or C185S mutant of USP22 does not have this function. (2). USP22 stabilizes PPAR γ expression in the nucleus rather than in cytoplasm. (3). USP22 was able to upregulate the transcriptional activity of PPAR γ . (4). The ChIP experiments confirmed PPAR γ antibody were enriched for the ACLY and ACACA promoter region in control cells compared to USP22 depleted cells. These results are presented in **Fig. 5a, 5b, 5i, 5j, 6a, 6b, 6c, and 6d in revised version** (please also see Major question 1).

3). As review's suggestion we performed deubiquitination assay in HCC tissues and xenograft tumor tissues. (1). The ubiquitination level of PPAR γ was decreased in cancer tissues with high USP22 expression compared with adjacent normal tissues. (2). The ubiquitination level of PPAR γ was significantly upregulated in tumors formed from USP22 knockdown group compared with the control group. (3). High expression of PPAR γ relatively correlated with USP22 expression in nucleus fraction from HCC tissues. We provided this result in New Figure 8 (**Supplementary Fig. 6c, 6d, and Supplementary Fig. 7a. in revised version**).

New Figure 8. PPAR γ deubiquitination assay in HCC tissues. **a)** Ubiquitination assay of PPAR γ in HCC cancer tissues and adjacent normal tissues (Patient #1, #2 and #5) (**Supplementary Fig. 6c in revised version**). **b)** Ubiquitination assay of PPAR γ in tumors derived from MHCC-97H-shUSP22-1 and shControl cells. (**Supplementary Fig. 6d in revised version**). **c)** Western blot analysis of USP22 and PPAR γ expression in cytoplasmic and nucleus fractions of HCC cancer tissues. Patients #1, #2 and #5, USP22 high expression. Patient #3, #6, #7, USP22 low expression. (**Supplementary Fig. 7a in revised version**)

7) Given the positive correlation of USP22 and PPAR γ transcripts in HCC, authors should test if PPAR γ could serve as the transcription factor for USP22 expression and in this way maintain its own expression and transcriptional activity in HCC.

RE: Thanks for your suggestions. As review's suggestion we performed of PPAR γ antagonist experiment. We found that there are no significant alterations in *USP22* expression under treatment of an antagonist of PPAR γ (T0070907, 5nM, 24h, selleck) in HCC cells. We speculate that the correlation between *USP22* and PPAR γ in the TCGA database may not be directly derived from the transcriptional regulation of PPAR γ .

New Figure 9. PPAR γ antagonist was not affect *USP22* expression.

Specific Comments

- 1) The authors should revise the figures and their description in the text.
 - The calling of figures in the text should be harmonized in the order they are presented. As an example, ExtData1b is coming first instead of ExtData1a (the same for 1d and 1c; ExtData3a and 3b, ExtData6a-d).
 - The units on all graphs should be indicated (e.g. ExtData1b, 1c). The color code of bars should be indicated (ExtData1b, 1c, 2c, 2g). The authors should avoid using cut axes (Fig 2d, 2e, 6e, 6f).
 - The Fig1C heatmap of significantly changing metabolites should be re-verified as for several metabolites the differences are hardly noticeable as judged by color (e.g. FA C22:2, C24:2, LPC C20:0, C22:6). In the same line, as judging from Fig1b, there are about 30 metabolites that seems to be significantly changing in T vs NT samples, yet the heatmap has close to 50 lines. The authors should provide more information on metabolomics analyses to increase clarity of these important analyses.
 - The authors should specify how analyses in ExtData1b were conducted and what is exactly represented on graphics (number of USPs and the abbreviations on Y axis).
 - It is not clear how authors drawn the conclusion (line 123-126) on inverse association of USP22 with FA degradation in HCC. Besides the Fatty acid degradation is 5th item in the graphics and the authors do not comment on other items.

RE: 1). We have unified the names of the images in the new version, such as Fig1a,b,c and Supplementary Figure1a, b, c, etc.

2). In the revised version the order of the images has been adjusted according to the order of the result descriptions.

3). We have labeled all the different color bars in the new version of the image, and also avoided the application of cut axis.

4). We re-analyzed the metabolomics data of Figure 1C, described as follows: We identified a total of 176 metabolites, of which 47 (upregulated: 26; downregulated: 21) were significantly different between cancer and adjacent normal tissues (Fig. 1b). Interestingly, metabolites that are significantly upregulated in cancer tissues are almost all lipids and lipid-like metabolites (22/26) (Fig. 1b), which are mainly composed of fatty acids (FAs), phosphatidylcholine (PC), lysophosphatidylcholine (LPC), phosphatidylethanolamine (PE), lysophosphatidylethanolamine (LPE), and sphingomyelin (SM) (Fig. 1c).

5). The revised version of **Supplementary Fig 1a** is described as follows: The number of USP family members associated with prognosis (red represents unfavorable, green represents favorable) in various tumors was analyzed by Kaplan–Meier analysis based on TCGA database. (LIHC: Liver hepatocellular carcinoma, LGG: Brain Lower Grade Glioma, KIRC: Kidney renal clear cell carcinoma, ACC: Adrenocortical carcinoma, PRAD: Prostate adenocarcinoma, KICH: Kidney Chromophobe, SARC: Sarcoma, KIRP: Kidney renal papillary cell carcinoma, HNSC: Head and Neck squamous cell carcinoma, MESO: Mesothelioma, SKCM: Skin Cutaneous Melanoma, UCEC: Uterine Corpus Endometrial Carcinoma, THCA: Thyroid carcinoma, BRCA: Breast invasive carcinoma, LAML: Acute Myeloid Leukemia, OV: Ovarian serous cystadenocarcinoma, STAD: Stomach adenocarcinoma, LUSC: Lung squamous cell carcinoma, COAD: Colon adenocarcinoma, UVM: Uveal Melanoma, UCS: Uterine Carcinosarcoma, CESC: Cervical squamous cell carcinoma and endocervical adenocarcinoma, BLCA: Bladder Urothelial Carcinoma, READ: Rectum adenocarcinoma, LUAD: Lung adenocarcinoma, ESCA: Esophageal carcinoma, PAAD: Pancreatic adenocarcinoma, GBM: Glioblastoma multiforme, CHOL: Cholangio carcinoma)

6). In the previous version, the pathway enrichment data were obtained from the website (<https://hgserver1.amc.nl/cgi-bin/r2/main.cgi?&&species=hs&&>), and in the new version, we re-performed the GSEA pathway enrichment analysis for the TCGA HCC data, and the results are shown in **Supplementary Figure 1c**. In this result we can see that the synthesis of unsaturated fatty acids is in the second place and the degradation of fatty acids is in the seventh place. In addition, we verified whether USP22 affects lipid accumulation through fatty acid degradation in the third part of the results.

2) On numerous instances the conclusions come preliminary as they are not backed by the data presented in the corresponding section. As example, line 111-115, statement that lipid synthesis is induced in tumoral tissue is preliminary as increased levels of FA and other lipid species could be also caused by defects in their degradation or increased uptake. Similarly, line 154-156, the conclusion that UPS22 promotes de novo lipid synthesis is premature as authors did not study FAO. Finally, in discussion section, although the authors claim that FAO is not involved in the phenotype observed in cells with USP22 OE or KD, they did not formally rule it out (line 350-352). Also, the conclusion that USP22 regulates fatty acid biosynthesis through PPAR γ stabilization is overstatement (line 265-266 and line 288-289). The authors did not study the sites of ubiquitination involved and did not demonstrate that it is through this specific

mechanism USP22 controls PPAR γ for ACC and ACLY transcription. The alternative formulation could be: PPAR γ contributes to ACLY and ACC driven lipogenesis downstream of USP22.

RE: Thanks for your suggestions. We reexamined and revised the manuscript and provide all information in revised version.

1). We first confirmed that USP22 promotes the synthesis of glucose-derived fatty acids by [U- 13 C] glucose tracing experiments in Fig.2d and Supplementary Fig. 4b in revised version.

2). To clarify whether USP22 is involved in lipid accumulation in HCC cells by inhibiting fatty acid oxidation, we first observed in the metabolomic data of MHCC-97H-shUSP22 and MHCC-97L-USP22 cells that acylcarnitine of both USP22 knockdown and overexpression showed an upregulation trend, which might suggest that the acylcarnitine alteration may be not related to USP22 expression changes. In addition, we performed GSEA pathway enrichment analysis of MHCC-97H-shUSP22-1/2 transcriptome data and found that USP22 was only associated with fatty acid synthesis and not with fatty acid degradation. We then examined the expression of fatty acid degradation-related enzymes (CPT1A, CPT2, ACOX1, ACADL and ECSH1) in USP22 knockdown and overexpression cell lines, and found that these enzymes were unchanged after USP22 knockdown or overexpression. CPT is the key enzyme for process of fatty acid degradation, therefore, we further analyzed the correlation between USP22 and CPT in TCGA database. The results revealed that USP22 did not correlate with both CPT1A and CPT2, respectively. These results are presented in Supplementary Figure 3d,3e; Supplementary Figure 5a,5c,5d,5e and 5i in revised version.

3). Thanks for your suggestions. As review's suggestion we performed USP22 affecting fatty acid uptake experiment. By adding C 13 -labeled palmitic acid to the medium of SNU449-Control and USP22 overexpressed cells, the intracellular C 13 -labeled palmitic acid content was detected after 30 min, and we found that USP22 overexpression failed to affect the intracellular C 13 -labeled palmitic acid. In addition, we found that the expression of CD36 and FABP were not altered by USP22 knockdown or overexpression in HCC cells (New Figure 10). Overall, these data suggest that USP22-mediated lipid accumulation most likely results from the promotion of fatty acid synthesis rather than the inhibition of fatty acid degradation as well as the promotion of fatty acid transport.

New Figure 10. USP22 did not affect fatty acid uptake in HCC cells.

3) Authors should comment that in two cell lines USP22 GOF produces distinct phenotypes which are suggestive of FAO defects upon USP22 overexpression (Fig2c vs Fig 2g). To this end, the authors should specifically analyze the carnitine-FA conjugates and demonstrate the status of FAO in MHCC-97 cell models. It would be important to address whether USP22 affects HCC line tumorigenic properties by acting on their FAO.

RE: Thanks for your suggestions. We will conduct further experiments and data analysis for FAO and were basically able to rule out the possibility that the USP22-driven fatty acid accumulation originated from its inhibition of FAO.

4) The choice of HCC lines for analyses should be explained. It is not clear why not the same lines were used for GOF and LOF experiments especially that MHCC97 lines (L vs H) are known to have different proteomic and metabolic profiles compared to non-metastatic HCC lines (PMID: 30588254, 15243804). What is the relevance of Bel7402 cell line? The analyses in SNU449 line should be commented. What are the expression levels of USP22 and how metabolic profiles compare in these different cell lines (parental and engineered)? The authors should provide at least basic characterization of the engineered cell lines (proliferation/survival/morphology in depleted cells and in cells overexpressing USP22? In the same line, given that it is embryonic kidney line, it is difficult to understand why HEK293 model with OE of USP22 was chosen for proteomics analyses (Fig3a).

RE: 1) The reason why we chose MHCC-97L and MHCC-97H for our study, as you can see in Supplementary Fig. 3b we found that among the HCC cell lines owned by our lab, MHCC-97H had the highest USP22 expression and MHCC-97L had the lowest expression. That's why we generated USP22 stably knocked down cell line in MHCC-97H and transduced USP22 in MHCC-97L for overexpression cell lines.

2). In order to exclude cell specificity, we also overexpressed USP22 in HUH7, HepG2, SNU449 and Bel-7402 and found that USP22 significantly promoted the accumulation of fatty acids and triglycerides, ACC and ACLY expression and clone formation in each cell. These results are presented in Supplementary Fig. 4 in revised version.

3). The reason of why HEK293T was selected for proteomic study, because HEK293T easily engineering by exogenous plasmid transduction to overexpress target proteins. To clarify whether the similar results in HCC cell lines, we performed the same treatment and proteomic analysis in MHCC-97H. And 6 of the top 10 proteins were present in the results of MHCC-97H proteomic study, including PPAR γ . This result provide as Supplementary table1 in revised version.

5) The data in section line 218-230 seem repetitive to the ones presented earlier in the manuscript. The analyses in HCC provide the same message as in Fig1. The info on cell lines would be more appropriate in section where the cell line work was initiated (Fig2). It is important to include analyses of PPAR γ and lipogenic enzymes alongside USP22 presented on ExtData4b.

RE: Thanks for your comment. We have rewritten the results section in the revised

version. We reexamined the expression of USP22, PPAR γ , ACC and ACLY in each cell line and found a high consistency. This result is presented in Supplementary Fig. 7b in revised version.

6) The methodology should be better described to facilitate the understanding of the work.

-The methodology of bioinformatics analyses should be elaborated indicating what datasets and how were analyzed.

-For metabolomics analyses, the details on patient selection should be provided (tumor type, patient sex, comorbidity, primary/secondary cancer...). How the adjacent tissue was controlled should be specified. The status (fasting/fed) and preferably time of the day should be indicated (lipid metabolism is circadian). The complete list of the compounds identified should be provided as a table. Given that authors comment on enrichment of specific lipid species in their metabolomics analyses (e.g. Fig.1d, 2c), they should detail the parameters of the metabolomics analyses including the information on all metabolites identified.

- In the same line, it is difficult to understand what is exactly presented as metabolomics findings e.g. on ExtData 2b,c and Fig2b. Authors should detail why panels contain different metabolites (e.g. Carn-C3:0 or FA C15) and how normalization was performed? This clarifying information should be introduced in the result section (specify the untargeted or targeted analyses, compound identification, data normalization and bioinformatic analyses).

RE: Thanks for your comments.

1). We have added the following information to the methods section in the new version:
1. The methods pathway enrichment analysis for transcriptome data, and metabolomics data. 2. *In vitro* deubiquitination assay. 3. *In vitro* pulldown assay. 4. TG assay. 5. PPAR γ transcriptional activity analysis. 6. CHIP assay. 7. Nuclear and cytoplasmic protein extraction. 8. Metabolite characterization methods.

2). We detail the source dataset in the description of the results and in the Figure legends.

3). In our first part, metabolomic was performed using surgically resected HCC and normal adjacent tissues, and patients were fasted with water before surgery, as described: LC-MS-based nontargeted metabolomic analysis detecting differential metabolites between the tumor tissues and normal adjacent tissues (More than 2 cm from the edge of the tumor) of patients with HCC (n=10, 6 female and 4 male patients, the age range is between 48 and 60). All patients were diagnosed with HCC by postoperative pathology and were free of other cancers and chronic diseases.

4). We give all the metabolite information obtained from the metabolomic analysis of 10 cancer and normal adjacent tissues, MHCC-97H-USP22-sh and MHCC-97L-USP22 cells in the raw data.

5). We had a mistake on the previous in ExtData 2b, now we corrected it in the new version. In addition, for the data analysis of non-targeted metabolomics, we normalized the data by total peak area. For targeted FA detection, we normalized the data using protein dry weight and internal standard. For metabolic data of tissues, we performed statistical analysis using paired nonparametric tests, and for metabolomics data of cells,

we performed statistical analysis by t-test. In addition, the differential metabolites were identified based on their retention time, accurate mass, and spectrometric fragments as well as available standard compounds.

7) For analyses in cells, what are the differences in the experimental conditions as USP22 overexpression in one case increases and in other case has no impact on PPAR γ levels (Fig3b, 3c, Fig 6a, 6b ExtDat3b, 3i vs Fig3f, 3g).

RE: Thanks for your comments.

1). The previous version of Fig3b and 3c corresponds to the revised version of Figure4e, which is an exogenous pulldown assay, mainly in HEK293T cells transfected with exogenous USP22 and PPAR γ , and pulldown using magnetic beads, with the aim of detecting whether they interact with each other.

2). The previous version of ExtDat3b detected whether USP22 interacted with PPAR γ by using antibodies to PPAR γ after transfection of SFB-USP22 in HEK293T and enrichment by S-beads.

3). The previous version of Fig3i was an exogenous deubiquitination experiment performed in HEK293T, where we would add the proteasome inhibitor MG132 for 6h prior to sample collection, thereby inhibiting the ubiquitinating degradation of PPAR γ , with the aim of observing whether USP22 has a deubiquitinating effect on PPAR γ .

4). Previous versions of Fig3f and 3g are protein stability experiments where CHX (an inhibitor of RNA translation) is added to the cells for a defined period of time, thus observing the half-life of PPAR γ degradation.

5). The previous version of Fig6a and 6b shows the protein expression of PPAR γ in USP22 stable-transformed cell lines.

8) In Fig3, it is difficult to understand which tag corresponds to which overexpressed protein in analyses of complex/ubiquitination. It would be advisable to include this info in figure labeling. In the same section, the additional controls should be included. Panels Fig3h-j need control of non-specific binding of Ub-HA to beads as modulation of USP22 is expected to modify the poly-Ub. Also, authors should control the levels of Ub-HA in total extracts in all experimental conditions to demonstrate that the overexpression was similar.

RE: Thanks for your comments.

1). In the revised version of the image, we included the names of the label-related proteins in the figure.

2). We re-performed all deubiquitination experiments and added an expression assay for HA-Ub in total extracts in all experimental conditions in **Fig 5 and Supplementary Fig. 6**.

3). To exclude non-specific binding of beads to HA-Ub, we transfected HA-Ub alone in HEK293T cells and performed enrichment assay by using FLAG beads, and the results confirmed that the bind did not show non-specificity (**New Figure 11**).

New Figure 11. Testing of HA-Ub non-specificity by pulldown assay.

9) In Fig 3e authors demonstrate the complex between endogenous PPAR γ and USP22, how they explain the difference in migration of USP22 and PPAR γ in input and in IP samples?

RE: We re-examined the endogenous IP experiments and this phenomenon of band migration did not occur again. We revised these results in the revised version. Please see Fig 4c.

10) In Fig.4, the presentation of transcriptomic analyses should be improved. It is not clear how RT-qPCR analyses were conducted in Fig4c (and Fig6c,d) as there is no group assigned as 1. Neither it is clear how the bulk data were analyzed and how different is the expression of these 30-top gene hits presented on heat map.

RE: 1). Thank you for your suggestions, it was an oversight in our data processing, and the data after the correction of the internal calibration has been directly graphed. **2).** We redraw the heat map of the transcriptome data in Fig.3a. We selected the top 30 differential mRNAs with high or low expression in the corrected FDR ranking compared to the control group, and red and blue represent the fold increase or decrease in mRNA expression compared to the control group, respectively.

11) In Fig. 4f, the examples of correlative IHC analyses of USP22 and ACC/ACLY are not clear. From the captures presented, the samples of USP22 labelled as low or high expression seem inverted? Authors should provide the information on steatosis status and PPAR γ expression in these tumor samples.

RE: 1). We have made the correction in the revised version. The new version of the data is presented in Fig. 6g. **2).** We performed a chi-square test using the number of HCC patients with steatosis in the TMA versus the number of patients with positive expression of USP22, PPAR γ , ACC or ACLY, respectively, and the results are presented in Fig. 6i.

12) In ExtData5 and Fig.5, the authors show that knockdown or overexpression of USP22 affect cell proliferation. Does it also impact apoptosis? It would be important to characterize the status of PPAR γ in xenografts (expression, ubiquitination, transcriptional activity). Since the mechanism is transcriptional control by PPAR γ , the transcript levels of ACC and ACLY should be analyzed as well.

RE: Thanks for your comments.

1). We examined the expression of cleaved-PARP and cleaved-caspase3 in HCC cells with knockdown or overexpression of USP22, and found that the USP22 did not affect the expression of cleaved-PARP and cleaved-caspase3 (New Figure 12).

New Figure 12. USP22 did not affect the expression of cleaved-PARP and cleaved-caspase3.

2). We examined the protein expression of PPAR γ and its ubiquitination level, as well as the protein expression of ACC and ACLY in MHCC-97H-xenograft tumor tissues. The results revealed that the protein expression of ACC, ACLY and PPAR γ was decreased in tumors formed by USP22 shRNA-expressing compared to tumors formed by control shRNA-expressing MHCC-97H cells, while the ubiquitination level of PPAR γ was significantly upregulated. We also investigated the transcript levels of ACC and ACLY by qPCR in USP22 or PPAR γ engineered HCC cells. **The results are presented in revised version as Fig. 3c; Fig. 6e; Supplementary Fig. 5b; Supplementary Fig. 6d and Supplementary Fig. 7c.**

13) The text of the manuscript requires further editing. For example, it would be advisable to improve clarity of the formulations in lines 31-34, 52-54, 331-332, 332-334 which are difficult to follow.

RE: Thanks for your comments. We have edited entire of manuscript. Please see revised version.

14) Authors should revise the reference list.

-There are double references e.g. 5=11, 31=43.

-Line 78-80, used citations give impression all related to HCC which is not the case for

Ref 19 and 8.

-Relevance of ref 30 in the context of line 92-95 is not clear.

-The reference to the statement on line 320 should be added. According to the WHO, HCC is in top five cancer related deaths.

-Not accurate statements, as in line 92-93. There is a wealth of literature on the anti-tumorigenic as well as pro-tumorigenic role of PPAR γ depending on the cancer type and genetic context. Authors should make an effort to introduce it pointing to the gaps of knowledge on the upstream control of PPAR γ expression and activity in relevance to their work.

RE: Thanks for your comments. We have corrected all mistakes and improved the writing. Please see revised version.

15) Line 93-95, the authors should cite earlier report on Akt2 driven induction of PPAR γ in the context of steatosis associated liver cancer (PMID 22334075) in addition to PMID 28394260. In general, authors should mention in introduction the complexity of HCC disease (different groups with distinct transcriptional signatures as established by works of Zucman-Rossi lab and others). Furthermore, authors should overview the metabolomics findings in HCC and highlight the advancements that their study brings.

RE: Thanks for your comments. We have cited all relevant references and improved the writing. Please see revised version.

Reviewer #2 (Remarks to the Author); expert on ubiquitin system:
Revision NCOMMS-20-38410A-Z

In the present manuscript, Ning and collaborators identify the ubiquitin-specific protease 22 (USP22) as a key player of de novo fatty acid synthesis, involved in the regulation of the PPAR γ -ACLY/ACC axis in HCC tumorigenesis. In particular, they demonstrate that USP22 interacts with and promotes PPAR γ stabilization through K48-linked deubiquitination, thus inducing and increasing transcription of the lipid synthesis enzymes ACC and ACLY. Interestingly, the authors find that USP22 expression is positively correlated with PPAR γ expression in HCC, and simultaneously, high expression of USP22 and PPAR γ or USP22, ACC and ACLY contributes to tumor progression and it is associated with poor prognosis.

HCC is one of the most frequent solid tumors worldwide, characterized by clinical aggressiveness, resistance to conventional chemotherapy, and high lethality. For these reasons, there is an urgent need to better define the molecular alterations involved in the pathogenesis of HCC to develop new innovative therapeutic strategies. Albeit USP22 has already been described as promising therapeutic target in HCC (Ling S, et al. Gut 2020;69:1322–1334), the new role of the USP22-PPAR γ -ACLY/ACC axis on the regulation of the lipogenesis and its implication in HCC tumorigenesis, provide a new therapeutic approach targeting fatty acid synthesis for the treatment of this tumor.

The manuscript is well structured and the data shown are interesting and well presented. I still have a few concerns about the conclusions made from the obtained results. In particular, co-localization, protein stability assays and ubiquitylation experiments were performed using overexpressed proteins only and not in HCC cells, thus dampening my enthusiasm.

RE: Thanks for reviewer's important suggestions and comments, and agreed that our work is interesting and well presented. We have carefully respond the below issues and fixed them in the revised manuscript. Please refer to the following replies.

Comments

Figure 3d: the co-localization between USP22 and PPAR γ should be performed for endogenous proteins.

RE: Thanks for your comments. We performed endogenous immunofluorescence experiments in MHCC-97H and MHCC-97L cell lines, and the results confirmed that both USP22 and PPAR γ were mainly localized in the nucleus. We provided this result in New Figure 13 (Fig. 4f in revised version).

New Figure 13. Localization of USP22 and PPAR γ in HCC cells. f) Triple immunofluorescence (IF) staining for USP22 (red), PPAR γ (green), and nuclei (DAPI, blue) was performed in MHCC-97L and MHCC-97H cells. Scale bars, 10 μ m. (Fig. 4f in revised version).

Figure 3e: There is an upward shift of the protein bands shown in the IP lines compared to the corresponding input lines. This point needs to be explained.

RE: Thanks for your comments. We re-examined the endogenous IP experiments and this phenomenon of band migration did not occur again. We revised these results in the revised version. We provided this result in New Figure 14 (Fig. 4c in revised version).

New Figure 14. Endogenous IP experiments of USP22 and PPAR γ in HCC cells. Cell lysates of MHCC-97L, HUH7, HepG2, SNU-449 and Bel-7402 cells were immunoprecipitated with IgG or USP22 antibodies, and immunoblot assays were performed using USP22 or PPAR γ antibodies. (Fig. 4c in revised version).

Figure 3f: the ability of USP22 to stabilize PPAR γ should be investigated by CHX experiments analyzing endogenous levels of PPAR γ in USP22 stably knocked down in MHCC-97H and/or USP22 transduced overexpression in MHCC-97L.

RE: Thanks for your comments. We re-examined the CHX experiments in MHCC-97H-sh1/2 (MHCC-97H cells transduced with two independent USP22 shRNAs) and MHCC-97L-USP22 (MHCC-97L cells stably overexpressed USP22) cell lines, and the results clarified the protein-stabilizing effect of USP22 on PPAR γ . We provided this result in New Figure 15 (Fig. 5i, j in revised version).

New Figure 15. USP22 stabilizes PPAR γ in HCC cells. Cell Stability analysis of PPAR γ protein in MHCC-97H-shUSP22-1/2 cells (i), MHCC-97L-USP22, MHCC-97L-USP22 C185S cells (j) and treated with 40 μ M cycloheximide (CHX) for indicated times. (Fig. 5i, j in revised version).

Figures 3h, l and j: to appreciate the regulation of USP22 on PPAR γ ubiquitylation, endogenous levels of ubiquitylated PPAR γ should be evaluated in the presence of ectopic expression of USP22 or USP22 C1856S, and in genetic depletion of USP22 in HCC cell lines.

Response: In the revised version, endogenous levels of ubiquitylated PPAR γ had been evaluated in the presence of ectopic expression of USP22 or USP22 C1856S, and in genetic depletion of USP22 in HCC cell lines. We provided this result in New Figure 16 (Fig. 5a, b in revised version).

New Figure 16. USP22 deubiquitinates PPAR γ in HCC cells. Ubiquitination assay of PPAR γ in MHCC-97H-shUSP22-1/2 cells (a) or MHCC-97L-USP22, MHCC-97L-USP22 C185S cells (b) treated for 6h with 10 μ M MG132. (Fig. 5a, b in revised version).

It is my opinion that the study could increase the novelty by investigating the potential role of USP22 in pVHL-mediated ubiquitylation of PPAR γ (Metabolism 2020 Sep;110:154302. doi: 10.1016/j.metabol.2020.154302).

RE: Thanks for your comments. To identify the deubiquitination site by USP22 targeting on PPAR γ , we performed several studies as followed. First, based on literatures and reviewer's suggestions, we confirmed the E3 ubiquitin ligase of pVHL and CUL4B regulated lysine sites. We found that USP22 significantly decreased the pVHL and CUL4B involved ubiquitination, it's indicated that USP22 regulates deubiquitination of PPAR γ through other lysine sites. Next, to identify which lysine sites on the PPAR γ were involved in USP22 deubiquitination process, we generated the several fragments of PPAR γ based on different protein domain, and found that the domains which containing DBD domain could deubiquitinated by USP22. Consequently, we mutated all 7 lysine sites on the DBD domain, and found the Lys-169 may important site for USP22 deubiquitination of PPAR γ . We provided this result in New Figure 17 (Fig. 5d, 5e, 5f, 5g, 5h and Supplementary Fig. 6e and 6f in revised version).

New Figure 17 on the next page.

New Figure 17. USP22 deubiquitinates and stabilizes PPAR γ . **a)** Ubiquitination assay of PPAR γ in HEK293T cells cotransfected with HA-Ub, Flag-PPAR γ , Flag-PPAR γ -K404/434R, Myc-USP22 or V5-pVHL and treated with 10 μ M MG132 for 6 h. (Fig. 5d in revised version). **b)** Ubiquitination assay of PPAR γ in HEK293T cells cotransfected with HA-Ub, Flag-PPAR γ , Flag-PPAR γ -K240/265R, Myc-USP22 or V5-CUL4B and treated with 10 μ M MG132 for 6 h. (Fig. 5e in revised version). **c)** Ubiquitination assay of FL (full length), AD (AF-1-DBD), DH (DBD-Hinge), D (DBD), DHL (DBD-Hinge-LBD) and HL (Hinge-LBD) domain of PPAR γ in HEK293T cells cotransfected with HA-Ub and Myc-USP22, and treated with 10 μ M MG132 for 6 h. (Fig. 5f in revised version). **d)** Ubiquitination assay of DBD domain of PPAR γ in HEK293T cells cotransfected with HA-Ub, Myc-USP22, Flag-DBD, Flag-DBD-K117R, Flag-DBD-K132R, Flag-DBD-K142R, Flag-DBD-K161R and Flag-DBD-K169R and treated with 10 μ M MG132 for 6 h. (Fig. 5g in revised version). **e)** Ubiquitination assay of PPAR γ -K156/157R in HEK293T cells cotransfected with HA-

Ub, SF-PPAR γ -K156/157R, Myc-USP22 and treated with 10 μ M MG132 for 6 h. (Supplementary Fig. 6f in revised version). f) Ubiquitination assay of PPAR γ in HEK293T cells cotransfected with HA-Ub, Myc-USP22, SF-PPAR γ and SF-PPAR γ -5KR (K169/240/265/404/434R) and treated with 10 μ M MG132 for 6 h. (Fig. 5h in revised version). g) Cell lysates of MHCC-97H, MHCC-97L and HUH7 were immunoprecipitated with IgG or USP22 antibodies, and immunoblotted with antibodies against USP22, CUL4B and pVHL. (Supplementary Fig. 6e in revised version).

In light of authors' conclusions, should be important to analyze the endogenous levels of PPAR γ , ACC and ACLY in Figure 1f.

RE: Thanks for your comments. Immunoblotting of these 10 paired HCC and adjacent normal tissues showed that the high expression of ACC (5/10), ACLY (7/10) and PPAR γ (9/10) in cancer tissues.own in the figure below. The high expression of PPAR γ significantly associated USP22 expression in cancer tissues (New Figure18).

New Figure18. PPAR γ expression correlated with ACC and ACLY expression. Western blot analysis of PPAR γ , ACC and ACLY expression in 10 paired HCC and adjacent normal tissues

In Figure 5h, the levels of PPAR γ should be showed.

RE: Thanks for your comments. We examined the protein expression of PPAR γ and its ubiquitination level, as well as the protein expression of ACC and ACLY in MHCC-97H-xenograft tumor tissues. The results revealed that the protein expression of ACC, ACLY and PPAR γ was decreased in tumors formed by USP22 shRNA-expressing compared to tumors formed by control shRNA-expressing MHCC-97H cells, while the ubiquitination level of PPAR γ was significantly upregulated. The results are presented in Supplementary Fig. 6d and Supplementary Fig. 7c. Please also see Reviewer1's Major question 6.

For in vivo studies, the authors use in vivo xenografts (flank tumor) from HCC cell lines. An orthotopic HCC model would have been more relevant for supporting authors conclusions.

RE: Thank you for your suggestion. However, due to the COVID-19 pandemic our

animal experiments were suspended indefinitely. In our future work, we intend to construct conditional knockout mice of USP22 for further study.

The role of USP22 in the regulation of HIF1 α in HCC deserves to be discussed.

RE: Thanks for your comments. We have cited all relevant references and discussed and improved the writing. Please see revised version.

MW should be added in each western blot.

Response: Thank you for your suggestion. As reviewer's suggestion we gave the molecular weight of each band in all images in the raw western blot data.

Reviewer #3 (Remarks to the Author); expert on metabolism, metabolomics and transcriptomics:

The authors investigated the molecular mechanism about the critical role of elevated de novo lipogenesis (DNL) in hepatocellular carcinoma (HCC) development. They performed large number of computational and experimental analysis and identified ubiquitin-specific protease 22 (USP22) as a key regulator for de novo fatty acid synthesis. They found that USP22 directly interacts with, deubiquitinates and stabilizes PPAR γ through K48-linked deubiquitination, and in turn, this stabilization increases ACC and ACLY transcription.

In addition, they found that USP22 promoted the de novo synthesis of fatty acid-based on labelling studies. They also reported that USP22 expression positively correlates with PPAR γ expression, and simultaneously, high expression of USP22 and PPAR γ or USP22, ACC and ACLY is associated with a poor prognosis in HCC.

I agree with the authors that, they identified a previously undescribed USP22-regulated lipogenesis molecular mechanism that involves the PPAR γ -ACLY/ACC axis in HCC tumorigenesis. However, I do not agree that the mechanism they suggested in their study is supported by the experiments they performed. I have serious concerns related to the interpretation of the results. It would be useful to clarify the points below before the publication of the paper.

RE: Thanks for reviewer's important suggestions and comments, and agreed that our work is a previously undescribed USP22-regulated lipogenesis molecular mechanism that involves the PPAR γ -ACLY/ACC axis in HCC tumorigenesis. We have carefully respond the below issues and fixed them in the revised manuscript. Please refer to the following replies.

1) The authors reported that USP22 was highly expressed in HCC and cholangiocarcinoma but not in other cancer types. Based on TCGA database and Human Pathology Atlas, this statement is not correct. It is actually the other way around. <https://www.proteinatlas.org/ENSG00000124422-USP22/pathology>

RE: Thank you for your comments. The data you mentioned we exported from the website (<http://gepia2.cancer-pku.cn/#index>) and in revised version we exported all the tumor data and presented them in **Supplementary Fig. 2**. The results do suggested that USP22 was highly expressed in HCC and cholangiocarcinoma but not in other cancer types. In additionally we had chance to take look the article "USP22 Deubiquitinates CD274 to Suppress Anticancer Immunity" (31399419) , they also mentioned similar findings in their study.

2) Subsequently, pathway enrichment analysis based on the TCGA database revealed that USP22 was significantly negatively associated with FA degradation in HCC

(Extended Data Fig. 1D). Their analysis indicated that the differences in the amount of fatty acids may be due to the alterations in FA oxidation rather than DNL. This may be one of the reasons why the authors could not verify the changes in the expression of FASN and SCD. In that case, the entire paper has to be rewritten considering this alternative potential mechanism.

RE: Thank you for your comments. Firstly, we re-performed the GSEA pathway enrichment analysis based on TCGA HCC data, and the results are shown in **Supplementary Figure 1c**. In this result, we found that the synthesis of unsaturated fatty acids was ranked second place, but the degradation of fatty acids was ranked seventh place. To further clarify whether USP22 is involved in lipid accumulation in HCC cells by inhibiting fatty acid oxidation, we observed the metabolomic data of MHCC-97H-shUSP22 and MHCC-97L-USP22 cells, and found that the acylcarnitine upregulated in both USP22 knockdown and overexpression cells, which might suggest that the acylcarnitine alteration may be not affected by USP22 expression. In addition, we performed GSEA pathway enrichment analysis of MHCC-97H-shUSP22-1/2 transcriptome data and found that USP22 was only highly associated with fatty acid synthesis, not with fatty acid degradation. Then we examined the expression of fatty acid degradation-related enzymes (CPT1A, CPT2, ACOX1, ACADL and ECSH1) in USP22 knockdown and overexpression cell lines, and found that these enzymes were not altered in USP22 knockdown or overexpression cells. CPT is the key enzyme for process of fatty acid degradation, therefore, we further analyzed the correlation between *USP22* and *CPT* in TCGA database. But the *USP22* expression did not correlate with both *CPT1A* and *CPT2* expression, respectively. **These results are presented in revised version Supplementary Figure 1c; Supplementary Figure 3d, 3e; Supplementary Figure 5a, 5c, 5d, 5e, 5i.**

3) It is known that the expression of FASN and SCD is increased or decreased based on the activity of DNL. It is unexpected to see that the expression of the FASN and SCD is not changed in all experiments that the authors performed.

RE: Thank you for your question. Although FASN and SCD are strongly associated with lipid synthesis, indeed, USP22 may be not involved regulation of these two genes at least in HCC cells. We found that among the 4 enzymes of fatty acid synthesis genes (ACLY/ACC/FASN/SCD), USP22 only effect the expression of ACC and ACLY in HCC cells entire our study, and in both TCGA HCC database and two independent HCC TMA assays (LV1021, HLivH180Su11), and confirmed that USP22 significantly positive correlate with ACC and ACLY expression. In addition, we also found that USP22 could regulates the PPAR γ -mediated transcription of ACLY and ACC. We speculate that ACC and ACLY are more dependent on PPAR γ regulation during USP22-mediated fatty acid synthesis, but FASN and SCD may be regulated by other lipogenic transcription factors such as SREBF1 and PPAR α .

4) If the authors focus on the DNL, the authors should measure TAG levels in the cell and animal experiments. It is really surprising that the authors have not measured TAG

in any of the experiments performed.

RE: Thank you for your suggestion. Previously we focused on the process of fatty acid de novo synthesis. In revised version, we measured the triglyceride content in all the stably transformed cell lines mentioned in the manuscript by oil red staining and TG assay kit.

5) In Figure 2 about the cell experiments, how the cell growth is affected? Is it normalized with cell growth? As I mentioned above, the authors should measure TAG levels in Figure 2b.

RE: Thank you for your question. Knockdown of USP22 expression can inhibit HCC cell growth, and to avoid data discrepancies due to cell number, we used total peak area or dried protein weight and internal standard control for correction of metabolomics data. In revised version, we measured the triglyceride content in all the stably transformed cell lines mentioned in the manuscript by oil red staining and TG assay kit. These results are presented in revised version Fig. 2e, f; Fig. 3g, h; Fig. 7c-f; Supplementary Fig. 4d, e.

6) Based on Figure 3, this reviewer was not surprised to see that there is a link between USP22, PPARG and HADH which is involved in fatty acid oxidation. It is interesting that none of the genes on the list is involved in DNL. As I mentioned above, I strongly suggest the authors investigate the specific role of USP22 in lipid metabolism specifically in fatty acid oxidation.

RE: Thank you for your question. To verify USP22 whether definitively binds to HADHA (we think the reviewer mistyped the gene name, HADH should be refer to HADHA in our prey list), endogenous IP assay was conducted by both USP22 and HADHA antibodies MHCC97H cells. We found that there is no interaction between USP22 and HADH in at least MHCC-97H HCC cells (New Figure 19).

New Figure 19. Endogenous IP experiments of USP22 and HADHA in HCC cells. Cell lysates of MHCC-97H cells were immunoprecipitated with IgG or USP22 or HADH antibodies, and immunoblot assays were performed using USP22 or HADH antibodies.

7) Based on Figure 4, the authors should investigate the role of GCLM, SLC7A1, NNMT and other redox related genes. I think these genes have a critical role in alterations of the lipids specifically in fatty acid oxidation.

RE: Thank you for your comments. As you mentioned, the genes are involved in glutathione synthesis, such as GCLM and SLC7A11, were downregulated in USP22 knock-downed cells, and we also found that the knockdown of USP22 upregulated glutathione levels. These suggest that the possibility of USP22 may influence redox reactions to regulate tumor progression through glutathione synthesis. This would be a good direction to further investigation.

However, in the present study, we mainly focused on USP22 promoting ACC and ACLY expression through stabilizing PPAR γ , which in turn promotes lipid synthesis and tumorigenesis in HCC cells. In addition, we performed GSEA pathway enrichment analysis of MHCC-97H-shUSP22-1/2 transcriptome data and found that USP22 was only associated with fatty acid synthesis and not with fatty acid degradation. We then examined the expression of fatty acid degradation-related enzymes (CPT1A, CPT2, ACOX1, ACADL and ECSH1) in USP22 knockdown and overexpression cell lines, and found that these enzymes were unchanged after USP22 knockdown or overexpression. Additionally, in order to clarify whether USP22 affects fatty acid uptake, we performed the following experiments. By adding C¹³-labeled palmitic acid to the medium of SNU449-Control and USP22 overexpressed cells, the intracellular C¹³-labeled palmitic acid content was detected after 30 min, and we found that USP22 overexpression failed to affect the intracellular C¹³-labeled palmitic acid. In addition, we found that the expression of CD36 and FABP were not altered by USP22 knockdown or overexpression in HCC cells (New Figure 10). Overall, these data suggest that USP22-mediated lipid accumulation most likely results from the promotion of fatty acid synthesis rather than the inhibition of fatty acid degradation as well as the promotion of fatty acid transport. Please see New Figure 10.

In this study, we also examined the expression of fatty acid oxidation and transport-related enzymes, to clarify that USP22-mediated lipid accumulation most likely results from the promotion of fatty acid synthesis rather than the inhibition of fatty acid degradation as well as the promotion of fatty acid transport.

8) In Figure 5i, the authors measured the lipid structures. It is surprising that the abundant FA structures (C16:0, C118:0) are not on the list. C18:1 is also showing an opposite trend. I strongly recommend the authors to measure the total TAG levels as well.

RE: Thank you for your comments. In the previous version the Figure 5i indicated metabolomics assay performed on xenograft tumor tissues. Due to the complexity of the *in vivo* environment, the results may slightly different from the *in vitro* cell experiments, however the majority of lipid molecules were downregulated in the USP22 knockdown group compared to the control group. In addition, we tried detect triglycerides by using frozen sections for oil red staining or by using the TG kit using originally tumor tissues from -80 refrigerator (past almost 2 years), unfortunately, we

did not get results due to the long storage time period, and due to the COVID-19 pandemic the animal experiments have been suspended indefinitely.

9) In Figure 51, It also interesting to see that Glutathione levels are significantly increased and this supports the changes in the expression of the genes in Figure 4. The carnitine levels are significantly increased as well.

RE: Thank you for your comments. Indeed, elevated glutathione was found in our MHCC-97H-shUSP22 lines or tumors formed from this cell lines. This might be related to the increase in cellular reactive oxygen species after knockdown. Since this subject lies beyond the scope of our current investigation, to clarify the exact mechanism, we will conduct further studies in subsequent experiments.

10) In Figure 6 i and j, both control and shUSP22 had 500 Total Volume. On the other hand, shACC, shACLY and shPPARG had around 100 Total Volume. I suggest the authors add an explanation for these differences.

RE: Thank you for your comments. The tumors in Figure 6i were derived from MHCC-97H cells, while Figure 6j were derived from MHCC-97L cells. These two cell lines have different in tumor growth rate. Tumors bearing with MHCC-97H cells growing fast, conversely tumors bearing MHCC-97L cells will growing slowly. We can see that the tumors formed with MHCC-97H cells bigger than tumors formed from MHCC-97L in in vivo experiments. **These results are presented in revised version as Fig. 2g, h, i; Fig. 7g-l; Supplementary Fig. 4g-i.**

11) USP22 have been reported as a prognostic gene in the human pathology atlas. I suggest the authors cite Uhlen et al, 2017 Science paper to support their analysis.

RE: Thanks for your comments. We have cited all relevant references and discussed and improved the writing. Please see revised version.

REVIEWER COMMENTS

Reviewer #1 (Remarks to the Author):

I would like to thank authors for providing insightful data in this revised version of the manuscript. Their findings shed more light on the functional interaction between USP22 and PPAR γ for lipid anabolism in HCC cells. Moreover, they perform additional analyses to further demonstrate role of USP22 as a deubiquitinase of PPAR γ and identify the sites on PPAR γ that they propose are targeted by USP22. However, their additional analyses on the characterization of PPAR γ transcriptional activity and chromatin binding in cells with USP22 OE or KD are not sufficient to support a major claim of their study. The evidence presented in the paper do not show that de-ubiquitination of specific Lys residues by USP22 in PPAR γ is required to fire up transcription of selective metabolic enzymes in FA synthesis. This revised manuscript also did not provide a larger context of how the USP22 driven de-ubiquitination of PPAR γ would contribute to liver cancer associated with NAFLD/NASH/MAFLD and how it is related to different transcriptional groups that are known in HCC. Finally, although the flow of the story is improved, the manuscript is still difficult to read because of long phrasing and the use of unconcrete language to which the title is an example: "USP22-dependent accumulation of lipidomes drives hepatocellular carcinoma progression by stabilizing PPAR γ ". It is also an example of an overstatement as authors did not model the progression of HCC in their study. Altogether, these shortcomings dampen my enthusiasm about the work which would need further experimenting to support the main conclusion.

Specific comments regarding the revised version of the manuscript:

- The choice of cell models for the study should be explained in the main text of the manuscript. Again, it is not clear in this revised text version why the GOF and LOF experiments of USP22 were conducted in different cell lines (line 148-151). Judging from the USP22 protein levels that authors show in Suppl.Fig.3b and Suppl.Fig.7b, there is not much difference between MHCC-97L and H lines.
- The description of the metabolomics analyses both in HCC samples and in cell lines is insufficient. The information if it is targeted or untargeted analysis should be included. If it is targeted analyses than the enrichment is questionable as it will depend on the database used. If it is untargeted than it would be important to list all metabolites detected to show if there is a bias towards identification of polar/non-polar metabolites.
- In Fig3a, unbiased DAVID-GO pathway bioinformatic analyses should be performed on RNA-Seq to show what are the pathways controlled by USP22 in MHCC-97H cells.
- In Fig3f and 3g, how authors explain that knockdown of ACC doesn't impact TG and FA16:00 and FA 18:0 content. It is not clear what these measurements represent as "Relative expression of FA16:0 and FA18:0".
- Line 223, the transcript of CPT1 does show significant inverse correlation with USP22 ($r=-0.153$) contrary to how the authors describe it in the text.
- Deubiquitination of PPAR γ presented in Fig5c shows decrease of PPAR γ -Ub signal in presence of His-USP22, but the assay lacks control of untreated PPAR γ .
- In analyses of K-R mutants of PPAR γ the cut-off of the effect is not clear. The Fig.5h lacks the controls of samples without expression of USP22. For example, Fig.5g shows lack of effect of Myc-USP22 overexpression on ubiquitination of K169R mutant and there is equally no effect on poly-ubiquitination of K156/157R mutant (Sup. Fig.6f). Given these findings one would prefer to combine mutations of K156/157/169R to establish if these sites are targeted by USP22.
- The authors did not link the de-ubiquitination of PPAR γ on specific residue with the expression of ACC and ACLY for FA synthesis and subsequential proliferation or survival of HCC cells. The ChIP-Seq or ChIP-qPCR with PPAR γ K/R mutants and the proliferation or xenograph assays would be possible approaches to reinforce this claim.
- The authors should comment in the text on the differences in the half-life of endogenous PPAR γ (Fig.5i and Fig.5j) in two stable cell lines. The WB panels of vinculin are overexposed for the quantification and should be substituted with less exposed panels.
- Do authors suggest that de-ubiquitination impacts nuclear retention of PPAR γ (Fig.6a)? If it is what the authors wanted to stress, then they also should test nuclear/cyto distribution of PPAR γ K/R mutants (K156/157/169R). Moreover, it is surprising to find in this analysis (Fig.6a) such a pronounced effect on PPAR γ protein levels given a depletion of USP22 (less than 30% decrease in

expression). Moreover, these fractionation analyses of two cell lines show that PPAR γ is differentially distributed between Cyto and Nuc with MHCC-97H showing practically exclusive nuclear localization of PPAR γ . It is an intriguing finding especially that IF analyses on Fig4f did not show much difference between two cell lines. How authors explain it?

-The conclusion that "USP22 significantly modulated PPAR γ transcriptional activity by DNA binding" is misleading (lines294-295). The Fig.6b is an ELISA based assay that assess if there is PPAR γ protein in the extract. The luciferase assay in combination with the CHIP would be more relevant approach to address this question.

-The PPRE that the authors identified doesn't align with the classic PPRE sequence (Fig.6c). The evidence from the CHIP-Seq (published or in-house) should be provided showing the enrichment pick of PPAR γ in the gene regions of ACLY and ACACA. Also, the CHIP-qPCR assay and specifically the quantification that was used needs clarification (Fig.6d). What enrichment did authors observed in PPAR γ immunoprecipitates compared to negative controls (IgG beads)?

-The evidence that authors show to rule out Akt signaling upstream or downstream of USP22 in PPAR γ stabilization and its transcriptional activation are insufficient. It is not evident why authors conducted experiment in this set-up (serum deprivation for 24hours). Already the findings showed in Sup. Fig.7d suggest that USP22 might act upstream of pAkt (not clear what is the site that authors analyzed) which is in line with a recent report (34226501). It would be advisable to discuss at least in the discussion possible functional crosstalks between USP22, PPAR γ and insulin signaling in HCC that emanate from this study.

- It is not clear how authors accessed steatosis in tissue arrays thus the correlation analyses with other histological stainings are difficult to evaluate (Fig.6i)

-The cell density in the cultures used for TG-Oil red analyses is different (Fig.2e). The lipid synthesis could be impacted by the cell number and also the Oil Red staining technically might influence it (especially washing-out steps). It would be advisable to do these analyses in cultures of the same density and to use the fluorescent dye like Bodipy for evaluation.

- The correlative analyses of PPAR γ /USP22/lipogenic enzyme transcripts with survival are re-grouped and are easier to comprehend. However, there is no analysis of the triple correlation, thus it is not clear if it holds the route. It would be important (suggested in initial review) to analyze how these findings relate to known transcriptional groups in HCC (e.g. in 17187432).

-Regarding the PPAR γ isoform analyses, although authors are right that, in physiology, the expression of PPAR γ 2 form is largely restricted to adipose tissue, it is also inducible form of PPAR γ that is found upregulated in cancer cells. As a minimum the authors have to clearly indicate in the text that they focussed exclusively on PPAR γ 1 form.

-The analyses of apoptosis (New Figure 12 that is not included in the manuscript) are insufficient to rule out the impact of USP22 OE or KD on cell survival and apoptosis. It is surprising that H line shows higher level of cleaved PARP compared to L line, the authors should comment on it in the section when they chose to use these two cell lines for their study. The conditions under which this experiment was conducted are not clear. The immunoblot analyses should be complemented with Tunnel assay in xenographs samples. Given the claim of potential therapeutic effect of USP22 inhibition in HCC, the authors should provide some insights to the mechanisms of this cancer growth inhibition.

Minor comments:

-The exogenous pull-down is a confusing term (line 240) and from provided information the assay is the co-immunoprecipitation with transiently overexpressed proteins.

-When authors describe polyubiquitination of PPAR γ in USP22 overexpressing cells, they mention on several occasions "inhibition of polyubiquitination" (e.g. line 254, line 276-277). This is confusing as it suggests USP22 acts upstream of PPAR γ ubiquitination (not addressed in this study).

-Figure legends need editing to include sufficient information on the methodology and conditions used as well as statistical analyses that were conducted. Example of such scarcely described legend is Suppl. Fig.2.

-The text needs editing to harmonize the use of plurals, prefer to use specific verbs instead of general (e.g. regulation, contribute, control, mediate). Example of the phrases that need editing to clarify the message is line 322-323 or line 327-329. The sub-headings should be edited to convey the conclusions of the sub-chapters, ex "PPAR γ contributes to USP22 mediated ACC and ACLY expression (line 285)". The authors also should be specific in how they use the term "expression"

(transcription, translation, transcript or protein levels, sometimes it is also applied to metabolite).

-As already mentioned in the initial review, in the text of the manuscript, the conclusions often come before the evidence in support. The example is the first paragraph, line 121-123 claiming that lipid synthesis was significantly upregulated in HCC. There is also persistent issue with overstating in the conclusions e.g. line 318: "Collectively, PPAR γ as a transcription factor upregulates ACC and ACLY expression during USP22 mediated HCC progression". The HCC progression was not modeled in this study and the evidence in support that de-ubiquitination of PPAR γ by USP2 activates transcription of ACC and ACLY are weak.

-Calling of Fig.2j is missing (line189).

-On Fig.3a the heatmap shows changes from -1 to +1. If it is log₂ fold change, it has to be indicated in figure legend.

Reviewer #3 (Remarks to the Author):

The authors did excellent work during the revision of the paper. I suggest the publication of the manuscript in its current version.
to support their analysis.

REVIEWER COMMENTS

Reviewer #1 (Remarks to the Author):

I would like to thank authors for providing insightful data in this revised version of the manuscript. Their findings shed more light on the functional interaction between USP22 and PPAR γ for lipid anabolism in HCC cells. Moreover, they perform additional analyses to further demonstrate role of USP22 as a deubiquinase of PPAR γ and identify the sites on PPAR γ that they propose are targeted by USP22. However, their additional analyses on the characterization of PPAR γ transcriptional activity and chromatin binding in cells with USP22 OE or KD are not sufficient to support a major claim of their study. The evidence presented in the paper do not show that de-ubiquitination of specific Lys residues by USP22 in PPAR γ is required to fire up transcription of selective metabolic enzymes in FA synthesis. This revised manuscript also did not provide a larger context of how the USP22 driven de-ubiquitination of PPAR γ would contribute to liver cancer associated with NAFLD/NASH/MAFLD and how it is related to different transcriptional groups that are known in HCC. Finally, although the flow of the story is improved, the manuscript is still difficult to read because of long phrasing and the use of unconcrete language to which the title is an example: "USP22-dependent accumulation of lipidomes drives hepatocellular carcinoma progression by stabilizing PPAR γ ". It is also an example of an overstatement as authors did not model the progression of HCC in their study. Altogether, these shortcomings dampen my enthusiasm about the work which would need further experimenting to support the main conclusion.

RE: We are appreciated the reviewer's important recommendations and remarks. Indeed we have shown that de-ubiquitination of specific Lys residues by USP22 at PPAR γ in the last version (Fig5). And also, as reviewer's suggestion we are revised the title as "USP22 regulates lipidome accumulation by stabilizing PPAR γ in hepatocellular carcinoma" in revised version. We have carefully responded below issues and fixed them in the revised manuscript. Please refer to the following replies.

Specific comments regarding the revised version of the manuscript:

1. The choice of cell models for the study should be explained in the main text of the manuscript. Again, it is not clear in this revised text version why the GOF and LOF experiments of USP22 were conducted in different cell lines (line 148-151). Judging from the USP22 protein levels that authors show in Suppl.Fig.3b and Suppl.Fig.7b, there is not much difference between MHCC-97L and H lines.

RE: Thanks for your suggestions. First, results in Suppl. Fig. 3b and Suppl. Fig. 7b was representative image from three replicate immune-blotting experiments in HCC cell lines. We quantified the expression by Image J and observed that USP22 expression was highest in MHCC-97H cell line and lowest in MHCC-97L cell line, and USP22 relatively low expressed in SNU449, HepG2, Bel-7402 and HUH7 cells. Furthermore, to exclude the cell specificity, we chose all of the above cell lines for this study. We have also made further modifications in the revised version.

New Figure 1. Protein expression of USP22 in HCC cells and THLE-2 cells.

a) Analysis of USP22 protein expression in 10 HCC cell lines (Bel-7402, Huh7HUH7, Hep3B, MHCC-97L, SUN-449, HepG2, HB611, SMMC-7721, HCC-LM3 and MHCC-97H) and one normal liver cell line (THLE-2). (upper panel: western blots; lower panel: quantification of the western blotting image.) **b)** Quantification of USP22 protein expression in Suppl. Fig. 7b.

2. The description of the metabolomics analyses both in HCC samples and in cell lines is insufficient. The information if it is targeted or untargeted analysis should be included. If it is targeted analyses than the enrichment is questionable as it will depend on the database used. If it is untargeted than it would be important to list all metabolites detected to show if there is a bias towards identification of polar/non-polar metabolites.

RE: Thanks for your suggestions. The metabolomic analysis of the HCC tissues and cell lines involved in Fig.1 and Fig.2 were used LC-MS based non-targeted metabolomics, and the complete peak list will provide as raw data in final submission. Additionally, for the data analysis of non-targeted metabolomics, we normalized the single metabolite by total peak area. In the heat map we list the metabolites that are significantly different compared to the control cells or normal tissue. We also added the relevant descriptions in the Fig.1 and Fig.2 legends.

3. In Fig3a, unbiased DAVID-GO pathway bioinformatic analyses should be performed on RNA-Seq to show what are the pathways controlled by USP22 in MHCC-97H cells.

RE: Thanks for your suggestion. We performed GO-BP (biological process) analysis using significantly different genes ($\text{LogFC} \geq 1$, $Q < 0.05$) in the transcriptome data and found that USP22 was closely associated with the lipid synthesis pathway.

New Figure 2. GO-biological process (GO-BP) analysis based on differential genes (LogFC ≥ 1 , Q < 0.05) in transcriptome data of MHCC-97H-shUSP22-1 cells (left) and MHCC-97H-shUSP22-2 cells (right). (Supplementary Fig. 5a)

4. In Fig3f and 3g, how authors explain that knockdown of ACC doesn't impact TG and FA16:00 and FA 18:0 content. It is not clear what these measurements represent as "Relative expression of FA16:0 and FA18:0".

RE: Thanks for your questions. In deed knockdown of ACC and ACLY deceased the TG and FA16:00 and FA 18:0 content compared to Control cells, however transduction of USP22 in ACC or ACLY knockdown cells could not significantly promote intracellular fatty acid and TG accumulation. The "Relative expression of FA16:0 and FA18:0" indicates relative contents of FA16:0 and FA18:0 are corrected by using the corresponding internal standards (FA 16:0-d3 and FA 18:0-d3). We have corrected as "Relative contents of FA16:0 and FA18:0" in the revised version.

5. Line 223, the transcript of CPT1 does show significant inverse correlation with USP22 (r=-0.153) contrary to how the authors describe it in the text.

RE: Thanks for your question. We think the reviewer means CTP2 not the CTP1 in the question. Basically, the degree of correlation mainly relied on the magnitude of the absolute value of the correlation coefficient, if the absolute values less than ± 0.3 representing weak or no correlation. We revised this result as: USP22 was not correlated with *CPT1A* or weakly correlated with *CPT2* (R= -0.153, p=0.003), in the new version.

6. Deubiquitination of PPAR γ presented in Fig5c shows decrease of PPAR γ -Ub signal in presence of His-USP22, but the assay lacks control of untreated PPAR γ .

RE: Thanks for your suggestion. We re-performed the *in vitro* deubiquitination assay with the corresponding control group.

New Figure 3. *In vitro* deubiquitination assay of ubiquitinated PPAR γ protein with purified His-USP22. (Fig. 5c in revised version)

7. In analyses of K-R mutants of PPAR γ the cut-off of the effect is not clear. The Fig.5h lacks the controls of samples without expression of USP22. For example, Fig.5g shows lack of effect of Myc-USP22 overexpression on ubiquitination of K169R mutant and there is equally no effect on poly-ubiquitination of K156/157R mutant (Sup. Fig.6f). Given these findings one would prefer to combine mutations of K156/157/169R to establish if these sites are targeted by USP22.

RE: Thanks for your suggestions. We re-performed the deubiquitination assay on PPAR γ -5KR protein and confirmed that USP22 did not deubiquitinate the PPAR γ -5KR protein. In addition, the result of Sup. Fig. 6f confirm that USP22 deubiquitinates PPAR γ -K156/157R protein, which means that K156 and K157 are not USP22 targeting deubiquitination sites. Combining the results of Fig.5d, 5e and 5g, we consider that K169/240/265/404/434 on PPAR γ are the deubiquitination sites for USP22.

New Figure 4. Ubiquitination assay of PPAR- γ 5KR protein in HEK293T cells cotransfected with HA-Ub, Myc-USP22, and SF-PPAR- γ 5KR (K169/240/265/404/434R) and treated with 10 μ M MG132 for 6 h. (Fig. 5h in revised version)

8. The authors did not link the de-ubiquitination of PPAR γ on specific residue with the expression of ACC and ACLY for FA synthesis and subsequential proliferation or survival of HCC cells. The ChIP-Seq or ChIP-qPCR with PPAR γ K/R mutants and the proliferation or xenograph assays would be possible approaches to reinforce this claim.

RE: Thanks for your suggestions. We generated Flag-PPAR γ -5KR or Flag-PPAR γ transduced cell lines with simultaneously knockdown of USP22 expression. Subsequently, we performed ChIP experiment and confirmed that knockdown of USP22 significantly reduced the PPAR γ enriched the promoter regions of *ACLY* and *ACACA*, however, knockdown of USP22 did not affect in the Flag-PPAR γ -5KR overexpression cells. In addition, the cell proliferation assays confirmed that knockdown of USP22 expression in PPAR γ -5KR overexpressing cells had a less inhibitory effect compared to Flag-PPAR γ transduced cells.

New Figure 5. USP22 promotes ACACA/ACLY transcription as well as cell proliferation via PPAR γ but not PPAR γ -5KR. a) ChIP analysis of PPAR γ or PPAR γ -5KR binding to the ACLY and ACACA promoters in MHCC-97H-PPAR γ + shUSP22-1/2 and MHCC-97H-PPAR γ -5KR + shUSP22-1/2 cells. qPCR was performed with primers specific to the PPAR γ -binding motifs. Data were normalized to the input. b) Proliferation assays were performed on MHCC-97H- PPAR γ + shUSP22-1/2 cells and MHCC-97H-PPAR γ -5KR + shUSP22-1/2 cells and OD values were calculated. The data shown represent the means (\pm SD) of biological triplicates. **p <0.01, ANOVA test. (Supplementary Fig. 7e and 7f in revised version)

9. The authors should comment in the text on the differences in the half-life of endogenous PPAR γ (Fig. 5i and Fig. 5j) in two stable cell lines. The WB panels of vinculin are overexposed for the quantification and should be substituted with less exposed panels.

RE: Thanks for your suggestions. We replaced the less exposed vinculin image for revised version (Fig. 5i and 5j in revised version). In addition, we commented the differences in the half-life of endogenous PPAR γ expression after 6 hours CHX treatment in revised version.

10. Do authors suggest that de-ubiquitination impacts nuclear retention of PPAR γ (Fig. 6a)? If it is what the authors wanted to stress, then they also should test nuclear/cyto distribution of PPAR γ K/R mutants (K156/157/169R). Moreover, it is surprising to find in this analysis (Fig. 6a) such a pronounced effect on PPAR γ protein levels given a depletion of USP22 (less than 30% decrease in expression). Moreover, these fractionation analyses of two cell lines show that PPAR γ is differentially distributed between Cyto and Nuc with MHCC-97H showing practically exclusive nuclear localization

of PPAR γ . It is an intriguing finding especially that IF analyses on Fig4f did not show much difference between two cell lines. How authors explain it?

RE: Thanks for your questions. As we and others showed that most of USP22 and PPAR γ localized nucleus, but this depends on the cell lines. In here, we speculate that USP22 deubiquitinates PPAR γ and this deubiquitination may stabilized and increased PPAR γ expression and impacted nuclear retention. Next, as we mentioned above K169/240/265/404/434 on PPAR γ are the deubiquitination sites of USP22. Subsequently, we generated Flag-PPAR γ -5KR transduced cell lines with simultaneously knockdown of USP22 expression. Western blot revealed that USP22 knockdown was not altered the expression of Flag-PPAR γ -5KR in the nucleus. And we replaced less exposed western blot image for Fig. 6a. Here we claim that knockdown of USP22 decreased PPAR γ expression in the nucleus, and the expression of protein patterns may slightly different between replicates experiments, however the effected trend should be the same. As we mentioned in manuscript most of experiments repeated at least three times in this study. In addition, we cannot directly compare the results of immunofluorescence and fractionated protein expression. In IF, we only examined the endogenous PPAR γ and USP22 expression and colocalization, without any modification of USP22 expression. Indeed, we clearly observed that PPAR γ is present in nucleus and cytoplasm in both cell lines.

New Figure 6. Western blot analysis of USP22 and Flag-PPAR γ ^{5KR} in nucleus proteins of MHCC-97H-Flag-PPAR γ -5KR+shUSP22 cells. (Supplementary Fig. 7a in revised version)

11.The conclusion that "USP22 significantly modulated PPAR γ transcriptional activity by DNA binding" is misleading (lines294-295). The Fig.6b is an ELISA based assay that assess if there is PPAR γ protein in the extract. The luciferase assay in combination with the ChIP would be more relevant approach to address this question.

RE: Thanks for your suggestions. We have edited the description in the revised version. And we also added a luciferase reporter gene experiment in new version. (Supplementary Fig. 7b in revised version).

New Figure 7. USP22 upregulates the transcriptional activity of PPAR γ . HEK293T cells were transfected with indicated plasmids. Cells were harvested after 24 hours transfection and the luciferase activity was measured by Promega Dual-Luciferase Reporter assay system. (Supplementary Fig. 7c in revised version)

12.The PPRE that the authors identified doesn't align with the classic PPRE sequence (Fig.6c). The evidence from the ChIP-Seq (published or in-house) should be provided showing the enrichment pick of PPAR γ in the gene regions of ACLY and ACACA. Also, the ChIP-qPCR assay and specifically the quantification that was used needs clarification (Fig.6d). What enrichment did authors observed in PPAR γ immunoprecipitates compared to negative controls (IgG beads)?

RE: Thanks for your suggestions. The consensus sequence of PPRE is AGGTCA, however not all PPAR γ regulating genes have classical PPRE sequence such as FGF1 (AGGTGA, *Nature*. 2012 May 17; 485(7398): 391–394.) and ERp44 (TCACTGATGGTGACAGTTGAGAT, *Endocrinology*, July 2010, 151(7):3195–320), but these PPRE contain similar sequences. The PPRE sequence of *ACLY* has been reported as *ACTCCAGGTCCCAAAG* (*Metabolism Clinical and Experimental* 110 (2020) 154302), and the consensus sequence could be the *AGGTCC*. Next, since *ACACA* has never been studied with PPRE, we predicted the PPRE element on the *ACACA* promoter by online website (<https://epd.epfl.ch/index.php>), and we found one potential PPRE element within 1000bp of *ACACA* promoter region. We reformed ChIP experiments and edited this in the revised version. In addition, we performed *ACACA* and *ACLY* promoter luciferase activity assay and provide this results in the revised version (Supplementary Fig. 7d in revised version). Please, refer to question #8 for ChIP-qPCR assay experiments and we standardized the PPAR γ and IgG immunoprecipitates by input.

New Figure 8. PPAR γ promotes promoter activity of ACACA and ACLY. ACACA or

ACLY promoter plasmid was transfected into HEK293T cells with or without transduced SFB-PPAR γ . Cells were harvested after 24 hours and promoter activity was measured by Promega Dual-Luciferase Reporter assay system. (Supplementary Fig. 7d in revised version)

13. The evidence that authors show to rule out Akt signaling upstream or downstream of USP22 in PPAR γ stabilization and its transcriptional activation are insufficient. It is not evident why authors conducted experiment in this set-up (serum deprivation for 24hours). Already the findings showed in Sup. Fig.7d suggest that USP22 might act upstream of pAkt (not clear what is the site that authors analyzed) which is in line with a recent report (34226501). It would be advisable to discuss at least in the discussion possible functional crosstalks between USP22, PPAR γ and insulin signaling in HCC that emanate from this study.

RE: Thanks for your suggestions. For this experiment we seeded same number of cells and after cell attachment deprived serum for 6 hours (for equilibration of metabolic homeostasis) then incubated with or without 10% serum media for 24 hours. As you mentioned in the question USP22 might acts upstream of p-AKT(S473) (here we detected this site), but with or without serum media incubation USP22 still regulates PPAR γ expression, therefore we are claim USP22 regulates PPAR γ stabilization independently from AKT activation in this study. We edited some description in the revised version. We also have discussed this in discussion section in the new version.

14. It is not clear how authors accessed steatosis in tissue arrays thus the correlation analyses with other histological stainings are difficult to evaluate (Fig.6i)

RE: Thanks for your suggestions. After the tissue microarray slides stained with HE, and the degree of steatosis was interpreted by the professional pathologist. And a chi-square test was then performed using data from previous immunohistochemistry.

15.The cell density in the cultures used for TG-Oil red analyses is different (Fig.2e). The lipid synthesis could be impacted by the cell number and also the Oil Red staining technically might influence it (especially washing-out steps). It would be advisable to do these analyses in cultures of the same density and to use the fluorescent die like Bodipy for evaluation.

RE: Thanks for your suggestions. We re-performed the oil red staining with same density of cell numbers instead of staining with Bodipy (due to COVID19 pandemic commercial unavailable recently) and replaced the images in the revised version. The results were essentially same as the last time.

New Figure 9. USP22 upregulates the accumulation of triglycerides in cells via PPAR γ .

Oil red staining assay in MHCC-97H-shUSP22 and MHCC-97L-USP22 cell lines. (Fig. 2e in revised version)

16. The correlative analyses of PPAR γ /USP22/lipogenic enzyme transcripts with survival are re-grouped and are easier to comprehend. However, there is no analysis of the triple correlation, thus it is not clear if it holds the route. It would be important (suggested in initial review) to analyze how these findings relate to known transcriptional groups in HCC (e.g. in 17187432).

RE: Thanks for your suggestions. We have analyzed triple correlation and found that simultaneously high expression levels of PPAR γ /USP22/ACC (ACLY) had worse overall survival in HCC cohorts (Supplementary Fig 8e). In addition, PPAR γ is positively correlated with ACC/ACLY in both TCGA database and microarray data (Supplementary Fig 8a and 8b). In addition, we have discussed our findings with specifically activated pathway of AKT in HCC in *Discussion* section.

New Figure 10. Correlation analysis of PPAR γ with ACC/ACLY. a) The correlation analysis of PPAR γ with ACC/ACLY protein expression in HCC tissue microarrays. b) The correlation analysis of PPAR γ with ACACA/ACLY mRNA expression in TCGA LIHC database. R represents the Pearson correlation coefficient. c) Kaplan–Meier curves of the survival analysis of USP22/PPARG/ACACA (or ACLY) copositive patients based on the prognosis data of TCGA HCC database. (Supplementary Fig. 8a, 8b and 8e in revised version)

17. Regarding the PPAR γ isoform analyses, although authors are right that, in physiology, the expression of PPAR γ 2 form is largely restricted to adipose tissue, it is also inducible form of PPAR γ that is found upregulated in cancer cells. As a minimum the authors have to clearly indicate in the text that they focussed exclusively on PPAR γ 1 form.

RE: Thanks for your suggestions. We have added the description of the PPAR γ isoforms in the *Introduction* section and highlighted that this study was conducted on PPAR γ 1.

18. The analyses of apoptosis (New Figure 12 that is not included in the manuscript) are insufficient to rule out the impact of USP22 OE or KD on cell survival and apoptosis. It is surprising that H line shows higher level of cleaved PARP compared to L line, the authors should comment on it in the section when they chose to use these two cell lines for their study. The conditions under which this experiment was conducted are not clear. The immunoblot analyses should be complemented with Tunnell assay in xenographs samples. Given the claim of potential therapeutic effect of USP22 inhibition in HCC, the authors should provide some insights to the mechanisms of this cancer growth inhibition.

RE: Thanks for your suggestions. As we mentioned above to avoid cell specificity we chose not only 97H and 97L cells, but also used SNU449, HUH7 and HepG2 cells conformed FA distribution and ACC or ACLY expressions. To further clarify whether USP22 is associated with apoptosis during tumor growth, we examined the apoptosis by Tunnell assay in the tumors formed from 97H-shCon and 97H-shUSP22-1 cells. The results showed that apoptosis rarely occurred in the tumors and USP22 knockdown did not affect the apoptosis of cells in tumors. Overall, fatty acids are fuels for tumor growing, in this study we identified a novel mechanism of USP22 deubiquitinates PPAR γ and subsequently increases lipogenic enzyme transcripts and FA accumulation. Therefore, inhibition of USP22 may benefit to tumors caused by lipidome accumulation.

New Figure 11. Tunnell assay with tumors formed from 97H shControl and shUSP22-1 cells. a) A representative picture of Tunnell assay. b) Statistical analysis of the number of positive cells per high magnification field of view for Tunnell assay was performed. Unpaired two-tailed Student's t test. The data shown represent the means (\pm SD) of biological three times repeat.

Minor comments:

-The exogenous pull-down is a confusing term (line 240) and from provided information the assay is the co-immunoprecipitation with transiently overexpressed Exogenous

proteins proteins.

RE: Thanks for your suggestions. We have revised in the new version: The specific interaction between USP22 and PPAR γ was confirmed by co-immunoprecipitation assay with exogenously transduced USP22 and PPAR γ in HEK293T cells.

-When authors describe polyubiquitination of PPAR γ in USP22 overexpressing cells, they mention on several occasions “inhibition of polyubiquitination” (e.g. line 254, line 276-277). This is confusing as it suggests USP22 acts upstream of PPAR γ ubiquitination (not addressed in this study).

RE: Thanks for your suggestions. We have revised in the new version: (line 254-256) In contrast, overexpression of USP22 increased deubiquitylation of PPAR γ but not the enzyme-dead mutant in MHCC-97L cells. Moreover, we found that USP22 significantly decreased the Lys-48-linked polyubiquitylation of PPAR γ . (line 276-277) We also found that USP22 did not deubiquitylate the five lysine sites mutated PPAR- γ 5KR (K169/240/265/404/434R) protein.

-Figure legends need editing to include sufficient information on the methodology and conditions used as well as statistical analyses that were conducted. Example of such scarcely described legend is Suppl. Fig.2.

RE: Thanks for your suggestions. We have revised Figure legends in the new version.

-The text needs editing to harmonize the use of plurals, prefer to use specific verbs instead of general (e.g. regulation, contribute, control, mediate). Example of the phrases that need editing to clarify the message is line 322-323 or line 327-329. The sub-headings should be edited to convey the conclusions of the sub-chapters, ex “PPAR γ contributes to USP22 mediated ACC and ACLY expression (line 285)”. The authors also should be specific in how they use the term “expression” (transcription, translation, transcript or protein levels, sometimes it is also applied to metabolite).

RE: Thanks for your suggestions. We have reworded the verbs in the text. From general verbs (e.g. regulate, contribute, control, mediate) to specific verbs (e.g. upregulate, downregulate, promote, inhibit). The description of line 322-323 was changed to: To determine whether the enhanced de novo synthesis of fatty acids promote cell growth, we modified the expression of PPAR γ or ACC in USP22-knockdown or ectopically expressed cells. The sub-headings of line 285 was changed to: USP22 increases ACC and ACLY expression by stabilizing PPAR γ . As we understanding in most of case Upright letters (e.g. USP22, PPAR γ , ACLY) indicate protein expression, *Italic letters* (e.g. *USP22*, *PPARG*, *ACLY*) indicate *mRNA* or gene expression. We edited incorrect descriptions in the revised version.

-As already mentioned in the initial review, in the text of the manuscript, the conclusions often come before the evidence in support. The example is the first paragraph, line 121-

123 claiming that lipid synthesis was significantly upregulated in HCC. There is also persistent issue with overstating in the conclusions e.g. line 318: "Collectively, PPAR γ as a transcription factor upregulates ACC and ACLY expression during USP22 mediated HCC progression". The HCC progression was not modeled in this study and the evidence in support that de-ubiquitination of PPAR γ by USP2 activates transcription of ACC and ACLY are weak.

RE: Thanks for your suggestions. The description of line 121-123 was changed to: These results confirmed that the contents of lipidome was significantly upregulated in HCC tissues, but what caused the abnormal upregulation of lipidome in HCC tissues remains unclear. The description of line 318 was changed to: Collectively, PPAR γ as a transcription factor upregulates ACC and ACLY expression under USP22 deubiquitination. Our work focuses on HCC tumor growth, so we changed progression to tumor growth.

-Calling of Fig.2j is missing (line189).

RE: Thanks for your suggestion. We added Fig. 2j at line 189.

-On Fig.3a the heatmap shows changes from -1 to +1. If it is log₂ fold change, it has to be indicated in figure legend.

RE: Thanks for your suggestion. We modified the legend of Fig3a.

REVIEWER COMMENTS

Reviewer #1 (Remarks to the Author):

I would like to thank authors for an extensive experimental work that they have conducted in the revision process. They have adequately addressed my concerns and I have no additional questions.